# Cross-presentation of dead cell-associated antigens shapes the neoantigenic landscape of tumor immunity

Kok Haw Jonathan Lim ●[1,2,3,4,5,17], Oliver Schulz[1,17], Irene Lobon ●[3], Tomas Castro-Dopico ●[1], Luis Zapata[6,7], Evangelos Giampazolias[1,14], Bruno Frederico[1,15], Carlos A. Castellanos ●[1], Michael D. Buck ●[1], William Stainier ●[1], Probir Chakravarty[8], Gavin Kelly ●[8], Neil C. Rogers ●[1], Ana Cardoso[1,16], Sonia Lee[1], Brian Vash[9], Stephanie Maiocco[9], Raj Mehta[10], Jessica Strid[2], Samra Turajlić[3,11,12,13] & Caetano Reis e Sousa ●[1]✉

Type 1 conventional dendritic cells (cDC1s) acquire and cross-present tumor antigens to prime CD8⁺ T cells. Whether this selects for specific neoantigens is unclear. DNGR-1 (CLEC9A), a cDC1 receptor for F-actin exposed on dead cells, promotes cross-presentation of cell-associated antigens. Here we show that DNGR-1-deficient mice develop chemically induced tumors more rapidly and at higher incidence, and these are more frequently rejected on transplantation into wild-type recipients. Whole-exome sequencing reveals enrichment of predicted neoantigens derived from mutated F-actin-binding proteins. Consistent with this observation, tethering model antigens to F-actin enhances DNGR-1-dependent cross-presentation. These results suggest that DNGR-1-mediated recognition of F-actin exposed by dead cancer cells favors priming of CD8⁺ T cells specific for cytoskeletal neoantigens, which can then drive immune escape of cancer cells lacking or reverting those mutations. Thus, neoantigen cross-presentation by cDC1 can determine the immune visibility of the tumor mutational landscape and sculpt cancer evolution by immunoediting.

Priming of anti-cancer cytotoxic CD8⁺ T cell depends largely on the proficient (neo-)antigen cross-presentation (XP) and cross-dressing capabilities of type 1 conventional dendritic cells (cDC1s)[1–3]. In mouse models, cDC1s are required for the rejection of immunogenic tumors[4–6] and the success of anti-cancer adoptive T cell therapy[6] and immune-checkpoint inhibition[7,8]. In human cancer datasets, abundance of cDC1s in the tumor microenvironment is associated with improved survival and with response to immunotherapy[9].

In both mice and humans, cDC1s can be universally identified by their selective high expression of DNGR-1 (a.k.a. CLEC9A)[10–15]. DNGR-1 binds to F-actin exposed by dead cells and allows cDC1s to detect necrotic cell debris, including from dead cancer cells. DNGR-1 signaling

upon recognition of dead cell cargo in cDC1 phagosomes can lead to phagosomal rupture, facilitating access of dead cell-associated (neo-) antigens to the MHC class I processing and presentation pathway[16]. This so-called cross-presentation results in tumor antigens being presented by cDC1 MHC class I molecules and, as such, differs from 'cross-dressing' in which the entire MHC class I-antigen complex is transferred from the cancer cell to the cDC1. Experiments with DNGR-1-deficient mice suggest that cDC1 XP can impact immunity to cytopathic viruses as well as cancer[17]. Nevertheless, DNGR-1-deficient mice are still able to control immunogenic transplantable tumors[18,19]. This is partly due to redundancy in XP pathways but also due to the activity of secreted gelsolin (sGSN), a plasma protein that competes for binding to F-actin

**Fig. 1 | Increased susceptibility of DNGR-1-deficient mice to chemical carcinogenesis. a**, Kaplan-Meier plot of tumor incidence in WT ($n = 59$), RAG1[KO] ($n = 43$), BATF3[KO] ($n = 11$) and DNGR-1[KO] ($n = 46$) mice. Pairwise comparison between groups was tested using log-rank (Mantel-Cox) test on Kaplan-Meier plot (right); no significant differences between DNGR-1[KO] versus BATF3[KO], or versus RAG1[KO] groups. Data were pooled from four independent experiments, except for the BATF3[KO] subgroup, which is from one experiment. **b**, Bar graph comparing the total number of tumors (left panel, $P = 0.0159$) and tumor burden quantitated in Fiji ImageJ (middle panel, $P = 0.0035$), in AOM-DSS colitis-associated carcinoma model in co-housed WT ($n = 7$) versus DNGR-1[KO] ($n = 7$) mice. Raw images of individual colon (right panel) of the WT versus DNGR-1[KO] mice, harvested on day 111. Data are represented as mean tumor burden ± standard error of the mean (s.e.m.), and groups were compared using unpaired $t$-test with Welch's correction. Data are from one experiment.

and acts as a natural inhibitor of DNGR-1 engagement[18]. Indeed, a role for DNGR-1-mediated XP-dependent control of transplantable tumors, spontaneous or therapy-induced, can be more readily revealed in mice lacking sGSN[18,19]. However, even in sGSN-sufficient mice, there might be a role for DNGR-1 in eliciting anti-cancer immunity that is masked by the use of rapidly growing transplantable tumors. By analogy, many mouse cancer cell lines grow similarly in wild-type (WT) and RAG1[KO] mice[20–22], yet T (and B) cells are clearly important for cancer immunity.

An important function of the immune system is to eradicate pre-malignant and malignant cells in the early phase of tumor development[23]. For this reason, the use of carcinogens rather than transplantable tumor cell lines in mice can offer a more refined experimental approach for assessing the importance of a given loss-of-function immune parameter on anti-cancer immunity. Using the chemical carcinogen 3-methylcholanthrene (MCA), Schreiber and colleagues demonstrated that primary cell lines derived from MCA-induced sarcomas in immunodeficient RAG2[KO] mice are highly immunogenic when

transplanted into secondary naïve WT recipients[24]. Their subsequent work[25] formed the basis for the immunoediting hypothesis wherein tumors that develop in immunocompetent WT hosts are 'edited' and enriched for clones that escaped immunosurveillance[26,27]. The use of carcinogens in mice has also been instrumental for demonstrating a role for γδ T cells[28,29] or NK cells[30,31] in cancer immunity. Here, we use chemical carcinogenesis models to show that XP by DNGR-1 has a function in cancer immunoediting, affecting the immunogenicity of tumors by selecting neoantigens associated with actin-binding proteins. Consistent with that notion, we show that DNGR-1-dependent XP preferentially selects F-actin anchored antigens.

## Results

### DNGR-1 restrains carcinogenesis
In humans, high expression of *CLEC9A* is associated with better overall survival (OS) and progression-free survival (PFS) outcomes across several cancers[32], irrespective of *sGSN* (Extended Data Fig. 1a,b). This

may reflect the role of DNGR-1 as a surrogate marker for cDC1 abundance in the tumor microenvironment, or it might indicate a more prominent role for DNGR-1 in cancer immunity in humans compared to mice bearing transplantable cancers. A key difference between the two is the length of time needed for tumor development. We hypothesized that cell death and selection occurring over a long period of time in chemical carcinogenesis mouse models is more likely to engage DNGR-1-dependent immunity than might be the case with transplantable tumors. We challenged WT C57BL/6 mice or DNGR-1[KO] (two different strains, *Clec9a*[cre/cre] or *Clec9a*[egfp/egfp]), RAG1[KO] or BATF3[KO] mice (all fully backcrossed onto a C57BL/6 background) with MCA and monitored tumor development. Notably, we observed that DNGR-1[KO] mice exposed to MCA developed tumors significantly earlier than WT mice, much like immunocompromised RAG1[KO] (which lack T and B cells) or BATF3[KO] (which lack cDC1s[4] and have additional immune defects[33]) (Fig. 1a and Extended Data Fig. 2a). The median time to tumor development was 95 days in DNGR-1[KO] mice compared to 113 days in WT mice (hazard ratio (HR) 1.72, 95% confidence interval (CI) 1.11–2.68, pairwise log-rank test $P = 0.008$) (Fig. 1a). Additionally, in a two-step model of colorectal carcinogenesis with azoxymethane (AOM) and dextran sodium sulfate (DSS), DNGR-1[KO] mice displayed significantly higher tumor burden compared to WT controls (Fig. 1b). Collectively, these data suggest that DNGR-1 is part of the immune barrier to development of chemically induced tumors.

## DNGR-1 links XP to immunoediting

We generated primary cell lines from MCA-induced fibrosarcomas and performed a series of transplantation experiments in secondary naïve syngeneic WT recipient mice (Fig. 2a). The majority ($n = 16/24$, 67%) of primary cell lines derived from tumors that had grown in DNGR-1[KO] mice, like those originating from immunodeficient RAG1[KO] mice ($n = 13/19$, 68%), were controlled and/or rejected when transplanted into naïve syngeneic WT recipient mice (Fig. 2b–e and Extended Data Fig. 2b–d). In contrast, only 42% ($n = 11/26$) of primary cell lines from tumors derived from MCA-treated WT mice were rejected upon secondary transplantation (Fig. 2b–e and Extended Data Fig. 2b–d). Together with the initial observation of shorter latency for tumor development, this finding suggests that tumors derived from DNGR-1[KO] mice are more immunogenic than those from WT mice, that is, display less immunoediting.

We confirmed that control of the regressor tumors derived from DNGR-1[KO] mice was immune dependent: the cancer cells expressed MHC class I and were able to grow unimpeded in secondary RAG1[KO] recipient mice, as well as in WT mice depleted of CD8[+] T cells (Extended Data Fig. 3a–e). However, like our previous findings with transplantable tumor models[18,19], all regressor cell lines were controlled and/or rejected in DNGR-1[KO] secondary recipient mice as effectively as in WT mice (Extended Data Fig. 3d,e). Interestingly, where tested, rejection was lost in BATF3[KO] mice (Extended Data Fig. 3e), as expected[4]. This finding suggests that DNGR-1 is dispensable for the immune-mediated control and/or rejection of transplantable primary cell lines, including ones derived from DNGR-1[KO] primary hosts, even if it shapes the immunogenicity of these tumors during their primary development.

## Mutational landscape of primary fibrosarcomas

To assess the impact of DNGR-1 on the cancer mutational landscape, we performed whole exome sequencing of DNA extracted from primary cell lines derived from each of the tumors in one representative experiment consisting of $n = 43$ individual mice: DNGR-1[KO] ($n = 14$), RAG1[KO] ($n = 11$) and WT ($n = 18$). We observed that the overall tumor mutational burden, as well as mutation burden across different categories (missense, truncating, synonymous and indels), was not substantially different across strains, except for a lower burden of indels in WT host-derived cells compared to RAG1[KO] host-derived cells (Wilcoxon test, $P = 0.05$) (Extended Data Fig. 4a,b). However, when comparing copy number

alterations, we found a significantly higher burden of copy-number deletions in WT-derived tumor cells compared to those from RAG1[KO] or DNGR-1[KO] mice (Extended Data Fig. 5a). There were no differences in copy-number gains among genotypes (Extended Data Fig. 5b). Significant gains on chromosome 15 covering a region containing the gene *Myc* were enriched in progressor cell lines ($n = 13/22$, 59%) compared to regressor cell lines ($n = 4/21$, 19%) (Fisher's exact test, $P = 0.01$) (Extended Data Fig. 5c–e).

## Nature of immunoediting by DNGR-1

To analyze immunoediting at the level of the mutational landscape, we extracted all putative strong neoantigens for all the primary sarcoma cell lines by using pVACtools[34] to predict mutant peptides with high affinity ($IC_{50} < 150$ nM) for H-2K[b] and H-2D[b]. Using this approach, we did not observe an overall difference in prevalence of strong neoantigens across cell lines from regressor compared to progressor tumors (Fig. 3a,b) or in tumors from different host backgrounds (Fig. 3c).

Our previous analysis of datasets in human cancers suggested a preferential role for sGSN in decreasing CD8[+] T cell priming against tumor neoantigens associated with F-actin[18]. Therefore, we assessed whether mouse primary fibrosarcomas from DNGR-1-deficient versus DNGR-1-sufficient mice might differ in a subset of predicted strong neoantigens specifically derived from F-actin binding proteins (FABP) ($n = 88$, Extended Data Table 1). Notably, we found that this was indeed the case: there was a clear enrichment in predicted FABP neoantigens in tumors derived from DNGR-1[KO] or RAG1[KO] mice when compared to cancers from WT mice (Fig. 3d). As a specificity control[18], we examined predicted strong neoantigens derived from microtubule binding proteins (MBP) ($n = 98$, Extended Data Table 2), a cytoskeletal component distinct from F-actin. Predicted MBP neoantigens were prevalent across all genotypes and were not enriched in tumors derived from DNGR-1[KO] mice (Fig. 3e).

## Immunogenicity of selected FABP neoantigens

We further examined individual FABP neoantigens and noted that there were 22 genes with mutations predicted to create strong neoantigens across multiple cell lines (Fig. 3f). To assess the immunogenicity of these putative neoantigens, we focused on those derived from DNGR-1[KO] regressor sarcoma lines. Thirteen peptides were synthesized and tested for their ability to bind to MHC class I using an RMA-S stabilization assay[35] (Extended Data Table 3). As positive controls, we used SIINFEKL (binder to H-2K[b]), derived from the model antigen ovalbumin (OVA), or 'mut spectrin' (binder to H-2D[b]), a peptide derived from mutated spectrin-β2 (ref. 25). We found that the majority of the putative neoantigen peptides (9 out of 13) indeed bound to either H-2K[b] or H-2D[b], as predicted (Fig. 4a). Next, we immunized mice with antigen-presenting cells (from cultures of bone marrow in FLT3L, enriched for cDCs), each pulsed with one of the nine peptides. Notably, re-stimulation of splenocytes from immunized mice with the corresponding peptides revealed a positive response to two of the nine peptides (88D4a and 88D12a; Fig. 4b) that was comparable in magnitude to the response to mut spectrin (Fig. 4c). Thus, at least some of the predicted neoantigen peptides are immunogenic, validating our bioinformatics pipeline to identify putative tumor antigens.

Interestingly, in one of the regressor cell lines derived from a DNGR-1[KO] host (88D2), we found two independent predicted strong neoantigens derived from mutations in a single FABP gene, *Sptb*, which encodes spectrin-β (Fig. 3f). This was reminiscent of the aforementioned mutated spectrin-β2 ($R_{913} \rightarrow L$), previously described by Schreiber and colleagues as a rejection neoantigen subject to immunoediting in an MCA-induced primary tumor from *Rag2*[–/–] mice[25]. To connect our work (in C57BL/6 mice) to the earlier study (in 129/Sv mice), we generated (C57BL/6-derived) MCA205 fibrosarcoma cell lines expressing high or low levels of WT or mutant $R_{913} \rightarrow L$ spectrin-β2 (Extended Data Fig. 6a,b) and inoculated them into WT or into sGSN[KO]

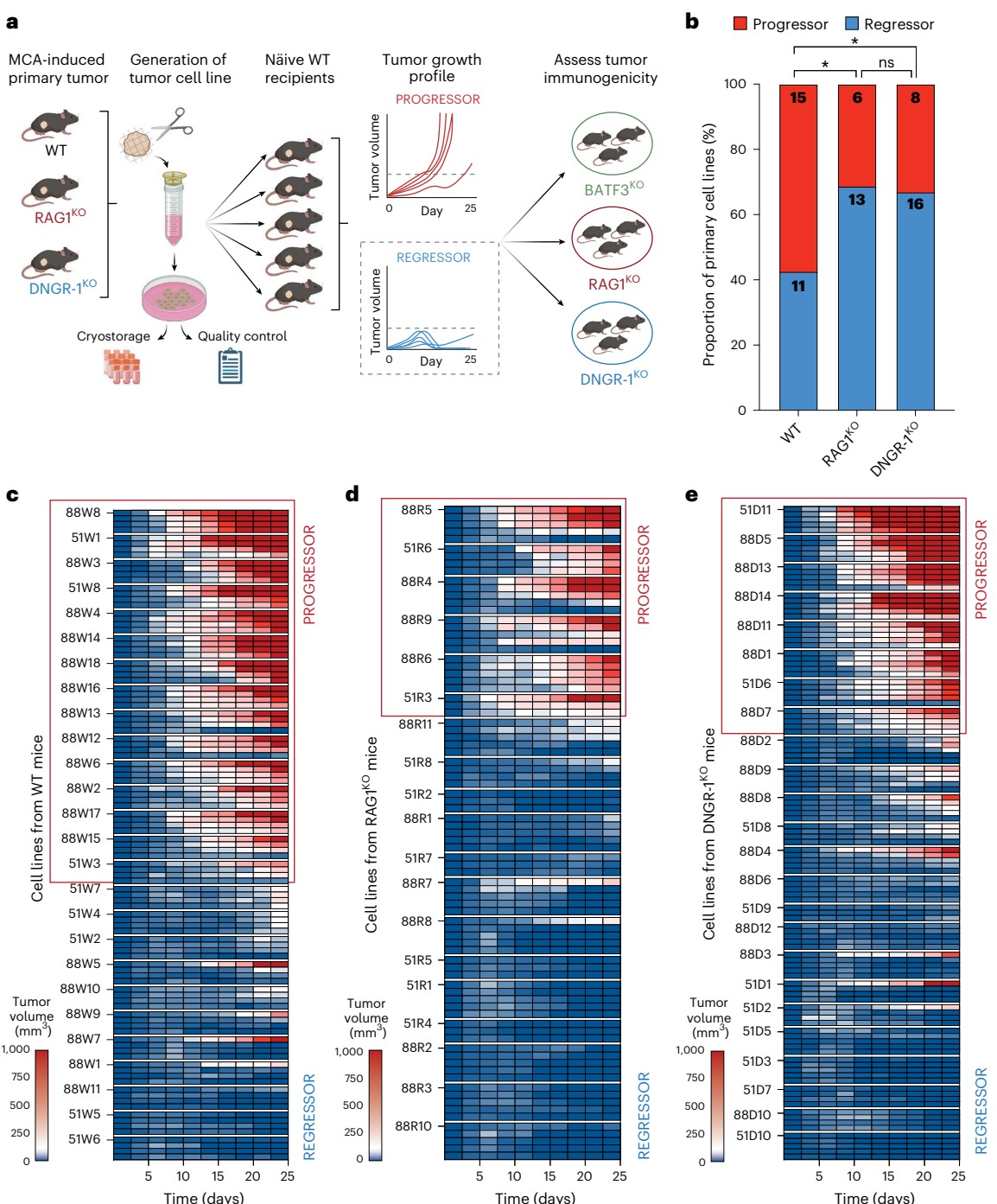

**Fig. 2 | Tumor cell lines from MCA-treated DNGR-1-deficient mice are highly immunogenic. a**, Experimental setup to investigate the growth phenotype and immunogenicity of primary tumor cell lines. **b**, Bar graph summarizing the proportion of regressors versus progressors: RAG1[KO] versus WT, OR = 2.96 (95% CI 0.80–9.18), $P$ = 0.04; DNGR-1[KO] versus WT, OR = 2.73 (95% CI 0.86–8.24), $P$ = 0.04. **c–e**, Heatmap summarizing the growth profile of each of the primary cell lines derived from tumors in MCA-treated WT ($n$ = 26) (**c**), RAG1[KO] ($n$ = 19) (**d**) and DNGR-1[KO] ($n$ = 24) mice (**e**), classified as progressors or regressors, following challenge in secondary WT hosts ($n$ = 3–5 mice per cell line). The proportion of regressors versus progressors in **b** were compared using one-tailed chi-square test. Where indicated, *$P$ ≤ 0.05; ns, not significant. OR, odds ratio. Schematic in **a** created with BioRender.com.

mice, in which DNGR-1 triggering is enhanced[18]. Tumors with high expression of mutant spectrin-β2 were controlled similarly in sGSN[KO] and WT mice (Fig. 4d, left). However, tumors with low expression of mutant spectrin-β2 were preferentially controlled in sGSN[KO] hosts (Fig. 4d, right), in a manner that depended on DNGR-1 (Fig. 4e). Collectively, these data indicate that DNGR-1 is important for increasing the antigenic visibility of tumors with low FABP neoantigen load. By salvaging F-actin-associated proteins for XP, the DNGR-1 pathway preferentially primes CD8[+] T cells against FABP neoantigens and thereby leads to immunoediting of the antigenic repertoire of cancer cells.

## F-actin-anchored antigens are superior substrates for DNGR-1-dependent XP

Given the above findings, we hypothesized that antigen association with F-actin within dying cells increases the efficiency of XP by cDC1s

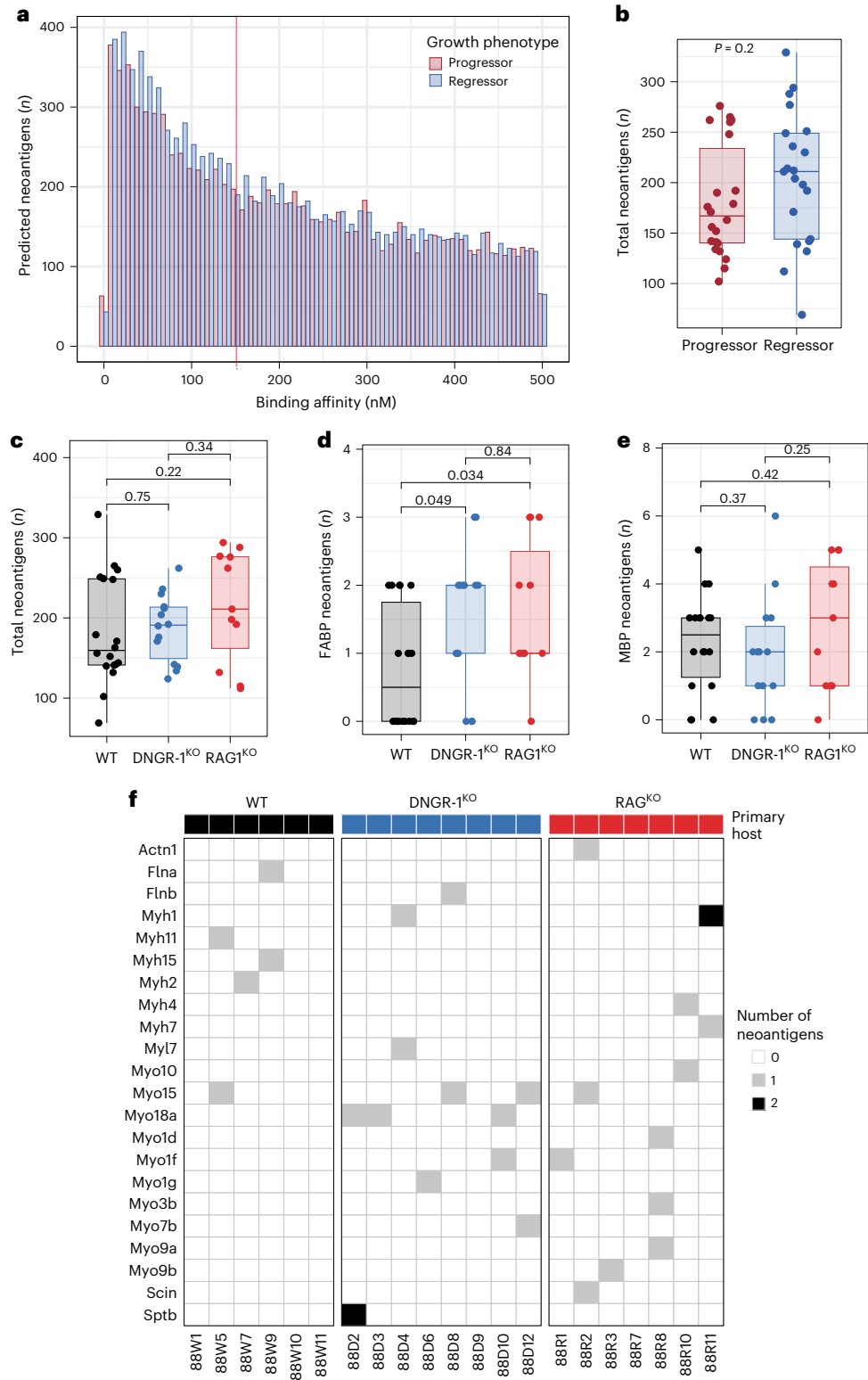

**Fig. 3 | Mutated FABP-derived neoantigens are enriched in primary tumor lines from DNGR-1-deficient mice. a**, Histogram showing the number of predicted neoantigens according to binding affinity (IC$_{50}$) scores, stratified according to progressor and regressor cell lines. **b**, Box and whisker plots comparing the total number of predicted strong neoantigens (binding affinity IC$_{50}$ < 150 nM) between progressor ($n = 22$) and regressor ($n = 21$) cell lines (two-sided Mann-Whitney U-test $P = 0.2$). Box plots display the median (central line) and the first and third quartiles (lower and upper box edges). Whiskers extend to 1.5× the interquartile range from each quartile. **c–e**, As in **b**, box and whisker plots comparing the number of predicted strong neoantigens (binding affinity IC$_{50}$ < 150 nM) between

cell lines from different host background (WT, DNGR-1$^{KO}$, RAG1$^{KO}$ with $n = 18$, $n = 14$ and $n = 11$, respectively) in all (WT versus DNGR-1$^{KO}$, $P = 0.75$; WT versus RAG1$^{KO}$, $P = 0.22$; WT versus RAG1$^{KO}$, $P = 0.34$) (**c**), FABP (WT versus DNGR-1$^{KO}$, $P = 0.049$; WT versus RAG1$^{KO}$, $P = 0.034$; WT versus RAG1$^{KO}$, $P = 0.84$) (**d**), and MBP genes (WT versus DNGR-1$^{KO}$, $P = 0.37$; WT versus RAG1$^{KO}$, $P = 0.42$; WT versus RAG1$^{KO}$, $P = 0.25$) (**e**). The only significant differences are between the number of predicted FABP neoantigens in DNGR-1$^{KO}$ or RAG1$^{KO}$ lines versus WT. **f**, Heatmap showing strong neoantigens predicted within the FABP genes in immunogenic regressor cell lines, clustered by primary host background of origin.

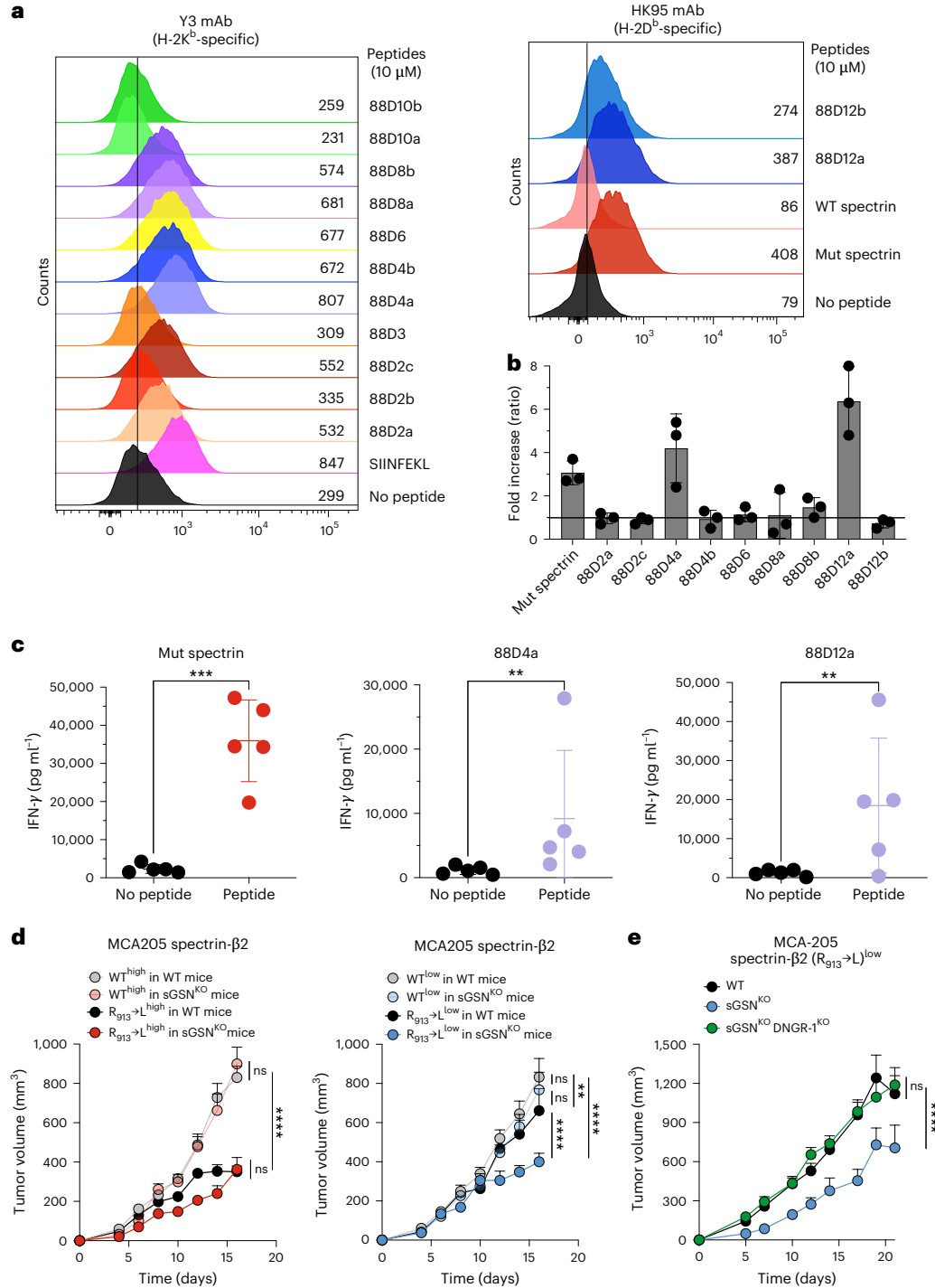

**Fig. 4 | Validation of MHC-I binding and immunogenic potential of neoantigen peptides from mutated FABPs. a**, Overlay histograms showing MHC-I binding for H-2K$^b$ (left graph) and H-2D$^b$ (right graph)-specific neoantigen peptides. Numbers next to histograms represent GeoMean fluorescence intensity. **b**, Ex vivo re-stimulation assay to assess the immunogenicity of neoantigen peptides. Scatter plot with bar showing normalized IFN-γ production of splenocytes from mice ($n$ = 3) immunized with neoantigen peptide-pulsed Flt3L BM-DC following restimulation with the corresponding peptide relative to cultures without peptide. The horizontal line represents a fold increase of 1; that is, induction of a peptide-specific CD8 T cell response was undetectable. Data shown are mean ± standard deviation (s.d.) of three biological replicates. **c**, Immunogenic peptides from **b** were retested in a separate experiment. Graphs show IFN-γ production of splenocytes from immunized mice after re-stimulation with neoantigen peptides derived from mutated spectrin β2 (left), mutated

myosin light chain 2 (88D4a, middle) or mutated myosin 7b (88D12a, right). Data in scatter plots represent five biological replicates. Also shown are the mean ± s.d. $P$ values were analyzed using a ratio paired t test: mut spectrin ($P$ = 0.0001), 88D4a ($P$ = 0.003) and 88D12a ($P$ = 0.0047). **d**, Tumor growth profile of either MCA205 mutant spectrin-β2 ($R_{913}$→L)$^{high}$ versus MCA205 spectrin-β2 WT$^{high}$ implanted into sGSN$^{KO}$ ($n$ = 9 and $n$ = 10 mice, respectively) and WT ($n$ = 11 and $n$ = 10 mice, respectively) (left) or MCA205 mutant spectrin-β2 ($R_{913}$→L)$^{low}$ versus MCA205 spectrin-β2 WT$^{low}$ implanted into sGSN$^{KO}$ ($n$ = 10 and $n$ = 10 mice, respectively) and WT ($n$ = 11 and $n$ = 12 mice, respectively) (right). Data are from one experiment. **e**, Tumor growth profile of MCA205 mutant spectrin-β2 ($R_{913}$→L)$^{low}$ implanted into sGSN$^{KO}$ ($n$ = 16), sGSN$^{KO}$ DNGR-1$^{KO}$ ($n$ = 19) mice, and WT ($n$ = 17) mice. Data are representative of two independent experiments. Mean tumor volumes ± s.e.m. in **d,e** were compared using the Bonferroni-corrected two-way ANOVA. Where indicated, ****$P$ < 0.0001, **$P$ = 0.0055, ns, not significant.

via the DNGR-1-dependent pathway. To test this hypothesis, we fused a His-tagged non-secreted version of OVA with either the actin-binding peptide LifeAct (LA)[36] or a mutated version of LA (mutLA) that cannot bind to F-actin[37] (Extended Data Fig. 7a). We transfected these constructs stably into HeLa cells and verified that the OVA fusion proteins were expressed at similar levels and in intact form and that LA-OVA but not mutLA-OVA co-localized with F-actin (Extended Data Fig. 7b–d). After induction of cell death by UV irradiation, both LA-OVA and mutLA-OVA were retained in cell corpses albeit to slightly different degrees, in contrast to red fluorescent protein (RFP), a co-expressed cytoplasmic protein that was largely lost (Extended Data Fig. 7d). We observed a similar pattern of antigen retention using OVA fused to an F-actin-specific affimer[38,39] as an alternative strategy (Extended Data Fig. 7e).

We fed necrotic HeLa cells expressing LA-OVA or mutLA-OVA to immortalized mouse splenic cDC1 cells (Mutu DCs[40]) and assessed XP by measuring the amount of IFN-γ in the supernatant after overnight co-culture with preactivated H-2K$^b$/OVA-specific OT-I T cells[41]. We found that dead cell-associated LA-OVA was much more efficiently cross-presented compared to mutLA-OVA (Fig. 5a). XP of LA-OVA dead cells could be completely blocked by an anti-DNGR-1 mAb[10] but not an isotype-matched control mAb (Fig. 5b). Furthermore, Mutu DCs expressing a mutant DNGR-1 that cannot bind to F-actin (2 W Mutu DC[42]) were able to cross-present soluble cell-free OVA but failed to cross-present LA-OVA dead cells (Fig. 5c). XP of dead cells containing mutLA-OVA, albeit much lower than that of LA-OVA cells (note y axis scale), remained DNGR-1-dependent, as it was also abolished by blocking DNGR-1 and lost in Mutu DCs expressing a mutant DNGR-1 (Fig. 5b,c).

To ensure that these observations were not unique to LA-mediated anchoring, we repeated the experiments with dead HeLa cells expressing F-actin-affimer-OVA (Aff-OVA). Again, Aff-OVA dead cells were superior substrates for XP compared to dead cells expressing OVA fused to the control affimer (CtrlAff-OVA) and were comparable to cells expressing LA-OVA (Fig. 5d). Finally, we confirmed our findings in non-immortalized cDC1 by carrying out XP assays with primary cDC1s purified from FLT3L bone marrow cultures (Fig. 5e).

Lastly, we immunized mice with dead HeLa cells expressing LA-OVA mixed with poly I:C, a strong adjuvant. Notably, we observed a greater expansion of endogenous OVA-specific (tetramer$^+$) CD8$^+$ T cells (Fig. 5f) and of OVA-restimulated IFN-γ$^+$ effector CD8$^+$ T cells (Fig. 5g) in mice immunized with dead cells expressing LA-OVA compared to mutLA-OVA. This difference in cross-priming of CD8$^+$ T cells induced by dead LA-OVA cells versus mutLA-OVA cells was further magnified in mice lacking sGSN (Fig. 5f,g). We conclude that dead cell-associated antigen that is anchored to F-actin is more efficiently cross-presented by cDC1 in a DNGR-1-dependent manner to efficiently elicit CD8$^+$ T cell responses in vivo.

## Discussion

Presentation of neoantigens by cancer cells underlies CD8$^+$ T cell-mediated cancer immunity. However, the initial priming and subsequent restimulation of cancer-specific T cells is mediated by an antigen-presenting cell, most often a cDC1 (refs. 1,9,43). How cDC1s acquire tumor antigens for presentation to T cells remains poorly understood. Transfer of preformed MHC:peptide complexes from cancer cells to cDC1 allows for the most faithful representation of the tumor antigenic repertoire[3,44,45]. However, other processes are likely at play, including XP, which involves capture of antigenic material from tumors and subsequent processing by cDC1. It is thought that dead cancer cells and tumor debris serve as antigen sources for XP by cDC1, but it remains unclear whether the handling and processing of such antigens imposes an antigen selection bias that can impact the epitope repertoire of effector anti-tumor T cells. Here, we addressed this issue by examining the immunoediting signatures of cancers from DNGR-1-deficient mice in which XP of dead cell-associated antigens is attenuated. Our results indicate that loss of DNGR-1-dependent XP preferentially reduces priming of FABP neoantigen-specific T cells without markedly affecting the response to other neoantigens. These T cells are functionally relevant for tumor control, as they 'force' the selection of immunoedited cancer cell 'escapees' that have lost the mutated FABP epitopes. Because loss of DNGR-1 results in less priming of FABP-specific T cells, there is correspondingly less immune pressure on tumors in DNGR-1$^{KO}$ mice to 'get rid of' FABP mutations. Consequently, the escapees are detected to a greater degree in WT mice compared to DNGR-1$^{KO}$ mice. It is important to note that the effect of DNGR-1 on the antigenic visibility of FABPs neoantigens is not 'all-or-none' but quantitative and further modulated by sGSN. This can be demonstrated by our reconstruction experiments using a cancer cell line bearing different levels of mutant spectrin-β2, a canonical FABP neoantigen[25].

In our experiments, around 40% of the tumors from WT C57BL/6 mice were rejected after transplantation, which suggests a lower level of antigen immunoediting than previously reported[24,26,27]. However, many of those earlier experiments were done with FVB or 129/Sv mice, which are more permissive for carcinogenesis and may therefore allow for greater immunoediting. Importantly, loss of rejection antigens is only one mechanism for immunoediting and tumors expressing immunogenic neoantigens can hide from the immune system by other means, including downregulating expression of rejection antigens and/or MHC molecules, producing immunosuppressive factors or co-opting regulatory T cells and suppressive myeloid cells[27,46]. Although our comparisons revealed a statistically significant increased prevalence of predicted FABP neoantigens in tumors from DNGR1$^{KO}$ mice, the dataset is relatively small and our functional testing covered only a subset of predicted candidates. Some peptides that scored negative in our in vivo vaccination protocol might still be immunogenic in the tumor context, and we did not assess potential downregulation of FABP transcripts as an alternative mechanism of immune escape. Such caveats mean our analysis likely underestimates, rather than overestimates, the contribution of FABP-specific immunoediting. Taken together, our findings support a role for DNGR-1-dependent XP in shaping the visibility of

**Fig. 5 | Antigen anchoring to F-actin in dead cells potentiates DNGR-1-dependent XP by cDC1s. a–e,** Stimulation of pre-activated OT-I CD8$^+$ T cells after overnight co-culture with Mutu DCs or Flt3L cDC1 and necrotic HeLa cells expressing the indicated fusion proteins. Graphs show concentration of IFN-γ in the supernatant of cultures. **a,** Comparison of the ability of LA-OVA- and mutLA-OVA-expressing HeLa cells to stimulate OT-I T cells following presentation of dead cell-associated antigen by Mutu DC. **b,** XP of cells as in **a** in the presence of anti-DNGR-1 (1F6) or isotype-matched control antibody (MAC49). **c,** Comparison of DNGR-1-deficient (KO) Mutu DC complemented with WT DNGR-1 (C9A/KO) or double tryptophan mutant (2 W/KO) DNGR-1 for ability to cross-present dead LA-OVA or mutLA-OVA cells. **d,** Comparison of F-actin-specific affimer-OVA (Aff-OVA) and control affimer OVA (CtrlAff-OVA) dead cells with LA-OVA dead cells. **e,** XP of (mut)LA-OVA cells by Flt3L cDC1. Plotted data represent mean of technical duplicates (**a**–**c**) or mean ± s.d. of technical triplicates (**d,e**).

Data are representative of three (**a**–**c**) and two (**d,e**) independent experiments, respectively. **f,g,** Induction of OVA-specific CD8$^+$ T cell responses following immunization of WT or sGSN$^{KO}$ mice with dead LA-OVA or mutLA-OVA cells mixed with poly I:C (25 μg per mouse). Scatter plots show data pooled from three (**f**) or two (**g**) independent experiments, and the number of biological replicates in each group ranges from n = 4 to 13. P values shown for pairwise comparison of groups in each graph were analyzed using ordinary one-way ANOVA and Tukey's multiple comparison test. **f,** Percentage (left) of H-2K$^b$/OVA pentamer$^+$ CD44$^+$ double-positive cells of total CD8$^+$ T cells and their absolute number (right). **g,** Percentage (left) of IFN-γ$^+$ CD44$^+$ double-positive cells of total CD8$^+$ T cells and their absolute number (right). Where pairwise comparisons are statistically significant, exact P values have been annotated on the graph, and where indicated, ns, not significant.

FABP neoantigens but also highlight layers of complexity that remain to be dissected in larger datasets and with complementary approaches. This includes analysis of a larger number of regressor versus progressor tumors across different genetic backgrounds.

We further present evidence that anchoring antigens to the DNGR-1 ligand F-actin greatly augments the ability of dead cells to serve as antigen donors for XP by cDC1s and for CD8+ T cell cross-priming in vivo. This suggests that F-actin-associated neoantigens in tumor

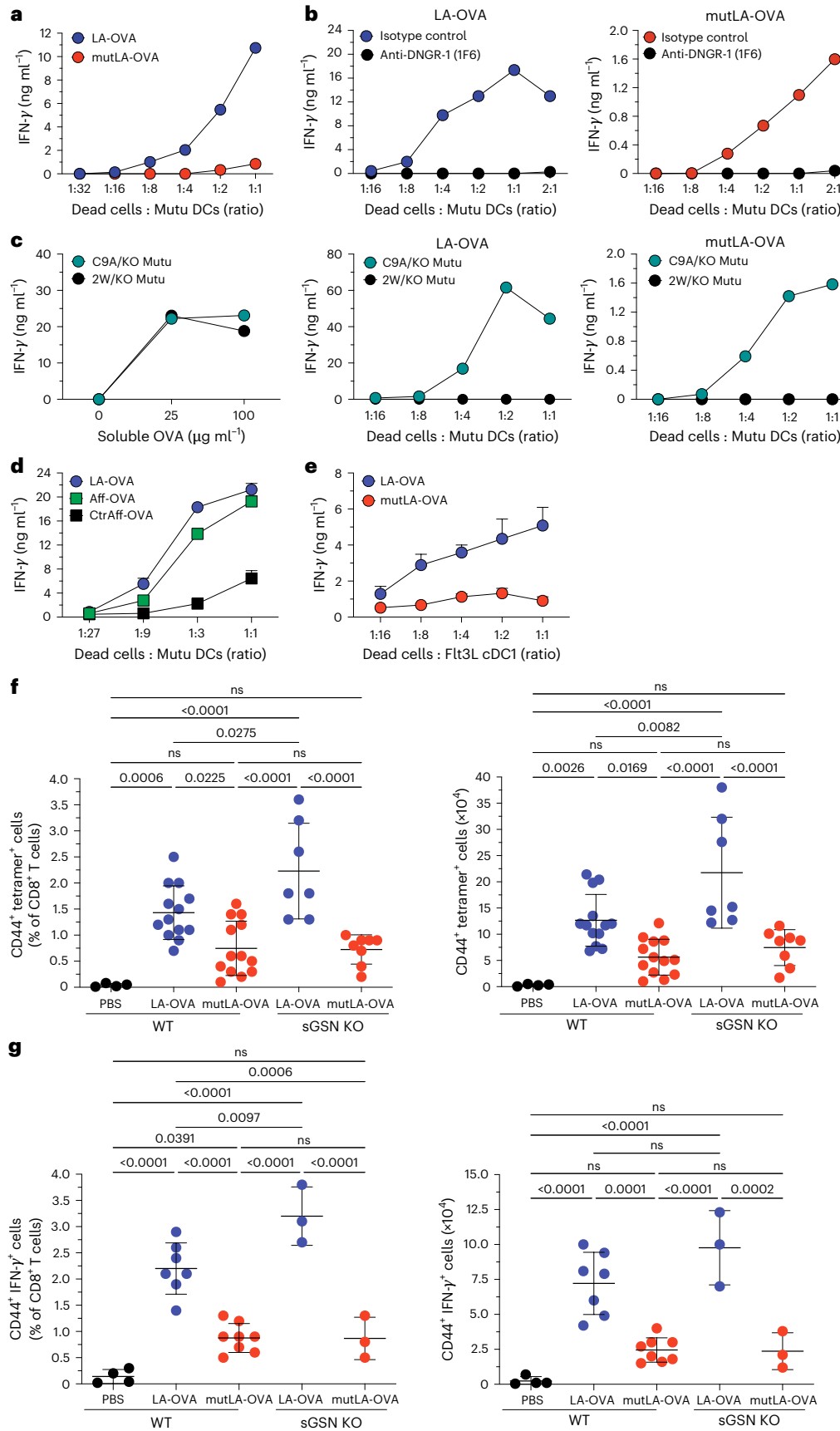

cells could bolster anti-cancer immunity and builds on our recent work showing that a transplantable tumor expressing LA-OVA as a model neoantigen is highly immunogenic, especially in mice lacking sGSN[18]. The notion that XP is affected by antigen localization within donor cells has been underexplored but is also supported by a recent study that used stable isotope labeling by amino acids in cell culture combined with an immunopeptidomic approach to directly identify and quantitate cross-presented peptides eluted from MHC class I complexes on DCs[47]. Anchoring to F-actin likely permits increased antigen retention in dead cell debris while also allowing more efficient engagement of the DNGR-1-dependent XP pathway during debris uptake by cDC1.

Interestingly, T cells specific for mutated FABPs have been found not only in mice but also in patients with cancer, underscoring the relevance of this class of neoantigens for cancer immunity[25,48–50]. Consistent with this notion, the prevalence of mutations in FABPs correlates with better outcomes in patients with cancer bearing low levels of sGSN transcripts in tumors and, presumably, greater DNGR-1-dependent XP[18]. Although we have not tested XP by human cDC1s, we believe that our findings are in principle translatable to humans, given the similarity of DNGR-1 and of other key components in the DNGR-1 XP pathway across species. As such, the physical anchoring of some (neo) antigens to F-actin might not only increase their immune visibility but potentially act as a determinant of the immunodominance for CD8[+] T cell responses to human cancers. This finding has potential implications for optimizing protocols for studying immune responses to cell-associated antigens and for potentiating cDC1-mediated anti-tumor immunity.

Although mice deficient in BATF3 and lacking cDC1s display severe defects in T cell responses to tumors, this is likely not solely due to defective XP but to loss of additional cDC1 and T cell functions in cancer immunity[9,51]. There are only a few studies of the role in anti-cancer immunity of proteins involved in cDC1-mediated XP. Mice with antigen-presenting cells deficient in SEC22B, a vesicle-trafficking protein, have been reported to display reduced ability to control immunogenic transplantable tumors[52]. Similarly, WDFY4, a protein predicted to be involved in vesicular formation and transport, was found to be necessary for XP by cDC1 and for immunity to transplantable fibrosarcoma[53]. We hypothesize that, like DNGR-1, these components of the cDC1 XP machinery might also shape the antigenic visibility of tumors to the CD8[+] T cell compartment. However, beyond proof-of-principle mouse models, it will be important to establish to what extent XP shapes the immune visibility of tumors in clinical settings. Deciphering the nature of immunoediting induced by cross-priming in human cancers, spontaneously and in response to immune and conventional therapies may further inform the dynamics of cancer evolution.

## Online content

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

[1]Immunobiology Laboratory, The Francis Crick Institute, London, UK. [2]Department of Immunology and Inflammation, Imperial College London, London, UK. [3]Cancer Dynamics Laboratory, The Francis Crick Institute, London, UK. [4]Division of Cancer Sciences, Faculty of Biology, Medicine and Health, The University of Manchester, Manchester, UK. [5]Advanced Immunotherapy and Cell Therapy Team, The Christie NHS Foundation Trust, Manchester, UK. [6]Centre for Evolution and Cancer, Institute of Cancer Research, London, UK. [7]Center for Mathematical Modelling, Universidad de Chile, Santiago, Chile. [8]Bioinformatics and Biostatistics, The Francis Crick Institute, London, UK. [9]Apple Tree Partners, Cambridge, USA. [10]Adendra Therapeutics Ltd., London, UK. [11]Skin and Renal Unit, The Royal Marsden NHS Foundation Trust, London, UK. [12]Cancer Dynamics Laboratory, Cancer Research UK Manchester Institute, The University of Manchester, Manchester, UK. [13]The Christie NHS Foundation Trust, Manchester, UK. [14]Present address: Cancer Immunosurveillance Group, Cancer Research UK Manchester Institute, The University of Manchester, Manchester, UK. [15]Present address: Early Oncology Research and Development, AstraZeneca, Cambridge, UK. [16]Present address: Medical Department, ADM Health and Wellness, London, UK. [17]These authors contributed equally: Kok Haw Jonathan Lim, Oliver Schulz. ✉e-mail: caetano@crick.ac.uk

## Methods

### Mice

RAG1[KO] (*Rag1*[−/−]), BATF3[KO] (*Batf3*[−/−]), DNGR-1[KO] (*Clec9a*[gfp/gfp] or *Clec9a*[cre/cre]) and WT mice on a C57BL/6 background were bred and maintained in the Biological Research Facility (BRF) at The Francis Crick Institute. Mice were held in individually ventilated cages (with a maximum of five animals per cage) within the individually ventilated cage green line system (Techniplast). Air movement was regulated by an air handling unit (Techniplast) under negative pressure (75 ACH/−20%). All cages were supplied with an automatic watering system, providing RO chlorinated to 2.45 ppm. The animals were kept under a 12 h light/dark cycle (7 a.m.–7 p.m. including dawn and dusk settings of 15 min), with a room temperature of 20 to 24 °C and humidity of 55% (±10%). The animals had ad libitum access to water and food, and were fed with Teklad Global Rodent Diet Sterilized 2018S (18% protein) (Inotiv).

DNGR-1 EGFP KI mice were generated in 129S6/C57BL/6 F1 ES cells and the expression of NK1.1 (the Klrb1c gene 'a' allele) was linked to Clec9a deficiency in early backcross KO mice indicating that the homologous recombination step targeted the chromosome of C57BL/6 origin. This strain has subsequently been backcrossed a total of 20 times to C57BL/6 J before crossing again to homozygosity. Genetic monitoring (carried out at Transnetyx) confirmed that three N20 mice (one each at generations F4, F5 and F6) were 100% BL/6 (versus 129) when tested using a panel of 120 distinguishing SNPs. Three F6 mice were also 100% BL/6 J (versus BL/6 N) when tested using a panel of 24 distinguishing SNPs. DNGR-1 iCre KI mice were generated in the Primogenix PRX-B6N ES cell line. The strain was backcrossed 10 times to C57BL/6 J before crossing again to homozygosity. Four N10F11 animals were tested as 99.9–100% BL/6 J by the MiniMUGA 10 K SNP panel at Transnetyx.

In all experiments, both male and female mice were used, unless otherwise indicated. Experiments were commenced when mice were between 6 and 14 weeks old. In all loss-of-function experiments comparing DNGR-1[KO] to WT strains, mice were co-housed with sex- and age- matched WT controls for at least 3 weeks to eliminate any microbiota-dependent effects. All animal experiments were performed upon prospective approval of a study plan by the BRF at The Francis Crick Institute and strictly adhered to the Animals (Scientific Procedures) Act 1986.

### Cells

Transplantable tumor cell line MCA205 expressing either WT or mutant ($R_{913}$→L) mouse spectrin-β2 was achieved using the sleeping beauty transposon system. WT spectrin-β2 in pMSCV vector was a gift from R. D. Schreiber[25]. In brief, WT spectrin-β2 was cloned into the pSBbi-GFP-hygromycin (GH) resistant vector (Addgene, 605414). A point mutation (2738 G→T) was introduced to generate the spectrin-β2 ($R_{913}$→L) using the QuikChange Lightning Site-Directed Mutagenesis Kit (Agilent Technologies, 210518). MCA205 cells were transfected with a mixture of plasmids: 0.4 μg transposase (Addgene) and 1.6 μg pSBbi-GFP-GH using Lipofectamine 2000 (Invitrogen, 11668019). For positive clone selection, GFP[+] cells were selected using fluorescence-activated cell sorting. RNA was extracted using RNeasy Mini Kit (Qiagen, 74106) and converted to cDNA using Random Hexamers (Invitrogen, N8080127) and SuperScript II Reverse Transcriptase (Invitrogen, 18064014). Quantitative reverse transcription polymerase chain reaction (RT-qPCR) was performed to validate spectrin-β2 expression using the SYBR Green PCR Master Mix (Thermo Fisher Scientific, 4309155) on a QuantStudio 7 Flex Real-Time PCR System (Thermo Fisher Scientific). Relative expression values were calculated from $2^{-\Delta\Delta Ct}$ method using *Actb* as reference gene and normalized to the spectrin-β2 expression levels of the parental line.

Primers for cloning and RT-qPCR – *Sptbn1* were as follows: forward: 5′-CCCCAAGCTTGGCCTTCACTTCTTCTTGCCGAAAAGG-3′; *Sptbn1* reverse: 5′-ACCCCAAGCTGGCCTATGACGACCACGGTAGCC-3′.

Primers for RT-qPCR – *Actb* were as follows: forward: 5′-CTAAG GCCAACCGTGAAAAG-3′; *Actb* reverse: 5′-ACCAGAGGCATACA GGGACA-3′. Primers for site-directed mutagenesis – *Sptbn1* mutant were as follows: forward: 5′-TGAACCAGATTGCACTGCAGCTG ATGCACAA-3′; *Sptbn1* mutant reverse: 5′-TTGTGCATCAGCTGCAGTGCA ATCTGGTTCA-3′.

All tumor cell lines, unless otherwise indicated, were cultured in R10 medium: RPMI 1640 medium (Gibco, 31870074) supplemented with 10% fetal bovine serum (FBS) (Sigma, F7524, Lot No. BCCB0090) (heat-inactivated), 50 μM 2-mercaptoethanol (Gibco, 31350-010), and 100 U ml[−1] penicillin, 100 μg ml[−1] streptomycin and 292 ng ml[−1] L-glutamine (Gibco, 10378-016), in humidified 37 °C 5% $CO_2$ incubator. Cells were washed with Dulbecco's phosphate-buffered saline (PBS) (Gibco, 14190169) before being dissociated with trypsin (0.25%) (Gibco, 15090-046) during passaging. For cryopreservation, cells were resuspended in full medium with 10% dimethyl sulfoxide (DMSO) (Sigma, D2650).

HeLa cells stably expressing OVA fusion constructs were generated by transfecting 50% confluent cells with a mixture of plasmids including 5 μg pSBbi-RFP (Addgene) coding for the gene of interest (see below) and 0.5 μg transposase (Addgene) using Lipofectamine 2000 (Invitrogen). Cells were grown in R10[+] (R10 plus 1 mM pyruvate, 10 mM HEPES and 0.1 mM non-essential amino acids). For positive selection of transfected HeLa cells, R10[+] was supplemented 2 days after transfection with puromycin (1 μg ml[−1]) for several passages until the population showed a unimodal distribution of RFP expression. The Mutu DC1940 line (Mutu DCs) was a gift from H. Acha-Orbea and was cultured in IMDM medium containing 10% FCS, 50 μM 2-mercaptoethanol, 100 U ml[−1] penicillin, 100 μg ml[−1] streptomycin. All media and media supplements were from Life Technologies except for FCS (Source Bioscience). Flt3L bone marrow-derived DCs were generated by culturing bone marrow cells in R10[+] containing 150 ng ml[−1] Flt3L for 9 days. XCR1[+] cDC1 in those cultures were enriched by positive selection using anti-XCR1 antibody and MACS beads (Miltenyi). The purity of the resulting cDC1 preparation was usually >90%.

### Plasmid and gBlocks

A His-tagged nonsecreted version of the model antigen OVA was used to generate the OVA fusion constructs detailed in Fig. 5 and Extended Data Fig. 7. For cloning of OVA fusion constructs, gBlocks (Integrated DNA Technologies) were synthesized with the sequence of the gene of interest and appropriate overhang sequences (Supplementary Table 1) for in-fusion cloning into the sleeping beauty plasmid pSBbi-RP (Addgene). The plasmid was digested with SfiI and the gBlocks were ligated (50 °C, 15 min) at a 1:1 weight ratio into the SfiI-linearized plasmid using the In-fusion Snap Assembly Master Mix (TaKaRa Bio).

### Reagents

Rat-anti-DNGR-1 antibody (1F6)[10] and isotype control antibody (MAC49) were provided by the Cell Services STP (Crick). PE-conjugated rat-anti-DNGR-1 antibody (1F6) was from eBioscience. Rabbit-anti-OVA antibody was from Sigma. HRP-conjugated rabbit β-actin antibody was from Cell Signaling Technology. HRP-conjugated goat-anti-rabbit and HRP-conjugated mouse-anti-His-Tag antibodies were from Southern Biotech. R-PE-conjugated H-2Kb/SIINFEKL tetramer (OVA pentamer) was from Proimmune. AF647-conjugated mouse-anti-MHC-I antibody (28-8-6) was from BioLegend and AF647-conjugated isotype control antibody (eBM2a) was from eBioscience. APC-conjugated mouse-anti-MHC-I (H-2D[b] specific) antibody (KH95) was from BioLegend. Unconjugated mouse-anti-MHC-I (H-2K[b] specific) antibody (Y-3) and AlexaFluor 647-conjugated goat-anti-mouse IgG were from Merck and Thermo Fisher Scientific, respectively. Fluorophore-labeled monoclonal antibodies against CD8, CD44 and CD19 were from BD Biosciences. Poly I:C was from Invivogen.

## In vitro XP

cDC1-mediated XP of HeLa cells expressing OVA fusion proteins was carried out as described recently[54]. Briefly, cells were UV-irradiated (240 mJ cm$^{-2}$) and left overnight in serum-free RPMI1640 medium to undergo secondary necrosis. Dead cells were added to Mutu DCs or Flt3L-derived XCR1$^+$ cDC1 ($1 \times 10^5$ per well) at the indicated ratio and cultured in 96-well round-bottom plates at 37 °C in R10$^+$ medium. To facilitate dead cell contact with DCs, plates were centrifuged at $200 \times g$ for 3 min at the start of the incubation. Pre-activated OT-I T cells ($2 \times 10^5$ per well) that had been prepared 5 days before the XP assay[41] were added after 4 h and OT-I T cell activation was determined by measuring IFN-γ levels in the supernatant of overnight cultures by ELISA.

## In vivo immunization with necrotic cells

Mice were injected intravenously with $1 \times 10^6$ UV-irradiated HeLa cells expressing LA- or mutLA-OVA mixed with poly I:C (25 μg per mouse) prior to injection. Six to eight days later, red blood cell lysed splenocyte suspensions were prepared from spleens of injected mice, stained with H-2K$^b$/SIINFEKL pentamer reagent, followed by anti-CD8α, anti-CD44 and anti-CD19 antibodies. Pentamer-positive (OVA-specific) CD8$^+$ T cells were analyzed by flow cytometry. For analysis of CD8$^+$ T cell function, splenocytes were re-stimulated with SIINFEKL peptide (1 μM) overnight, stained with fixable live/dead dye, anti-CD8α and anti-CD44 antibodies, and fixed and stained with anti-IFN-γ antibody using a Fix/Perm kit (Nordic MUbio).

Stained cells were acquired on an LSRFortessa (BD Biosciences) and data were analyzed using FlowJo software (Treestar). Statistical analysis of significance was performed using one-way ANOVA.

## Western blot

A total of $1 \times 10^6$ transfected and untransfected HeLa cells were lysed in buffer containing 1% NP-40 and protease inhibitors (Boehringer). Samples (10% v/v) were run in triplicate arrays of three (untransfected, LA-OVA and mutLA-OVA) on a 4% to 20% gradient gel (Bio-Rad) and transferred to PVDF membranes using a Trans-Blot Turbo system (Bio-Rad). Membranes were soaked with 5% milk, cut into three strips each containing the three samples and each strip was probed with a different primary antibody (anti-β-actin, His-Tag or OVA) as indicated. Bound HRP-conjugated primary (anti-β-actin, His-Tag) or secondary antibodies (OVA) were detected using a chemoluminescent HRP substrate (SuperSignal West Pico PLUS, Thermo Fisher Scientific) and images were acquired with an ImageQuantTM 800 imaging system (Amersham).

## Confocal microscopy

HeLa cells were cultured on poly-L-lysine-coated coverslips overnight at 37 °C, fixed in 4% paraformaldehyde/PBS for 15 min and washed and permeabilized in 0.1% Triton-X100/PBS for 4 min. After washing with PBS, cells were left in blocking buffer (1% BSA + 2% FCS in PBS) overnight and stained with a polyclonal rabbit OVA antibody (Merck) followed by fluorescent labeled anti-rabbit antibody together with fluorescent labeled phalloidin (Cytoskeleton) to reveal actin filaments. Coverslips were mounted using Fluoromount-G (Southern Biotech) and images were collected using a laser scanning confocal microscope (LSM 880, Zeiss).

Analysis of intensity profiles for OVA and phalloidin fluorescence in micrographs of LA-OVA and mutLA-OVA cells was performed in ImageJ. The resulting curves were smoothed (second-order smoothing with 10 neighbors) using Prism and plotted as overlay of pixel intensity curves across distance.

## Transplantable tumors

All cell culture procedures were performed in sterile condition in a laminar flow hood. Cultured tumor cells were washed with PBS, dissociated with trypsin (0.25%), and, where necessary, filtered through a 70-μm

cell strainer, before being washed in PBS three times. The final cell pellet was resuspended in PBS for the required concentration of cells accordingly ($0.5 \times 10^6$ cells per 100 μl) and injected subcutaneously in the shaved right flank of each mouse. Mice were monitored for tumor growth at least 2 to 3 times per week. The values of the longest (length, $l$) and perpendicular shortest (width, $w$) tumor diameters were measured using digital Vernier calipers, and tumor volume calculated using the formula: $0.5 \times (l \times w^2)$[55], expressed in the standard metric units mm$^3$. Mice bearing tumors were monitored until the pre-determined humane endpoint (for example mean tumor diameter reaching ≥15 mm, weight loss ≥15% or tumor ulceration) or for at least 30 days following tumor transplantation; none of which were exceeded in this work.

To assess the regressor phenotype of individual sarcoma cell lines from DNGR-1$^{KO}$ mice, cells were either transplanted into WT or immunodeficient (RAG$^{KO}$) mice. To further validate CD8 T cell dependence of regressor tumor rejection, cells were transplanted into WT mice treated either with an anti-CD8 depletion (clone 2.43, BioXCell) or an isotype matched control antibody (BioXCell). Antibody treatment (200 μg per mouse) started at day −2 before tumor cell inoculation and was continued at an interval of 3 days for a total of five injections per mouse.

## Carcinogenesis with MCA

Chemical-induced carcinogenesis experiments were instigated using a single dose of either 100 μg or 400 μg of MCA (Sigma-Aldrich, 213942) in 100 μl corn oil (Sigma-Aldrich, C8267), administered subcutaneously to the shaved right flank of each mouse. Mice were between 8 and 12 weeks old at the start of each experiment and both male and female mice were included. Tumors were clinically diagnosed when they reached a volume of 250 mm$^3$ (~8 mm in diameter) and demonstrated progressive growth over time, following a similar method of threshold calibration as previously described[24]. Mice were monitored at least once or twice weekly for up to 250 days.

## Generation of primary sarcoma cell lines

Subcutaneous flank tumors were harvested aseptically and transferred in RPMI1640 medium on ice. Respective tumor fragments were minced before digestion with Collagenase IV (Worthington, LS004188) (stock 10,000 U ml$^{-1}$, 1:50 dilution used) in a 37 °C shaker-incubator for 1 h; protocol adapted from Smyth et al.[30]. The product of the enzymatic digestion was passed through a 100-μm filter. The flow-through was then washed and cell pellet resuspended in fully reconstituted media as per normal cell culture conditions described above. These primary cell lines were cryopreserved at a low passage (usually passage 3), and simultaneously sent to the Cell Services STP at the Francis Crick Institute for quality check and mycoplasma screening, in preparation for later inoculation into secondary host mice.

## Quality control by short tandem repeat profiling

The primary fibrosarcoma cell lines generated were further characterized using the multiplex PCR assay designed to profile mouse cell lines using primers targeting 18 mouse loci with highly polymorphic short tandem repeats, as previously described and validated by the Consortium for Mouse Cell Line Authentication[56]. This short tandem repeat profiling was performed as part of the quality control process in curating a reference library for all the banked original primary cell lines, by the Cell Services STP at the Francis Crick Institute.

## Azoxymethane and DSS colon carcinoma model

The colitis-associated carcinoma model was established based on the protocol adapted from Mager et al.[57], which involved 2 doses of AOM (Sigma-Aldrich, A5486) 10 mg kg$^{-1}$ administered intraperitoneally on day 0 and day 19, and 3 cycles of 1.0 to 2.0% dextran sodium sulfate (DSS) (Thermo Fisher Scientific, J.14489.22) in normal drinking water (cycle 1, day 7 to 12; cycle 2, day 19 to 24; cycle 3, day 29 to 36). Both AOM and DSS were filtered through a 0.45 μm cellulose acetate filter

prior to in vivo administration, respectively. Mice were between 8 and 12 weeks old at the start of the experiment and both male and female mice were included. Mice were followed up for 111 days before euthanasia and harvest of whole colon. The luminal tumor-bearing surface of the colon was photographed and the tumor areas were quantitated using Fiji ImageJ, sum of which equate to the tumor burden per colon.

## Quantification of MHC class I expression of primary cell lines

Regressor sarcoma cell lines derived from primary MCA-induced tumors grown in DNGR-1$^{KO}$ mice were detached from the tissue culture dish using a cell scraper and subsequently stained with an AF647 conjugated mAb (clone 28-8-6) that recognizes both H-2K$^b$ and H-2D$^b$, or an isotype control mAb (clone eBM2a, mouse IgG2a) conjugated to the same fluorochrome. Stained cells were acquired on an LSRFortessa (BD Biosciences) and data were analyzed using FlowJo software (Treestar).

## Whole-exome sequencing

Cell pellets containing 2 to $5 \times 10^6$ cells were snap-frozen in liquid nitrogen and cryopreserved at the time when cells were inoculated into secondary host mice. DNA extraction of cell pellets for whole-exome sequencing (WES) was processed as a batch, as per manufacturer's guidelines, using the in-column DNeasy Blood and Tissue Kit (Qiagen, 69504). For the extraction of DNA from the tail snips of mice, the distal 5 mm of the tail snip was cut into small pieces and placed into the Precellys Lysing Kit (containing stainless steel beads, for hard tissue grinding) (Bertin, P000917-LYSK0-A) before the addition of 180 µl lysis buffer (Buffer ATL). After incubation for 1 h, this was placed into the Precellys 24 Tissue Homogenizer System (Bertin, P002391-P24T0-A.0) to dissociate and homogenize the tail fragments, prior to the subsequent steps as described above with the processing of cell pellets. All the respective final eluate containing DNA was collected in 100 µl nuclease-free water (Invitrogen, 4387936).

DNA samples were normalized to 187 ng, sheared using the Adaptive Focused Acoustics technology on the Covaris platform, and the libraries were prepared using NEBNext Ultra II DNA Library Prep (Illumina) protocol. Next, the samples were enriched using the Twist Mouse Exome panel (https://www.twistbioscience.com/products/ngs/fixed-panels/mouse-exome-panel) (Twist Bioscience). Libraries were then sequenced on the NovaSeq 6000 System (Illumina) at 100 bp paired end.

## Annotation of raw sequencing data

For WES samples, sequencing yield was ~40 million strand specific paired end 101 bp reads per sample for the WES data. The nf-core SAREK pipeline[58] (https://github.com/nf-core/sarek) (version 2.7.1), running on Nextflow[59] version 21.10.6, was used to call mutation from FASTQ files using default parameters. Briefly, FASTQ file per sample were trimmed for low quality using TrimGalore (version 0.6.4) (https://github.com/FelixKrueger/TrimGalore); reads were aligned to the Ensembl mouse GRCm38 genome using BWA-mem2 (ref. 60); duplicates were marked using GATK MarkDuplicates[61] (GATK version 4.1.7.0); base quality scores were recalibrated using GATK BaseRecalibrator and ApplyBQSR; Germline variants were called using GATK HaplotypeCaller; Somatic mutations variations between tumor and normal sample pairs were called with Strelka2 (ref.62) (version 2.9.10). Variants were annotated with snpEff[63] (version 4.3).

## Genomic variant processing

Multiple datasets were loaded into the R environment, including sample metadata, transcriptomic and genomic data. Transcriptomic data, including gene, transcript and protein IDs, were consolidated from multiple sources and duplicate entries were removed. The analysis was conducted using R libraries used including dplyr, tidyverse, readr, ggplot2, ggpubr and survminer. Expression data from the six different representative primary tumor cell lines was obtained for all genes, and genes that had fragments per million reads above zero in all six lines were labeled as commonly expressed genes.

Single-nucleotide variant and indel data were extracted from VCF files from all primary tumor cell lines and annotated using RefSeq. Strelka2 calls were then aggregated and summarized to calculate variant allele frequencies (VAF), with subsequent filtering based on the PASS flag. Mutations were classified as synonymous, frameshift_variant, stop_gained, stop_lost, start_lost, and missense. Clonality of mutations was assessed using a mixture modeling approach (mclust package with parameter k = 2) to classify mutations into clonal or subclonal categories based on their VAFs. Statistical tests, including Wilcoxon tests, were applied to compare differences across groups, with results incorporated directly into plots for immediate interpretation.

## Copy-number variation analysis

The copy-number variation analysis was conducted using CNVkit and R. Segmentation was performed with CNVkit, using the default cbs method from DNAcopy. Copy number gains and losses were called when their $\log_2$ copy number ratios were >0.6 and <−0.6, respectively. The proportion of gained and lost genome was calculated for each sample over the total of genomic length called. Minimum consistency segmentation was performed across all samples in the cohort to allow direct comparison of their copy number state at each genomic location. Copy number call frequencies were aggregated by sample group (host genotype and rechallenge phenotype) and plotted with ggplot2 in R.

## Neoantigen prediction of actin-binding protein mutations

Strelka2 VCF files containing mutations for each cell line were annotated with GT, AD and VAF fields. Mutations were then filtered requiring a PASS flag and an AD > 4 (at least five reads supporting the alternative allele). Peptide sequences corresponding to each mutation were annotated with the VEP Wildtype and Frameshift plugins (cache version 89, corresponding to Ensembl mouse GRCm38). Binding affinity was predicted with NetMHCpan using the pVACtools[34] pipeline. In short, the latest docker image of pVACtools was pulled from dockerhub griffithlab/pvactools:4.0.0 using Singularity 3.11.3. For each VCF, pVACseq was run with the options H-2K$^b$ and H-2D$^b$ NetMHCpan -e1 8,9,10 -b 500 −tdna-vaf 0.2 (ref. 64). The resulting filtered tsv files were merged and analyzed with R.

To identify a reasonable threshold in this dataset, we compared the distributions of binding affinities in regressors and progressors. There is currently no consensus on an optimal threshold and a balance of sensitivity versus specificity is necessary. Here, high affinity neoantigens were defined as those with a binding affinity $IC_{50} < 150$ nM, as suggested in other studies[65].

## Synthesis of predicted neoantigen peptides and binding to MHC-I

Peptides listed in Extended Data Table 3, as well as the two spectrin-β2 peptides[25], were synthesized by GL Biochem using solid-phase peptide synthesis. The peptides were analyzed by LC−MS on an Agilent 1260 LC-MSD (Agilent Poroshell 120 EC-C18 2.7 µm, 3.0 × 30 mm column), using a linear gradient of 5−100% B over 8.5 min at a flow rate of 0.425 ml min$^{-1}$. A binary solvent system [A: and H$_2$O/0.08% TFA/1% MeCN and B: MeCN/0.08% TFA] was used.

Peptide binding to MHC-I was assessed using the RMA-S stabilization assay[35]. Briefly, RMA-S cells were added to a 24-well plate ($5 \times 10^5$ cells per well) and grown overnight at 26° C. The next day, cells were pulsed with 10 µM peptide for 1 h before the plate was moved back to 37° C for at least 2 h. Peptide pulsed RMA-S cells were harvested in PBS/EDTA, washed and stained either with a mAb specific for H-2K$^b$ (clone Y-3) or H-2D$^b$ (clone KH95). Stained cells were acquired on an LSRFortessa (BD Biosciences) and data were analyzed using FlowJo software (Treestar).

## In vivo peptide immunization

Bone marrow cells from C57BL/6 mice were cultured in six-well plates in the presence of Flt3L (150 ng ml$^{-1}$). On day 8 (1 day before harvesting) Flt3L BM-DC were pulsed with 20 μM neoantigen peptides (three wells per peptide) or the mutated (mut) spectrin-β2 peptide together with polyIC (10 μg ml$^{-1}$). On day 9, peptide/polyIC-pulsed bone marrow DCs were harvested, washed two times in PBS and injected systemically (intravenously or intraperitoneally) into groups of three to five C57BL/6 mice ($3–5 \times 10^6$ cells per mouse). Peptide-immunized mice were sacrificed on day 6 or 7, and red blood cell lysed splenocytes were re-stimulated overnight with the respective peptide (10 μM) in a 96-well round-bottom plate. CD8 T cell activation was assessed by measuring IFN-γ in the culture supernatant by ELISA.

## Statistics

Statistical analyses were performed using R (as described above) or GraphPad Prism 9 for Mac OS (version 9.3.0, updated 2021). Data are shown as mean ± s.d. or ± s.e.m., unless otherwise stated. Data distribution was assumed to be normal, but this was not formally tested. In general, two-way ANOVA adjusted for multiple comparisons with Bonferroni correction was used to compare tumor growth profiles. Otherwise, specific tests performed within each experiment are indicated in the respective figure legends. In all instances, $P$ values are two tailed, and $P \leq 0.05$ is considered the threshold for statistical significance.

For transplantable tumor models, no statistical methods were used to pre-determine sample sizes, but our sample sizes are similar to those reported in our previous publications[18,19,22]. In all in vivo experiments, mice were randomly allocated to experimental groups, controlled for age and sex distribution. Data collection and analysis were not performed blind to the conditions of the experiments. No animals or data points were excluded from the final analyses, unless specifically reasoned in the respective figure legends.

For the MCA carcinogenesis model, we hypothesized that tumors developing in DNGR-1$^{KO}$ mice were enriched in mutations in the $n = 88$ genes encoding murine FABPs. Therefore, a power calculation for estimating the sample size required was based on the following assumptions. There are in total 88 genes of interest encoding for FABPs. The Twist exome kit used for whole-exome sequencing targets a total of ~38 Mb, with the 88 FABP genes comprising 334,760 of that, or 0.88% of the total region. Assuming that MCA fibrosarcomas have a high mutational burden of between 3,500 and 5,000 mutations based on previous literature, and the reference being aligned to is ~2.7 Gb, a power calculation is generated using binomial probabilities: pbinom[0, N, prob = (burden*334760/2.7e9), lower=FALSE]. Therefore, it is estimated that $n = 4–6$ mice are required for each experimental arm, for in excess of 95% probability of at least one mouse having a variant within the FABP exon.

Subsequently, for experiments testing the tumor growth profile of individual primary cancer cell lines, at least $n = 3–5$ mice were used per cohort, in line with previously published work, including by Schreiber and colleagues[24]. Additionally, to increase the confidence of tumor-calling, the tumor growth profile of the primary cell lines in secondary WT hosts was further analyzed using an exponential fitted (Malthusian) modeling and representation of tumor growth profiles as heatmaps. Likewise, for experiments testing the immunogenicity of neoantigen peptides in vivo or cross-priming of CD8 T cells in response to dead cell-associated antigen, at least $n = 3–5$ mice were used per cohort.

For the bioinformatic analysis of human cancer datasets, the association between *CLEC9A* and survival outcomes in patients with cancer were analyzed using data downloaded from the cBioPortal for Cancer Genomics platform (http://cbioportal.org/), with source data from The Cancer Genome Atlas Pan-cancer Atlas[66,67]. Kaplan-Meier plots were generated of PFS and OS in patients with cancer, and groups were compared by log-rank (Mantel-Cox) test.

## Reporting summary

Further information on research design is available in the Nature Portfolio Reporting Summary linked to this article.

## Data availability

Raw and processed whole exome sequencing data have been deposited in the European Nucleotide Archive under the accession code PRJEB100660. Source data are included in this article. A list of mouse FABPs ($n = 88$) (Extended Data Table 1) and MBPs ($n = 98$) (Extended Data Table 2) was compiled from human orthologs previously reported by Giampazolias et al.[18]. Source data are provided with this paper.

## Code availability

The code used to analyze the data in this study is available in GitHub: https://github.com/FrancisCrickInstitute/DNGR1_XP.

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

## Acknowledgements

We thank the Biological Research Facility, Chemical Biology STP and Cell Services STP at The Francis Crick Institute for their support throughout this project. We thank E. Nye, A. Suárez-Bonnet and

S. Priestnall from the Experimental Histopathology STP at The Francis Crick Institute and The Royal Veterinary College for their help in the initial optimization and validation of the carcinogenesis models. We are grateful to all members of the Immunobiology Laboratory for helpful discussions and suggestions. This work was supported by Adendra Therapeutics and The Francis Crick Institute, which receives core funding from Cancer Research UK (CC2090), the UK Medical Research Council (CC2090) and the Wellcome Trust (CC2090); an ERC Advanced Investigator grant (AdG 268670); a Wellcome Investigator Award (106973/Z/15/Z); and a prize from the Louis-Jeantet Foundation. K.H.J.L. was supported by a Wellcome Imperial 4i Clinical Research Fellowship (216327/Z/19/Z) and Crick Postdoctoral Clinical Fellowship. W.S. is funded by a Boehringer Ingelheim Fonds fellowship. S.T. is funded by CRUK (A29911), The Francis Crick Institute, which receives its core funding from CRUK (CC3044), the UK Medical Research Council (CC3044), the Wellcome Trust (CC3044), the National Institute for Health Research Biomedical Research Centre at the Royal Marsden Hospital and Institute of Cancer Research (A109), the Royal Marsden Cancer Charity, The Rosetrees Trust (A2204), Ventana Medical Systems (10467 and 10530), the National Institutes of Health (U01 CA247439) and the Melanoma Research Alliance (686061).

## Author contributions

K.H.J.L. and O.S. conducted all the experiments and analyzed data with assistance from T.C-D., E.G., B.F., C.A.C., M.D.B., W.S., A.C., S.L., B.V. and S.M. N.C.R. managed all mouse colonies. The whole-exome and RNA sequencing data were analyzed by I.L., L.Z. and P.C. G.K. provided support in biostatistics and mathematical modeling. K.H.J.L., O.S. and C.R.S. designed the study, interpreted the data and wrote the paper. R.M., J.S., S.T. and C.R.S. supervised the project. All authors reviewed and edited the paper.

## Funding

## Competing interests

B.V. and S.M. are employees of Apple Tree Partners. R.M. is an employee of and owns stock options in Adendra Therapeutics, which partly funded this work. S.T. has received speaking fees from Roche, Astra Zeneca, Novartis and Ipsen. C.R.S. owns stock options and/or is a paid consultant for Adendra Therapeutics, Montis Biosciences and Bicycle Therapeutics. C.R.S. is a Visiting Professor at Imperial College London and King's College London and holds an honorary professorship at University College London. The other authors declare no competing interests.

## Additional information

**Extended data** is available for this paper at https://doi.org/10.1038/s41590-025-02354-w.

**Correspondence and requests for materials** should be addressed to Caetano Reis e Sousa.

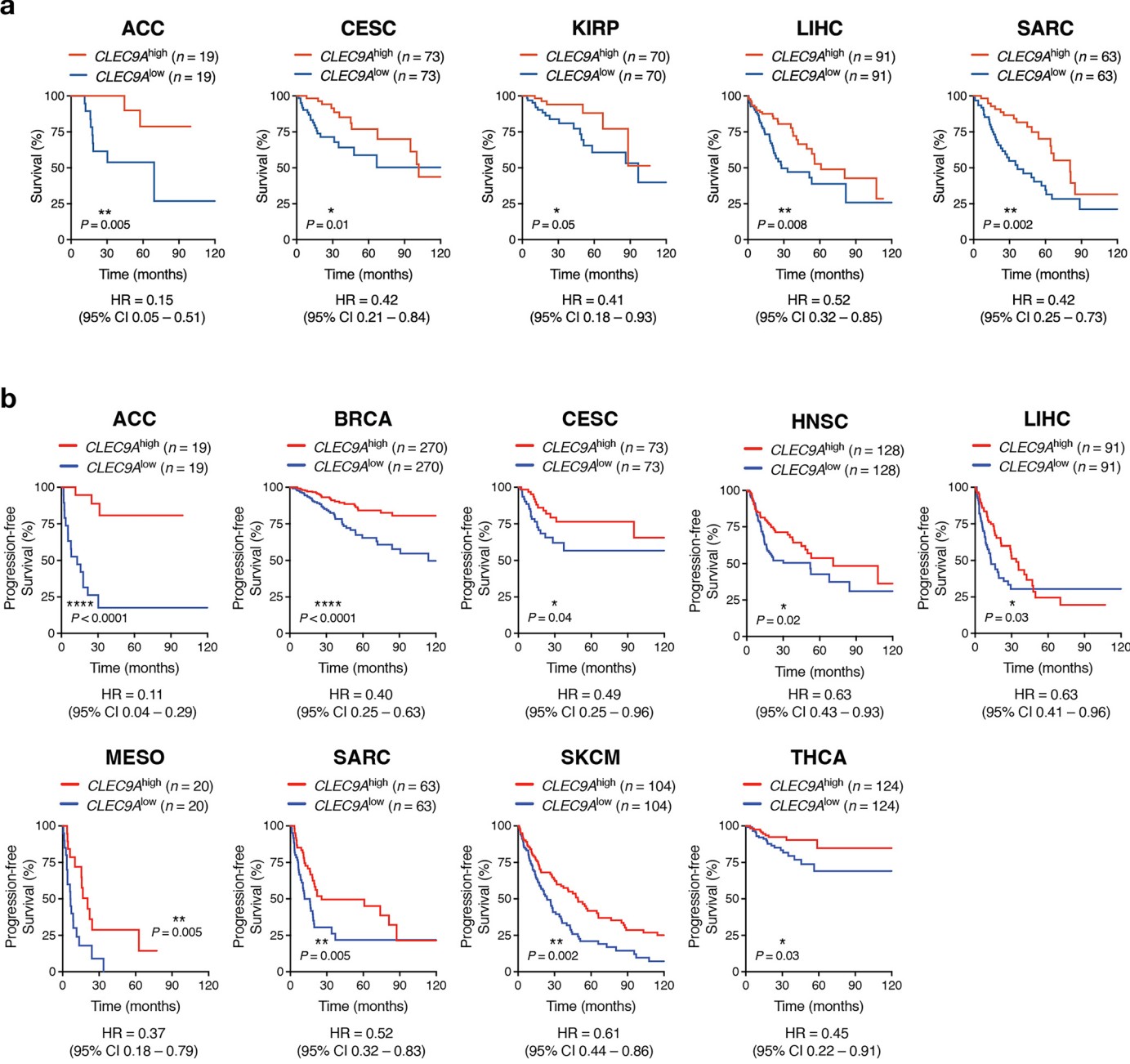

**Extended Data Fig. 1 | High expression of *CLEC9A* is associated with better survival outcomes in cancer. a-b,** Kaplan-Meier plots of (**a**) overall survival (OS) and (**b**) progression-free survival (PFS) in patients with cancer stratified based on the highest expression (top 25%, red line) versus the lowest expression (bottom 25%, blue line) of *CLEC9A*. A selection of cancer subtypes (out of *n* = 31 TCGA Pan-cancer Atlas studies in solid malignancies) in which there was a positive association between high *CLEC9A* expression and OS and PFS, respectively, are shown here. Data were downloaded from the cBioPortal platform (http://cbioportal.org/), and Log-rank (Mantel-Cox) test was applied for comparison between groups. For each plot, data are summarized as hazard ratio (HR) with 95% confidence interval (CI). Where indicated, *$P \le 0.05$ and **$P < 0.01$ (two-sided). Abbreviations: ACC, adrenocortical carcinoma; CESC, cervical squamous cell carcinoma and endocervical adenocarcinoma; KIRP, kidney renal papillary cell carcinoma; LIHC, liver hepatocellular carcinoma; and SARC, sarcoma. Data on OS in BRCA, breast invasive carcinoma; HNSC, head and neck squamous cell carcinoma; LUAD, lung adenocarcinoma; and SKCM, skin cutaneous melanoma; have been previously reported[32].

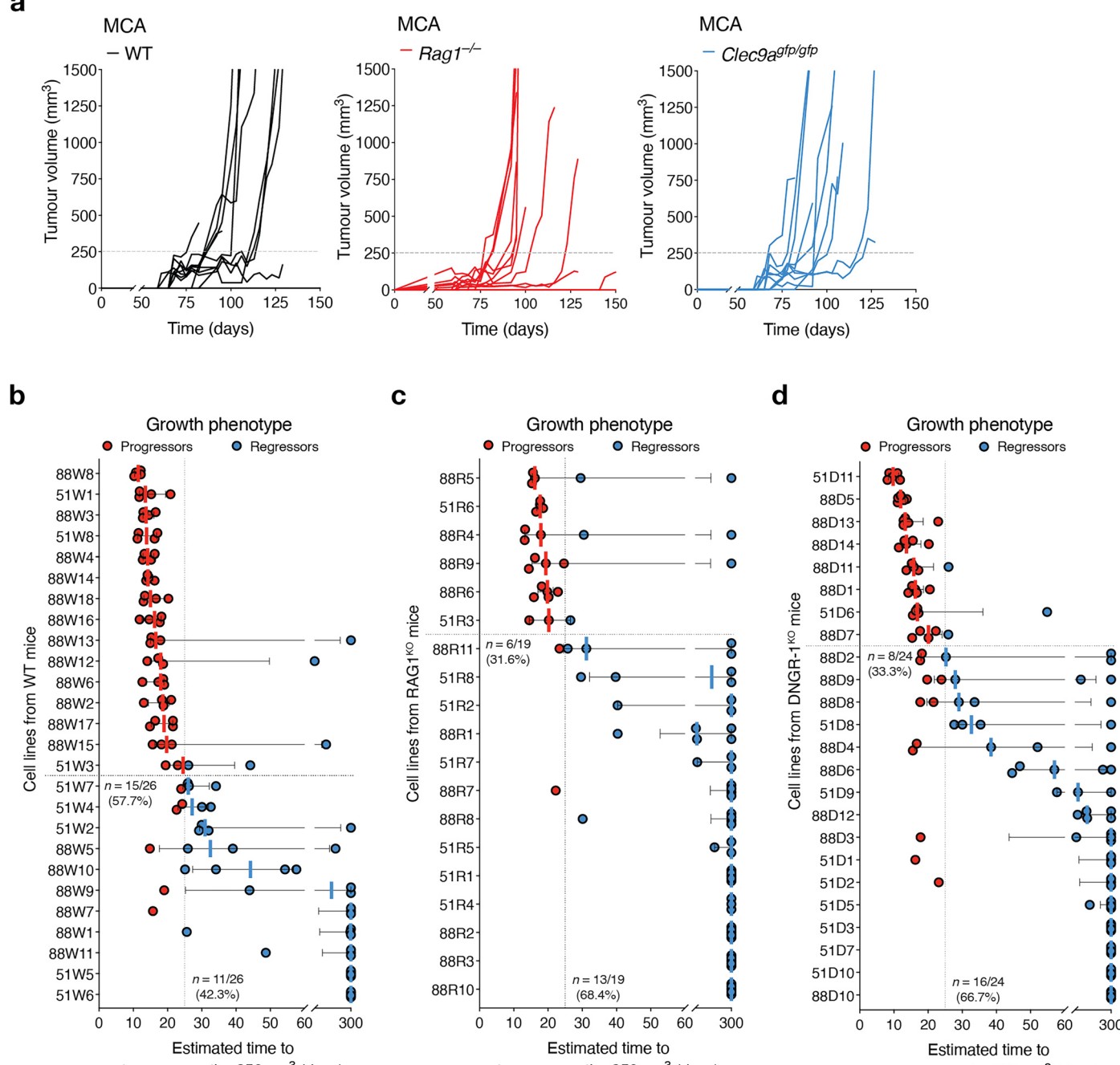

**Extended Data Fig. 2 | Growth phenotypes of primary tumors, and upon challenge in secondary naïve WT hosts. a**, Growth profile of individual tumors following s.c. inoculation with 3-methylcholanthrene (MCA) in WT ($n = 12$), RAG1[KO] ($n = 12$) and DNGR-1[KO] ($n = 11$) mice, during the surveillance period of 150 days. Data are plotted as tumor volume (mm³). Data are representative of four independent experiments; supporting data for Fig. 1a. **b-d**, Dot plots of the estimated time to establish tumors ≥250 mm³ for each primary tumor cell line, based on the respective fitted exponential (Malthusian) modeling for each individual growth profile, from (**b**) WT ($n = 26$), (**c**) RAG1[KO] ($n = 19$) and (**d**) DNGR-1[KO] ($n = 24$) hosts; supporting data for Fig. 2b–e, respectively. Data are presented as median and interquartile range. As indicated in red, progressors established tumors ≥250 mm³ within 25 days of challenge in naïve WT secondary hosts. Meanwhile, as indicated in blue, regressors are completely rejected or controlled (failed to establish tumors ≥250 mm³ within 25 days). For tumors which were completely rejected, time was assigned an arbitrary value of 300 days, indicating infinite time.

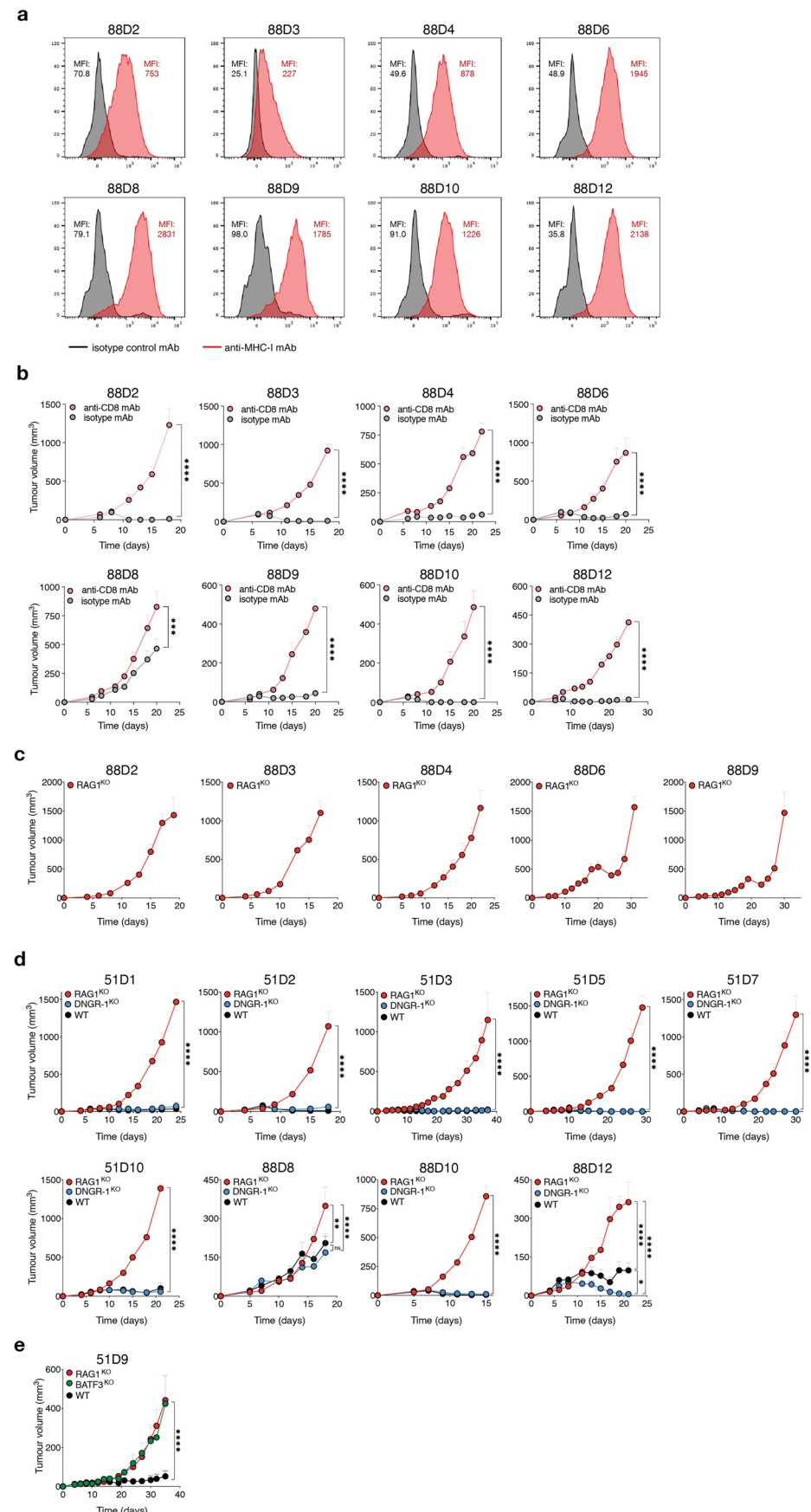

**Extended Data Fig. 3 | See next page for caption.**

**Extended Data Fig. 3 | Control of regressor tumor cell lines derived from DNGR-1^{KO} mice is dependent on T (and B) cells and cDC1s but not DNGR-1.**
**a**, Flow cytometry analysis of regressor tumor cell lines: 88D2, 88D3, 88D4, 88D6, 88D9, 88D10 and 88D12 cell lines. Overlay histograms represent staining for MHC-I (H-2K^b/H-2D^b). Numbers in the histogram overlay represent mean fluorescence intensity (MFI). **b**, Tumor growth profile of 88D2, 88D3, 88D4, 88D6, 88D9, 88D10 and 88D12 in WT mice ($n$ = 5 per cell line), following treatment with anti-CD8 or isotype monoclonal antibody (mAb) on Days -2, 1, 4, 8 and 11 of tumor cell inoculation. **c-e**, Tumor growth profile of all the regressor primary cell lines derived from MCA-treated DNGR^{KO} mice, following subcutaneous inoculation of cells in RAG1^{KO} (in red), and DNGR-1^{KO} (in blue) versus WT (in black) mice. **c**, Tumor growth profile of 88D2, 88D3 and 88D4 each in $n$ = 4 RAG1^{KO} mice respectively, and 88D6 and 88D9 each in $n$ = 3 RAG1^{KO} mice respectively. **d**, Tumor

growth profile of 51D1 in RAG1^{KO} ($n$ = 6), WT ($n$ = 7) and DNGR-1^{KO} ($n$ = 6) mice; 51D2 in RAG1^{KO} ($n$ = 4), WT ($n$ = 2) and DNGR-1^{KO} ($n$ = 3) mice; 51D3 in RAG1^{KO} ($n$ = 5), WT ($n$ = 11) and DNGR-1^{KO} ($n$ = 5) mice; 51D5 in RAG1^{KO} ($n$ = 5), WT ($n$ = 2) and DNGR-1^{KO} ($n$ = 3) mice; 51D7 in RAG1^{KO} ($n$ = 4), WT ($n$ = 2) and DNGR-1^{KO} ($n$ = 3) mice; 51D10 in RAG1^{KO} ($n$ = 4), WT ($n$ = 4) and DNGR-1^{KO} ($n$ = 4) mice; 88D8 in RAG1^{KO} ($n$ = 4), WT ($n$ = 5) and DNGR-1^{KO} ($n$ = 5) mice; 88D10 in RAG1^{KO} ($n$ = 4), WT ($n$ = 5) and DNGR-1^{KO} ($n$ = 5) mice; and 88D12 in RAG1^{KO} ($n$ = 4), WT ($n$ = 5) and DNGR-1^{KO} ($n$ = 5) mice.
**e**, Tumor growth profile of a representative regressor cell line derived in a DNGR-1^{KO} mouse, 51D9, implanted into BATF3^{KO} ($n$ = 5), RAG1^{KO} ($n$ = 5) mice, and WT ($n$ = 8) mice. Data are plotted as mean tumor volume (mm$^3$) ± s.e.m. Mean tumor volumes were compared using the Bonferroni-corrected two-way ANOVA. Where indicated, ****$P$ < 0.0001 and ns, not significant. In (**d**), where indicated, for 88D8: **$P$ = 0.002, and for 88D12: *$P$ = 0.0257. All data were from one experiment.

**a**

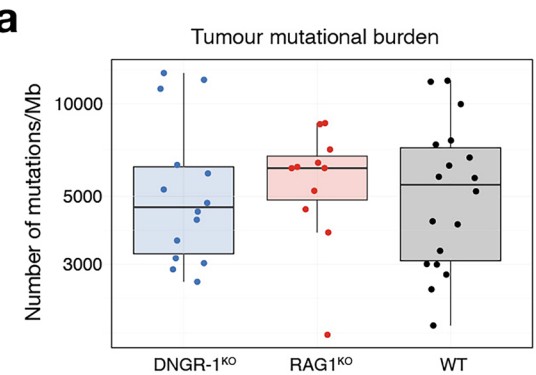

**b**

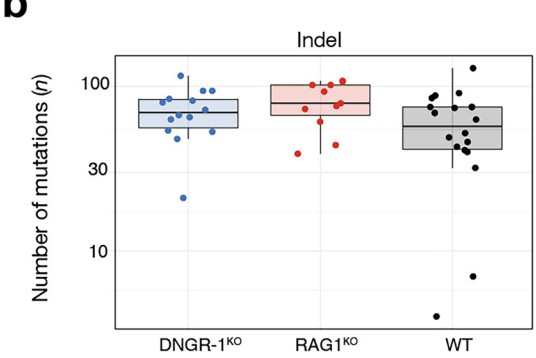
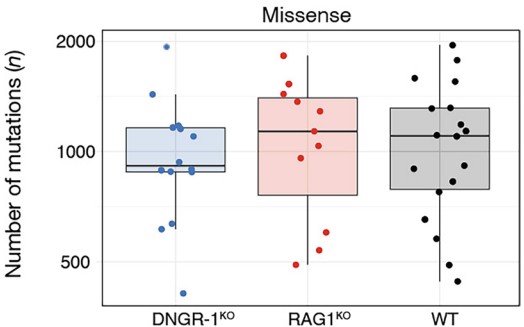

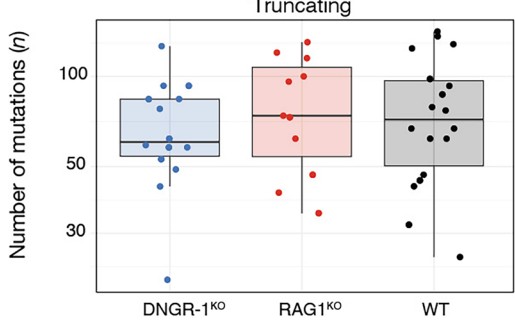
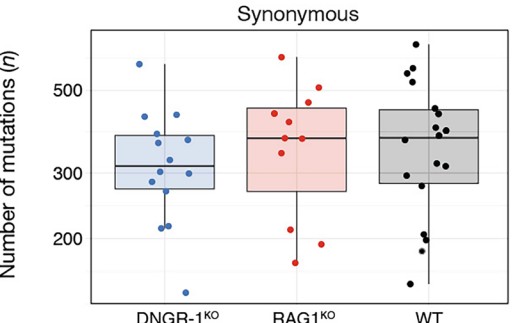

**Extended Data Fig. 4 | Genomic landscape of MCA-induced primary tumor cell lines. a**, Box and whisker plot of total mutational burden (mutations/Mb) across cell lines from different host background (*n* = 14 from DNGR-1^KO, *n* = 11 from RAG1^KO, *n* = 18 from WT). There were no significant differences between groups (Wilcoxon test). **b**, Box and whisker plots of clonal mutations across the different host background, including indels, missense, truncating and synonymous mutations. There were no significant differences except for a lower burden of indels in cell lines derived from WT compared to RAG1^KO hosts (two-sided Mann-Whitney U test, *P* = 0.05). In (**a-b**), box plots display the median (central line) and the first and third quartiles (lower and upper box edges). Whiskers extend to 1.5x the interquartile range from each quartile.

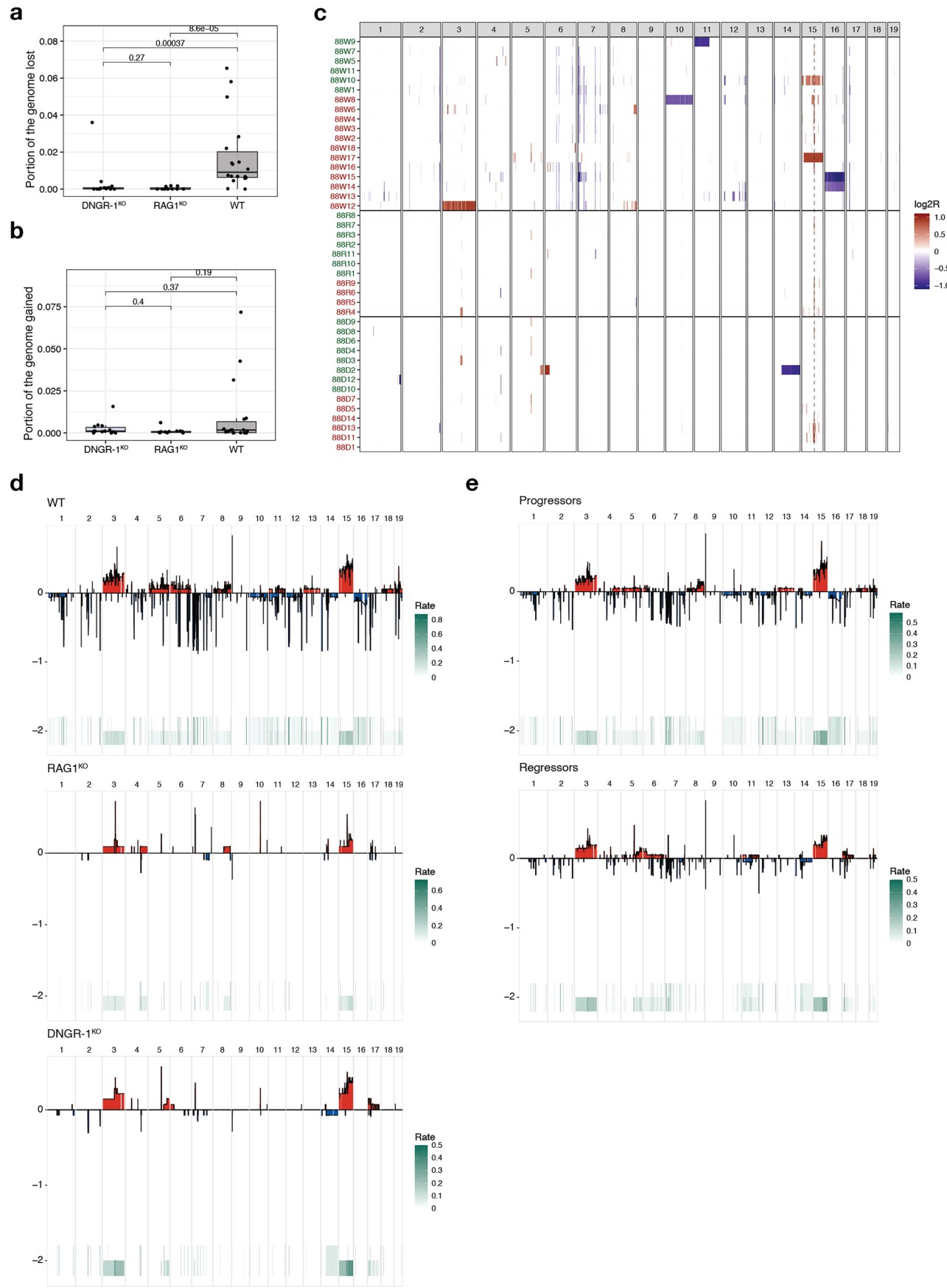

**Extended Data Fig. 5 | See next page for caption.**

**Extended Data Fig. 5 | Somatic copy number alterations (SCNAs) in tumor cell lines. a-b**, Box and whisker plots showing the burden of copy number (**a**) losses and (**b**) gains, between the different host background ($n = 14$ from DNGR-1$^{KO}$, $n = 11$ from RAG1$^{KO}$, $n = 18$ from WT), represented as portion of the genome. Box plots display the median (central line) and the first and third quartiles (lower and upper box edges). Whiskers extend to 1.5x the interquartile range from each quartile. Two-sided Mann-Whitney U tests were used to compare the distributions. **c**, SCNA calls along the genome for each cell line. Called segments are colored according to the segmented log2ratio, with blue for losses and red for gains. A vertical dashed line shows the location of the gene *Myc* in chromosome 15, which is recurrently gained. **d-e**, Frequency of SCNAs along the genome in each relevant group of cell lines, with gains in red and the height of the lines indicating the frequency of gain in the group of samples. Loss frequency is shown in the negative Y axis in blue. Cell line samples are grouped by (**d**) genotype of the host, and (**e**) growth phenotype upon secondary challenge in WT mice.

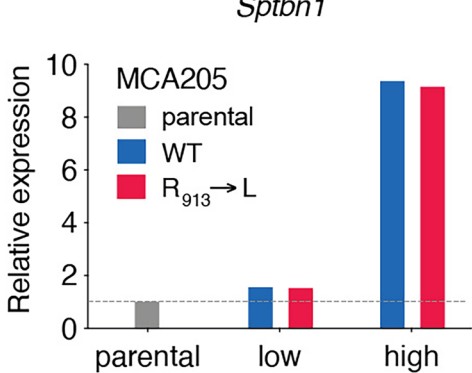

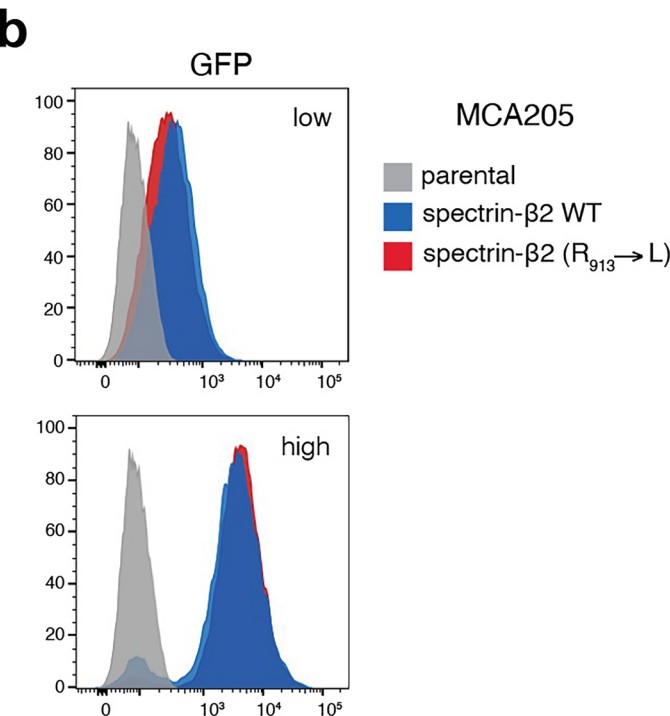

**Extended Data Fig. 6 | Generation of MCA205 spectrin-β2 mutant versus WT cell line. a**, Bar graph showing relative expression values of *Sptbn1* (encoding spectrin-β2), using *Actb* (encoding β-actin) as reference gene, and compared to parental MCA205, by quantitative reverse transcription polymerase chain reaction (RT-qPCR). **b**, Mean fluorescence intensity (MFI) of low (top) and high (bottom) expression of GFP in MCA205 parental versus MCA205 transfected with either WT or mutant spectrin-β2.

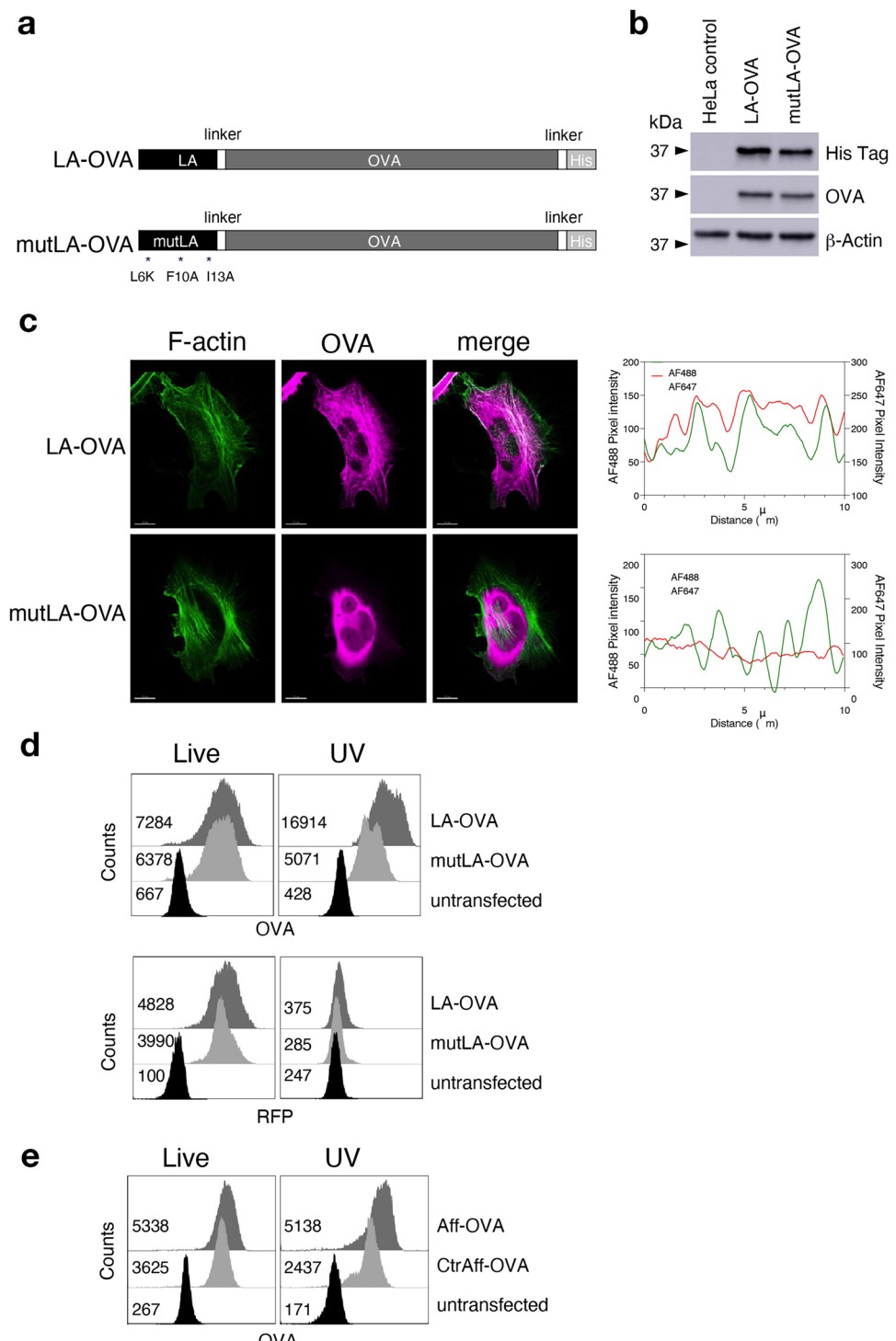

**Extended Data Fig. 7 | F-actin anchoring increases antigen retention in necrotic cells. a**, Schematic illustration of the two OVA fusion constructs, LA-OVA (top) and the control mutLA-OVA (bottom). **b**, Western blot analysis of lysates from transfected and untransfected HeLa cells probed with anti-His (top), anti-OVA (middle) and anti-β-actin (bottom) mAbs. **c**, Confocal analysis of HeLa cells expressing LA-OVA (left figure, top panels) or mutLA-OVA (left figure, bottom panels). Intensity profiles for OVA (AF647) and phalloidin (AF488) channels across 10 μm (white line indicated in the confocal images) are shown on the right. Scale bars shown in confocal images (left) correspond to a size of 5μm. **d-e**, Flow cytometry analysis of untransfected and transfected HeLa cells before (live) and after UV treatment (UV). Overlay histograms represent intracellular staining for OVA (**d**, top panels) and **e**). RFP fluorescence is shown for LA-OVA and mutLA-OVA cells (**d**, bottom panels). Numbers in the histogram overlay represent mean fluorescence intensity. Data are representative of 3 independent experiments.

## Extended Data Table 1 | List of all murine F-actin associated proteins (*n* = 88)

| No. | Gene symbol | Gene name | Chromosome location |
|-----|-------------|-----------|---------------------|
| 1 | Actr2 | ARP2 actin-related protein 2 | 11 |
| 2 | Actr3 | ARP3 actin-related protein 3 | 1 |
| 3 | Actn1 | Actinin, alpha 1 | 12 |
| 4 | Anln | Anillin, actin binding protein | 9 |
| 5 | Capza1 | Capping protein (actin filament) muscle Z-line, alpha 1 | 3 |
| 6 | Capza2 | Capping protein (actin filament) muscle Z-line, alpha 2 | 6 |
| 7 | Capzb | Capping protein (actin filament) muscle Z-line, beta | 4 |
| 8 | Carmil1 | Capping protein regulator and myosin 1 linker 1 | 13 |
| 9 | Cdc42 | Cell division cycle 42 | 4 |
| 10 | Cfl1 | Cofilin 1, non-muscle | 19 |
| 11 | Cfl2 | Cofilin 2, muscle | 12 |
| 12 | Ezr | Ezrin | 17 |
| 13 | Fscn1 | Fascin actin-bundling protein 1 | 5 |
| 14 | Flna | Filamin, alpha | X |
| 15 | Flnb | Filamin, beta | 14 |
| 16 | Pls1 | Plastin 1 | 9 |
| 17 | Fmn1 | Formin 1 | 2 |
| 18 | Fmn2 | Formin 2 | 1 |
| 19 | Gsn | Gelsolin | 2 |
| 20 | Ccdc88a | Coiled coil domain containing 88A | 11 |
| 21 | Capg | Capping protein (actin filament), gelsolin-like | 6 |
| 22 | Msn | Moesin | X |
| 23 | Myh1 | Myosin, heavy polypeptide 1, skeletal muscle, adult | 11 |
| 24 | Myh10 | Myosin, heavy polypeptide 10, non-muscle | 11 |
| 25 | Myh11 | Myosin, heavy polypeptide 11, smooth muscle | 16 |
| 26 | Myh13 | Myosin, heavy polypeptide 13, skeletal muscle | 11 |
| 27 | Myh14 | Myosin, heavy polypeptide 14 | 7 |
| 28 | Myh15 | Myosin, heavy chain 15 | 16 |
| 29 | Myh2 | Myosin, heavy polypeptide 2, skeletal muscle, adult | 11 |
| 30 | Myh3 | Myosin, heavy polypeptide 3, skeletal muscle, embryonic | 11 |
| 31 | Myh4 | Myosin, heavy polypeptide 4, skeletal muscle | 11 |
| 32 | Myh6 | Myosin, heavy polypeptide 6, cardiac muscle, alpha | 14 |
| 33 | Myh7 | Myosin, heavy polypeptide 7, cardiac muscle, beta | 14 |
| 34 | Myh8 | Myosin, heavy polypeptide 8, skeletal muscle, perinatal | 11 |
| 35 | Myh9 | Myosin, heavy polypeptide 8, non-muscle | 15 |
| 36 | Myo3a | Myosin IIIA | 2 |
| 37 | Myo3b | Myosin IIIB | 2 |
| 38 | Myo9a | Myosin IXA | 9 |
| 39 | Myo9b | Myosin IXB | 8 |
| 40 | Myl1 | Myosin, light polypeptide 1 | 1 |
| 41 | Myl2 | Myosin, light polypeptide 2, regulatory, cardiac, slow | 5 |
| 42 | Myl3 | Myosin, light polypeptide 3 | 9 |
| 43 | Myl4 | Myosin, light polypeptide 4 | 11 |
| 44 | Myl6 | Myosin, light polypeptide 6, alkali, smooth muscle and non-muscle | 10 |
| 45 | Myl6b | Myosin, light polypeptide 6B | 10 |
| 46 | Myl7 | Myosin, light polypeptide 7, regulatory | 11 |
| 47 | Myl9 | Myosin, light polypeptide 9, regulatory | 2 |
| 48 | Mylk2 | Myosin, light polypeptide kinase 2, skeletal muscle | 2 |
| 49 | Mylk | Myosin, light polypeptide kinase | 16 |
| 50 | Mylip | Myosin regulatory light chain interacting protein | 13 |
| 51 | Myo5a | Myosin VA | 9 |
| 52 | Myo5b | Myosin VB | 18 |
| 53 | Myo5c | Myosin VC | 9 |
| 54 | Myo6 | Myosin VI | 9 |
| 55 | Myo7a | Myosin VIIA | 7 |
| 56 | Myo7b | Myosin VIIB | 18 |
| 57 | Myo10 | Myosin X | 15 |
| 58 | Myo15 | Myosin XV | 11 |
| 59 | Myo18a | Myosin XVIIIA | 11 |
| 60 | Myo18b | Myosin XVIIIB | 5 |
| 61 | Myo1a | Myosin IA | 10 |
| 62 | Myo1b | Myosin IB | 1 |
| 63 | Myo1c | Myosin IC | 11 |
| 64 | Myo1d | Myosin ID | 11 |
| 65 | Myo1e | Myosin IE | 9 |
| 66 | Myo1f | Myosin IF | 17 |
| 67 | Myo1g | Myosin IG | 11 |
| 68 | Myo1h | Myosin IH | 5 |
| 69 | Diaph1 | Diaphanous related formin 1 | 18 |
| 70 | Rdx | Radixin | 9 |
| 71 | Scin | Scinderin | 12 |
| 72 | Spta1 | Spectrin alpha, erythrocytic 1 | 1 |
| 73 | Sptan1 | Spectrin alpha, nonerythrocytic 1 | 2 |
| 74 | Sptb | Spectrin beta, erythrocytic | 12 |
| 75 | Sptbn1 | Spectrin beta, nonerythrocytic 1 | 11 |
| 76 | Sptbn2 | Spectrin beta, nonerythrocytic 2 | 19 |
| 77 | Sptbn4 | Spectrin beta, nonerythrocytic 4 | 7 |
| 78 | Spire1 | Spire type actin nucleation factor 1 | 18 |
| 79 | Spire2 | Spire type actin nucleation factor 2 | 8 |
| 80 | Tmod1 | Tropomodulin 1 | 4 |
| 81 | Tmod2 | Tropomodulin 2 | 9 |
| 82 | Tmod3 | Tropomodulin 3 | 9 |
| 83 | Tpm1 | Tropomyosin 1, alpha | 9 |
| 84 | Tpm3 | Tropomyosin 3, gamma | 3 |
| 85 | Tpm4 | Tropomyosin 4 | 8 |
| 86 | Tpm2 | Tropomyosin 2, beta | 4 |
| 87 | Vasp | Vasodilator-stimulated phosphoprotein | 7 |
| 88 | Vil1 | Villin 1 | 1 |

## Extended Data Table 2 | List of all murine microtubule binding proteins (*n*=98)

| No. | Gene symbol | Gene name | Chromosome location |
|---|---|---|---|
| 1 | *Aspm* | Abnormal spindle-like microcephaly-associated protein | 1 |
| 2 | *Cep55* | Centrosomal protein of 55 kDa | 19 |
| 3 | *Ckap5* | Cytoskeleton-associated pr5otein | 2 |
| 4 | *Clip1* | CAP-Gly domain containing linker protein 1 | 5 |
| 5 | *Clip2* | CAP-Gly domain containing linker protein 2 | 5 |
| 6 | *Dctn1* | Dynactin subunit 1 | 6 |
| 7 | *Dctn2* | Dynactin subunit 2 | 10 |
| 8 | *Dlgap5* | Disks large-associated protein 5 | 14 |
| 9 | *Dnah1* | Dynein axonemal heavy chain 1 | 14 |
| 10 | *Dnah10* | Dynein axonemal heavy chain 10 | 5 |
| 11 | *Dnah11* | Dynein axonemal heavy chain 11 | 12 |
| 12 | *Dnah12* | Dynein axonemal heavy chain 12 | 14 |
| 13 | *Dnah14* | Dynein axonemal heavy chain 14 | 1 |
| 14 | *Dnah17* | Dynein axonemal heavy chain 17 | 11 |
| 15 | *Dnah2* | Dynein axonemal heavy chain 2 | 11 |
| 16 | *Dnah3* | Dynein axonemal heavy chain 3 | 7 |
| 17 | *Dnah5* | Dynein axonemal heavy chain 5 | 15 |
| 18 | *Dnah6* | Dynein axonemal heavy chain 6 | 6 |
| 19 | *Dnah7a* | Dynein axonemal heavy chain 7A | 1 |
| 20 | *Dnah7b* | Dynein axonemal heavy chain 7B | 1 |
| 21 | *Dnah7c* | Dynein axonemal heavy chain 7C | 1 |
| 22 | *Dnah8* | Dynein axonemal heavy chain 8 | 17 |
| 23 | *Dnah9* | Dynein axonemal heavy chain 9 | 11 |
| 24 | *Dnai1* | Dyneins intermediate chain 1 | 4 |
| 25 | *Dnai2* | Dyneins intermediate chain 2 | 11 |
| 26 | *Dnal1* | Dynein light intermediate chains 2 | 12 |
| 27 | *Dnal4* | Dynein light intermediate chains 4 | 15 |
| 28 | *Dnali1* | Dynein light intermediate chains 1 | 4 |
| 29 | *Dync1h1* | Dynein cytoplasmic 1 heavy chain 1 | 12 |
| 30 | *Dync1li1* | Dynein cytoplasmic 1 light intermediate chain 1 | 9 |
| 31 | *Dync1li2* | Dynein cytoplasmic 1 light intermediate chain 2 | 8 |
| 32 | *Dync2li1* | Dynein cytoplasmic 2 light intermediate chain 1 | 17 |
| 33 | *Dynll1* | Dynein light chain 1, cytoplasmic | 5 |
| 34 | *Dynll2* | Dynein light chain 2, cytoplasmic | 11 |
| 35 | *Dynlrb1* | Dynein light chain roadblock-type 1 | 2 |
| 36 | *Dynlrb2* | Dynein light chain roadblock-type 2 | 8 |
| 37 | *Dynlt1* | Dynein light chain Tctex-type 1 | 17 |
| 38 | *Dynlt3* | Dynein light chain Tctex-type 3 | X |
| 39 | *Katna1* | Katanin p60 ATPase-containing subunit A1 | 10 |
| 40 | *Katnb1* | Katanin p60 ATPase-containing subunit B1 | 8 |
| 41 | *Kif11* | Kinesin family member 11 | 19 |
| 42 | *Kif12* | Kinesin family member 12 | 4 |
| 43 | *Kif13a* | Kinesin family member 13A | 13 |
| 44 | *Kif13b* | Kinesin family member 13B | 14 |
| 45 | *Kif14* | Kinesin family member 14 | 1 |
| 46 | *Kif15* | Kinesin family member 15 | 9 |
| 47 | *Kif16b* | Kinesin family member 16B | 2 |
| 48 | *Kif17* | Kinesin family member 17 | 4 |
| 49 | *Kif18a* | Kinesin family member 18A | 2 |
| 50 | *Kif19a* | Kinesin family member 19A | 11 |
| 51 | *Kif19b* | Kinesin family member 19B | 5 |
| 52 | *Kif1a* | Kinesin family member 1A | 1 |
| 53 | *Kif1b* | Kinesin family member 1B | 4 |
| 54 | *Kif1c* | Kinesin family member 1C | 11 |
| 55 | *Kif20a* | Kinesin family member 20A | 18 |
| 56 | *Kif20b* | Kinesin family member 20B | 19 |
| 57 | *Kif21a* | Kinesin family member 21A | 15 |
| 58 | *Kif21b* | Kinesin family member 21B | 1 |
| 59 | *Kif22* | Kinesin family member 22 | 7 |
| 60 | *Kif23* | Kinesin family member 23 | 9 |
| 61 | *Kif24* | Kinesin family member 24 | 4 |
| 62 | *Kif26a* | Kinesin family member 26A | 12 |
| 63 | *Kif26b* | Kinesin family member 26B | 1 |
| 64 | *Kif27* | Kinesin family member 27 | 13 |
| 65 | *Kif2a* | Kinesin family member 2A | 13 |
| 66 | *Kif2c* | Kinesin family member 2C | 4 |
| 67 | *Kif3a* | Kinesin family member 3A | 11 |
| 68 | *Kif3b* | Kinesin family member 3B | 2 |
| 69 | *Kif3c* | Kinesin family member 3C | 12 |
| 70 | *Kif4* | Kinesin family member 4A | X |
| 71 | *Kif5a* | Kinesin family member 5A | 10 |
| 72 | *Kif5b* | Kinesin family member 5B | 18 |
| 73 | *Kif5c* | Kinesin family member 5C | 2 |
| 74 | *Kif6* | Kinesin family member 6 | 17 |
| 75 | *Kif7* | Kinesin family member 7 | 7 |
| 76 | *Kif9* | Kinesin family member 9 | 9 |
| 77 | *Kifap3* | Kinesin-2 associated protein | 1 |
| 78 | *Kifc1* | Kinesin family member C1 | 17 |
| 79 | *Kifc2* | Kinesin family member C2 | 15 |
| 80 | *Kifc3* | Kinesin family member C3 | 8 |
| 81 | *Klc1* | Kinesin light chain 1 | 12 |
| 82 | *Klc2* | Kinesin light chain 2 | 19 |
| 83 | *Klc3* | Kinesin light chain 3 | 7 |
| 84 | *Klc4* | Kinesin light chain 4 | 17 |
| 85 | *Map1a* | Microtubule-associated protein 1A | 2 |
| 86 | *Map1b* | Microtubule-associated protein 1B | 13 |
| 87 | *Map2* | Microtubule-associated protein 2 | 1 |
| 88 | *Map4* | Microtubule-associated protein 4 | 9 |
| 89 | *Map6* | Microtubule-associated protein 6 | 7 |
| 90 | *Map7* | Microtubule-associated protein 7 | 10 |
| 91 | *Map9* | Microtubule-associated protein 9 | 3 |
| 92 | *Mapre1* | Microtubule–associated protein RP/EB family member 1 | 2 |
| 93 | *Mapre2* | Microtubule-associated protein RP/EB family member 2 | 18 |
| 94 | *Mapre3* | Microtubule-associated protein RP/EB family member 3 | 5 |
| 95 | *Mapt* | Microtubule associated protein tau | 11 |
| 96 | *Pafah1b1* | Platelet-activating factor acetylhydrolase IB subunit alpha | 11 |
| 97 | *Plec* | Plectin | 15 |
| 98 | *Tubg1* | Tubulin gamma 1 | 11 |

**Extended Data Table 3 | List of predicted neoantigens derived from DNGR-1$^{KO}$ regressor sarcoma cell lines ($n$=13)**

| Gene name | Predicted neoantigen | Mutated epitope sequence | HLA allele | IC$_{50}$ score (nM) |
|---|---|---|---|---|
| Myo18a | 88D10a | **LSFRTFLL** | H-2K$^b$ | 17.5 |
| Myo1f | 88D10b | SS**H**VRSLPL (Q->H) | H-2K$^b$ | 113.88 |
| Myo7b | 88D12a | FAI**S**NSCYF (A->S) | H-2D$^b$ | 8.78 |
| Myo15 | 88D12b | YSTLN**T**EHF (S->T) | H-2D$^b$ | 64.68 |
| Myo18a | 88D2a | **L**AAKFGSL (R->L) | H-2K$^b$ | 108.1 |
| Sptb | 88D2b | QAFSTYSTV (R->S) | H-2K$^b$ | 109.46 |
| Sptb | 88D2c | TAFE**L**ELHL (R->L) | H-2K$^b$ | 143.11 |
| Myo18a | 88D3 | **TSWWMRSGL** | H-2K$^b$ | 83.06 |
| Myl7 | 88D4a | **V**NIDYKSL (G->V) | H-2K$^b$ | 14 |
| Myh1 | 88D4b | SIYKLT**V**AV (G->V) | H-2K$^b$ | 66.99 |
| Myo1g | 88D6 | S**S**FGKYMDI (R->S) | H-2K$^b$ | 38.78 |
| Myo15 | 88D8a | **T**AFMYPWV (P->T) | H-2K$^b$ | 51.11 |
| Flnb | 88D8b | SAYG**I**PASL (V->I) | H-2K$^b$ | 56.66 |

| | |
|---|---|

# Reporting Summary

## Statistics

For all statistical analyses, confirm that the following items are present in the figure legend, table legend, main text, or Methods section.

| n/a | Confirmed | |
|---|---|---|
| ☐ | ☒ | The exact sample size ($n$) for each experimental group/condition, given as a discrete number and unit of measurement |
| ☐ | ☒ | A statement on whether measurements were taken from distinct samples or whether the same sample was measured repeatedly |
| ☐ | ☒ | The statistical test(s) used AND whether they are one- or two-sided *Only common tests should be described solely by name; describe more complex techniques in the Methods section.* |
| ☐ | ☒ | A description of all covariates tested |
| ☐ | ☒ | A description of any assumptions or corrections, such as tests of normality and adjustment for multiple comparisons |
| ☐ | ☒ | A full description of the statistical parameters including central tendency (e.g. means) or other basic estimates (e.g. regression coefficient) AND variation (e.g. standard deviation) or associated estimates of uncertainty (e.g. confidence intervals) |
| ☐ | ☒ | For null hypothesis testing, the test statistic (e.g. $F$, $t$, $r$) with confidence intervals, effect sizes, degrees of freedom and $P$ value noted *Give P values as exact values whenever suitable.* |
| ☒ | ☐ | For Bayesian analysis, information on the choice of priors and Markov chain Monte Carlo settings |
| ☒ | ☐ | For hierarchical and complex designs, identification of the appropriate level for tests and full reporting of outcomes |
| ☒ | ☐ | Estimates of effect sizes (e.g. Cohen's $d$, Pearson's $r$), indicating how they were calculated |

*Our web collection on statistics for biologists contains articles on many of the points above.*

## Software and code

Policy information about availability of computer code

| Data collection | n/a |
|---|---|
| Data analysis | The code used to analyse the data in this study is available in: https://github.com/FrancisCrickInstitute/DNGR1_XP |

For manuscripts utilizing custom algorithms or software that are central to the research but not yet described in published literature, software must be made available to editors and reviewers. We strongly encourage code deposition in a community repository (e.g. GitHub). See the Nature Portfolio guidelines for submitting code & software for further information.

## Data

Policy information about availability of data

All manuscripts must include a data availability statement. This statement should provide the following information, where applicable:

- Accession codes, unique identifiers, or web links for publicly available datasets
- A description of any restrictions on data availability
- For clinical datasets or third party data, please ensure that the statement adheres to our policy

Raw and processed data (whole exome sequencing) are submitted to European Nucleotide Archive (ENA) under the accession PRJEB100660.

# Research involving human participants, their data, or biological material

Policy information about studies with <u>human participants or human data</u>. See also policy information about <u>sex, gender (identity/presentation), and sexual orientation</u> and <u>race, ethnicity and racism</u>.

| | |
|---|---|
| Reporting on sex and gender | For the bioinformatic analysis of human cancer datasets, the association between CLEC9A and survival outcomes in patients with cancer were analysed using publicly available data downloaded from the cBioPortal for Cancer Genomics platform [http://cbioportal.org/], with source data from The Cancer Genome Atlas (TCGA) Pan-cancer Atlas. The focus of our analyses were on survival outcomes, including progression-free survival and overall survival, and were not segregated based on demographic features. All datasets include male and female patients. |
| Reporting on race, ethnicity, or other socially relevant groupings | See above. |
| Population characteristics | See above |
| Recruitment | n/a |
| Ethics oversight | n/a |

Note that full information on the approval of the study protocol must also be provided in the manuscript.

# Field-specific reporting

Please select the one below that is the best fit for your research. If you are not sure, read the appropriate sections before making your selection.

☒ Life sciences   ☐ Behavioural & social sciences   ☐ Ecological, evolutionary & environmental sciences

For a reference copy of the document with all sections, see <u>nature.com/documents/nr-reporting-summary-flat.pdf</u>

# Life sciences study design

All studies must disclose on these points even when the disclosure is negative.

| | |
|---|---|
| Sample size | We hypothesised that tumours developing in DNGR-1 KO mice were enriched in mutations in the n=88 genes encoding murine F-actin binding proteins (FABPs). Therefore, a power calculation for estimating the sample size required for MCA carcinogenesis model was performed by G.K. (Bioinformatics and Biostatistics team at the Francis Crick Institute), based on the following assumptions. There are in total 88 genes of interest encoding for FABPs. The Twist exome kit used for whole exome sequencing targets a total of 37,895,407bp or ~38Mb, with the 88 FABP genes comprising 334,760 of that, or 0.88% of the total region. Assuming that MCA fibrosarcomas have a high mutational burden of between 3,500 and 5,000 mutations based on previous literature, and the reference being aligned to is ~2.7Gb, a power calculation is generated using binomial probabilities: pbinom[0, N, prob=(burden*334760/2.7e9), lower=FALSE]. Therefore, it is estimated that n = 4–6 mice are required for each experimental arm, for in excess of 95% probability of at least one mouse having a variant within the FABP exon.<br><br>For experiments testing the tumour growth profile of individual primary cancer cell lines, at least n=3-5 mice were used per cohort, in line with previously published work, including by Schreiber and colleagues (Shankaran et al. 2001).<br>Likewise, for experiments testing the immunogenicity of neoantigen peptides in vivo or cross-priming of CD8 T cells in response to dead cell-associated antigen, at least n=3-5 mice were used per cohort. |
| Data exclusions | No data were excluded from the analyses. |
| Replication | All attempts at replication were successful, and where indicated, were noted in the figure legends. |
| Randomization | n/a |
| Blinding | n/a |

# Reporting for specific materials, systems and methods

We require information from authors about some types of materials, experimental systems and methods used in many studies. Here, indicate whether each material, system or method listed is relevant to your study. If you are not sure if a list item applies to your research, read the appropriate section before selecting a response.

## Materials & experimental systems

| n/a | Involved in the study |
|-----|----------------------|
| ☒ | ☐ Antibodies |
| ☐ | ☒ Eukaryotic cell lines |
| ☒ | ☐ Palaeontology and archaeology |
| ☐ | ☒ Animals and other organisms |
| ☐ | ☒ Clinical data |
| ☒ | ☐ Dual use research of concern |
| ☒ | ☐ Plants |

## Methods

| n/a | Involved in the study |
|-----|----------------------|
| ☒ | ☐ ChIP-seq |
| ☐ | ☒ Flow cytometry |
| ☒ | ☐ MRI-based neuroimaging |

# Eukaryotic cell lines

Policy information about cell lines and Sex and Gender in Research

| | |
|---|---|
| Cell line source(s) | All cell lines generated in this study were from mice of both male and female sex. MCA205, HeLa and RMA-S cell lines were obtained from the Francis Crick Institute Cell Services Science Technology platform. MuTu DC cells were obtained from Hans Acha-Orbea. |
| Authentication | The primary fibrosarcoma cell lines generated have been further characterised using the multiplex polymerase chain reaction (PCR) assay designed to profile mouse cell lines using primers targeting 18 mouse loci with highly polymorphic short tandem repeats (STRs), as previously described and validated by the Consortium for Mouse Cell Line Authentication (Almeida et al. 2019). This STR profiling was performed as part of the quality control process in curating a reference library for all the banked original primary cell lines, by the Cell Services Science Technology Platform at the Francis Crick Institute. |
| Mycoplasma contamination | All cell lines, including all the primary fibrosarcoma cell lines generated have been independently screened negative for mycoplasma contamination by the Cell Services Science Technology Platform at the Francis Crick Institute. In general, cells were cultured in antibiotic-free RPMI 1640 media for at least 2 days, before the submission of 5 to 10mL of confluent supernatant or the flask of live cells for simultaneous fluorescence staining and agar culture. |
| Commonly misidentified lines (See ICLAC register) | n/a |

# Animals and other research organisms

Policy information about studies involving animals; ARRIVE guidelines recommended for reporting animal research, and Sex and Gender in Research

| | |
|---|---|
| Laboratory animals | RAG1 KO (Rag1-/-), BATF3 KO (Batf3-/-), DNGR-1 KO (Clec9agfp/gfp or Clec9acre/cre), and WT (wild-type) mice on a C57BL/6 background were bred and maintained in specific-pathogen free conditions in the Biological Research Facility at The Francis Crick Institute. Experiments were commenced when mice were between 6 to 14 weeks old. |
| Wild animals | The study did not involve wild animals. |
| Reporting on sex | In all experiments, both male and female mice were used, unless otherwise indicated. In all loss-of-function experiments comparing DNGR-1 KO to WT mice, they were co-housed with sex- and age- matched WT controls for at least 3 weeks to eliminate any microbiota-dependent effects. |
| Field-collected samples | The study did not involve samples collected from the field. |
| Ethics oversight | All animal experiments were performed upon prospective approval of a study plan by the Biological Research Facility at the Francis Crick Institute, and strictly adhered to the Animals (Scientific Procedures) Act 1986. |

Note that full information on the approval of the study protocol must also be provided in the manuscript.

# Clinical data

Policy information about clinical studies

All manuscripts should comply with the ICMJE guidelines for publication of clinical research and a completed CONSORT checklist must be included with all submissions.

| | |
|---|---|
| Clinical trial registration | For the bioinformatic analysis of human cancer datasets, the association between CLEC9A and survival outcomes in patients with cancer were analysed using publicly available data downloaded from the cBioPortal for Cancer Genomics platform [http://cbioportal.org/], with source data from The Cancer Genome Atlas (TCGA) Pan-cancer Atlas. |
| Study protocol | n/a |
| Data collection | n/a |

| Outcomes | The focus of our analyses were on survival outcomes, including progression-free survival and overall survival, provided by TCGA. |

# Plants

| Seed stocks | n/a |

| Novel plant genotypes | n/a |

| Authentication | n/a |

# Flow Cytometry

## Plots

Confirm that:

☐ The axis labels state the marker and fluorochrome used (e.g. CD4-FITC).

☐ The axis scales are clearly visible. Include numbers along axes only for bottom left plot of group (a 'group' is an analysis of identical markers).

☐ All plots are contour plots with outliers or pseudocolor plots.

☐ A numerical value for number of cells or percentage (with statistics) is provided.

## Methodology

| Sample preparation | Peptide pulsed RMA-S cells or regressor tumour cells were harvested in PBS/EDTA, washed and stained with appropriate anti-MHC-I mabs. Splenocyte suspensions were prepared from spleens of immunised mice, red blood cell lysed and stained with H-2Kb/SIINFEKL pentamer reagent, followed by anti-CD8α, anti-CD44 and anti-CD19 antibodies. |

| Instrument | Stained cells were acquired on a LSRFortessa (BD Biosciences). |

| Software | Data were analysed using FlowJo software (Treestar). |

| Cell population abundance | This study did not involve cell sorting. Cells analysed were either cell lines (RMA-S or regressor tumour lines) or total splenocytes. For each sample of RMA-S, regressor tumour or CD8+CD19- splenocytes at least 10000 events were recorded. |

| Gating strategy | CD8 T cells were first gated on scatter (FSC/SSC), followed by CD8/CD19. Gated CD8+CD19- cells were then plotted for CD44 and H-2Kb/OVA peptide pentamer and the percentage of OVA-specific CD8 T cells was determined from gating on H-2Kb/OVA peptide pentamer+CD44+cells. |

☐ Tick this box to confirm that a figure exemplifying the gating strategy is provided in the Supplementary Information.

