## [Peer Review File · Nature Immunology]

Cross-presentation of dead cell-associated antigens shapes the neoantigenic landscape of tumour immunity

Corresponding Author: Professor Caetano Reis e Sousa

Version 0:

Reviewer comments:

Reviewer #1

(Remarks to the Author)

Lim et colleagues examine the role of DNGR1 in shaping the repertoire of immunogenic antigens in chemically induced cancers. The group of Caetano Reis e Sousa has previously shown that DNGR1 is an F-actin-binding receptor involved in cross-presentation of dead cell-associated antigens by dendritic cells. Here, the authors show that chemically induced tumors develop earlier in DNGR1 KO mice in comparison to WT mice. Furthermore, tumors from DNGR1 KO mice transplanted in WT host are more efficiently rejected than tumors from WT mice. These results show that tumors from DNGR1 mice are more immunogenic and DNGR1 is playing a role in the priming of immunity against chemically-induced tumors. Interestingly, and as previously shown, tumors from DNGR1 KO mice transplanted in DNGR1 host are similarly rejected. Next the authors assessed the neoantigen repertoire of tumors developed in WT, DNGR1 KO and RAG KO mice. They did not find differences in tumor mutation burden, or total numbers of predicted neoantigens. In contrast, they found an accumulation of predicted neoantigens in F-actin binding proteins in DNGR1 KO and RAG KO tumors in comparison to WT tumors, and in agreement with their previous work. Finally, they authors suggest that the antigenic visibility of F-actin binding proteins can be dependent on DNGR1. The manuscript is well written, the data are convincing, and the findings are interesting and highly relevant for cancer immunotherapy as they may help improve the identification of cancer neoantigens for vaccines or cell therapy.

Fig. 1a: Once mice develop tumors, do they grow at the same rate? A growth map similar to the one in Fig 2 would be nice to have.

Fig. 2b: Can the authors comment on the regression of 40% of the tumors from WT mice after transplantation in WT mice. The authors seem to suggest that the visibility of neoantigens may be different in spontaneous tumors and in transplanted tumors because of differential mechanism of uptake by DCs. Alternatively, the immunosuppressive tumor microenvironment could allow for tumors expressing immunogenic neoantigens to hide from the immune system.

What is the MHC class I level of expression on the tumors?

Extended Data Fig. 4: c) is missing

Fig. 3: The authors should provide more information on their neoantigen prediction method. It does not appear that they used RNAseq to select for expressed mutations. How did they choose a binding IC50 of 150nm as a cutoff? Binders with IC50<50nm are usually considered strong binders. Additionally other factors, such as pMHC stability may play an important role in immunogenicity.

Fig. 4: The authors' claim that tumors from DNGR1 KO and RAG2 KO mice are enriched for immunogenic mutations in actin binding proteins would have been much stronger if they indeed showed these mutations are immunogenic. Neoantigen prediction enriches for immunogenic mutations but only ~10% of the predicted mutations are truly immunogenic. They authors could easily evaluate neoantigen immunogenicity by immunizing mice with peptide-based vaccines, or RNA-based vaccines.

The authors suggest that DNGR1 plays a role in antigenic visibility and show that the growth of MCA205 tumors expressing low levels of mutated spectrin-beta 2 is slower in sGSN ko mice with hyper active DNGR1. How does tumor growth of

tumors from DNDR1 KO mice compare after transplantation in WT, DNDR1 KO and sGSN KO mice? This experiment could expand and confirm the findings with MCA205 line. How do the levels of mutated and WT spectrin-beta 2 in the engineered MCA205 line in comparison to the endogenous levels of spectrin-beta 2?

Reviewer #2

(Remarks to the Author)

While cancer cells mutate and produce neoantigens, not all neoantigens can activate immune responses (PMID: 27198675). It is unknown how cDC1 chooses which neoantigens to cross-present and this is focus of Lim et. al., in this work. In an exciting set of studies, the authors extend Schreiber's immunoediting hypothesis, originally showing roles for NK cells and $\gamma\delta$ T cells, now to cDC1 via the F-actin binding necrotic cell sensor signaling receptor DNDR1. The authors show that 3-methylcholanthrene (MCA)-induced tumors derived from DNDR1KO mice are more immunogenic (regressors), suggesting impaired immunoediting. The authors convincingly show that the rejection of these immunogenic tumors was immune dependent (growing in RAG1KO mice) but also importantly dependent on BATF3 expressing cDC1 although the role of DNDR1 in control of these tumors was dispensable. They also show the presence of significantly more neoantigens in the 'less edited' tumors derived from DNDR1KO mice as well as RAG1KO mice, and that the neoantigens in tumors derived from DNDR1KO mice were surprisingly mostly derived from F-actin binding proteins (FABP). They compare these to microtubule binding proteins as specificity controls, which they find were in turn abundant in RAG1KO but not DNDR1KO. These are novel and exciting findings.

The authors show that a number of the FABPs in DNDR1KO derived tumors contained mutations, concordant with the higher total predicted neoantigens they identified in regressor vs progressor tumors derived from DNDR1KO mice. They establish tumor cell lines from the immunogenic regressor tumors and identify two cell lines, one from DNDR1KO and the other from RAG1KO mice, where two (as opposed to one) mutations were found within spectrin- β and Myh1 (striated muscle myosin heavy chain) in DNDR1KO and RAG1KO mice, respectively. Spectrin- β was notable because of the former identification of a different mutant spectrin- β 2 (R913L) isoform by Schreiber as a neoantigen subject to immunoediting in MCA-induced tumors. The authors use this mutation as a validated neoantigen emerging in less edited tumors to test its emergence under conditions of impaired DNDR1 mediated immune editing. They find that mutant spectrin- β 2 (R913L)-high tumors were not as aggressive as WT spectrin- β 2-high tumors growing to smaller volumes in both WT and sGSNKO mice. Also, only the growth of mutant spectrin- β 2 (R913L)-low tumors was significantly retarded in soluble gelsolin (sGSN)KO mice where DNDR1 activity is enhanced (sGSN inhibits DNDR1).

The study presents evidence in support of the thesis that DNDR-1 engagement during early stages of tumor formation comprises an important barrier against tumor growth and expression of neoantigens. This is an important and novel finding. However, there are major concerns with the conclusions drawn and relate to the incompleteness of the studies as presented. T cell investigations demonstrating priming of anti-tumor CD8 T cells specific for mutant spectrin- β 2 is lacking and would provide valuable evidence. In short, the molecular mechanism driving the process is missing. The major concerns are outlined below.

It is assumed that DNDR1 favors the cross-presentation of antigens associated with the actin cytoskeleton. This would be a significant finding in and of itself. It is presently an assumption that is critical for the conclusions drawn, yet there is no data to support this assumption. It is important to examine whether DNDR1 favors the cross-presentation of antigens associated with the actin cytoskeleton of necrotic dying cells over non-actin associated antigens.

What is the nature of the two mutations identified in 88D2 within spectrin- β encoding Sptb? Do either of them lead to the R913L mutation? Why was spectrin- β 2 (R913L) and not these mutations (if different) selected for further examination of DNDR1 dependent clearance of tumors bearing FABP neoantigens?

There is an equal number of FABP mutations in RAG1KO derived regressor tumors, which diminishes the case that loss of DNDR1 mediated immune editing specifically in FABP and the differences in the number of FABP neoantigens between DNDR1KO vs RAG1KO shown in Fig. 3d is not statistically significant. $P=0.84$.

Derivation of high and low expressing WT spectrin- β and mutant spectrin- β 2 (R913L) MCA205 fibrosarcoma cell lines and transplantation into WT, sGSNKO or sGSN/DNDR1DKO mice comprised a series of experiments designed to examine whether DNDR1 engagement promotes effective tumor clearance of tumors harboring FABP neoantigens. This series of experiments open a number of questions, which the authors should have addressed leaving these studies presently as incomplete. Specifically:

-Why is DNDR1 control of mutant spectrin- β 2 tumors evident only for low and not high spectrin- β 2 (R913L)-expressing tumors? F-actin and not proteins associated with F-actin should determine the level of DNDR1 engagement, and as such, the schematic model shown in Extended Data Fig. 7 equates the level of F-actin, and by extension F-actin binding proteins, with the level of cross-presentation; higher levels leading to higher DNDR-1 dependent cross-presentation. However, the results shown in Fig. 4b,c show a correlation between mutant spectrin- β 2 levels and DNDR1-dependent tumor control. Given that F-actin is the ligand for DNDR-1, another interpretation of these results is that low mutant spectrin- β 2 expressing tumor cells present higher F-actin levels to DNDR1. Why do the authors equate the level of F-actin exposed in a dying cell to the level of FABP expression? What is the level of F-actin expression in high vs low spectrin- β 2 expressing tumor groups in Fig. 4b, and how does it correlate with DNDR-1 mediated tumor control?

-Is tumor control in these experiments dependent on CD8 T cells? This is important to the message that DNDR1 dependent

immunoediting shapes visibility of tumors to CD8 T cells.

-The results presented suggest that mutations render an FABP a neoantigen but this needs to be demonstrated directly. Does mutant spectrin-β2 specifically generate an anti-tumor CD8 T cell response? Is this response specific to mutant and not WT spectrin-β2? Is this CD8 T cell response dependent on DNGR1? Can it be mapped to the spectrin-β2 R913L mutation by testing the T cell response to peptides overlapping the R913 residue.

Other points:

- Which pairwise comparison does $P=0.002$ using unpaired test refer to in Fig. 1a? Consider moving Extended Data Fig. 2b adjacent to Fig. 1a.
- Consider moving Extended Data Fig. 2d adjacent to Fig. 1b.
- Statistical analyses comparing growths of spectrin-β2 (R913L)-high vs WT spectrin-β2-high in Fig. 4b should be drawn.
- The authors need to indicate the significance of the P-values shown in Fig. 3c-e in the figure legend.
- Tumor growth curves of high vs low spectrin-β/spectrin-β R913L expressing MCA205 cells should be shown on the same graph with statistics drawn to formally exclude a role for spectrin-β2 expression levels in tumor immunogenicity and/or tumor growth.
- Lines 181-182 "...immunoedited escapees that have lost mutated FABP epitopes, which we detect in WT but with lower prevalence in DNGR1KO mice" is confusing and should be re-written. As stated, it is not concordant with the data in Fig. 4a which show prevalence of mutations in FABP encoding genes in DNGR1KO compared to WT.
- Extended Fig. 5: Does the genomic mutational landscape include both progressive and regressive primary tumor cell lines? Could this be the reason why there is no significant variation in tumor mutational burden?

Reviewer #3

(Remarks to the Author)

In this study, the authors perform a series of in vivo experiments to conclude that cross-presentation of dead cell-associated antigens is necessary to shape the neoantigenic landscape of tumor immunity. While the authors use a relatively logical approach yet the initial enthusiasm for this study is dampened by its very broad and generalized assumptions based on a limited research design or lack of supportive data. It seems that the study frequently overinterprets the impact of DNGR1 and makes very strong conclusions about its "causative role" in defining the neoantigenic landscape. In its present form, this study seems too preliminary for publication. Below I elaborate my major concerns.

1. The authors don't seem to present clear evidence that supports this very strong statement (and other similarly strong statements throughout the manuscript): "Priming of anti-cancer cytotoxic CD8+ T cell depends largely on the proficient (neo-)antigen cross-presentation (XP) and cross-dressing capabilities of type 1 conventional dendritic cells (cDC1s)." While cDC1 are definitely important, depending on the context, cDC2, pDCs, mDCs and even macrophages have been found to be important for neoantigen presentation. This very generalized assumption about cDC1 seems to pave the way for a research design that is completely biased toward cDC1 in its entire study. If this is true, then the authors need to use depletion strategies to exclude the roles of cDC2, pDCs and macrophages before completely concentrating their study on cDC1.
2. The study makes generalized assumptions about DNGR1 being the most important player in the phagocytosis/efferocytosis system. There are many receptor-ligand systems important for phagocytosis/efferocytosis like CD47-SIRP1a, calreticulin-LRP, complement receptors, several scavenger receptors on macrophages etc. that can be equally important. The authors haven't provided any comparative evidence to support their claims since all experiments were immediately done with DNGR1-KO mice. This is too preliminary. For such broad conclusions as drawn by the authors, a lot more comparative evidence needs to be provided before settling on experiments with the "best hit", which may very well be DNGR1 or perhaps another target?
3. This conclusion "DNGR-1 restrains carcinogenesis" is not entirely supported by the data. The analyses is restricted to one class of chemical carcinogen i.e., MCA. However, most cancers of carcinogenic-origin are not induced by this class of chemical carcinogen. Most cancers are driven by tobacco-derived carcinogens (e.g., lung cancer, head & neck cancer, some GI/GU cancers) or UV radiation (e.g., melanoma). Thus the authors do not use an appropriate carcinogen for such generalized conclusions. While it is true that some classical experiments in past were performed with MCA yet this choice is outdated for current contexts. Even for sarcoma in humans, the high neoantigenic version might be a byproduct of dMMR or MSI-like pathology rather than MCA per se. Thus, this part of the study design may not be up-to-date and/or clinically relevant.
4. A large majority of neoantigen-enriched CRC have MLH1 disruptions behind it, creating dMMR or MSI-like pathology. Such MLH1 disruptions are not necessarily chemical carcinogen-driven. The authors need to knock-out MLH1 in cancer cells to truly connect CRC-relevant neoantigen drivers to DNGR1 system (or others, as suggested above).
5. Lack of use of physiologically-relevant carcinogenic or genetic drivers of neoantigen landscape in this study might explain some of the confusing observations on overall mutational burden or neoantigen quality that the authors observe, which do not align with what has emerged in the clinical immunotherapy research in the last few years.
6. The author's results of few dominant neoantigens derived from F-actin coding genes driving the entire system do not align with the clinical reality of anticancer immunity relevant neoantigenic landscape i.e., physiological anticancer immunity in many cases is driven by stochasticity within the neoantigenic landscape rather than a few selected dominant neoantigens. This can again indicate that the author's setup may not be physiologically relevant.
7. Immunoediting is only concluded on the basis of transplantation rejection/latency in WT mice – which might be a very old school way of analyses. It would be more appropriate to subsequently deplete the relevant immunoediting players via antibody-based depletion in WT mice e.g., anti-CD8, anti-CD4 or NK1.1 to see if rejection/latency can be ameliorated thereby truly proving immune-driven responses rather than any transplantation-relevant "artefacts".

8. BATF3 KO mice may not be appropriate. While it is presented as a model where only cDC1 are disrupted yet it has been repeatedly proven recently that BATF3 also has a role in T cell biology and thus BATF3 KO also have direct T cell defects that may have nothing to do with cDC1 or their interface with T cells. This is an important problem that requires urgent attention considering the cDC1-centered conclusions of this study.
9. The authors have not provided any detailed immunophenotyping data for the tumors for various myeloid (macrophages, DC subsets, and their polarization states) or T cell subsets (Tregs, CD8T subsets, exhausted/cytotoxic cells etc.) thereby making it impossible to understand the immunological basis of these phenotypes. This makes it impossible to fully understanding possible confounding immune-factors.
10. The authors draw a specific conclusion about CD8+T cell priming without specifically proving this via anti-CD8 depletion or genetic targeted disruptions in CD8+T cell compartments. This is essential.
11. Any immune-checkpoint blockade based analyses is missing in this study which makes it impossible to connect immunotherapy-relevant processes to this study's conclusions.
12. The authors need to quantify cell death in vivo and prove to what extent are the antigens derived via dead/dying cells via phagocytosis by only cDC1 and no other phagocytic innate immune cells or even B cells also present in the tumor. How are cDC1 favored over other myeloid cells or B cells for a specific as well as preferential uptake of neoantigens? How are these cells the only one that form an interface with CD8+T cells, even though other cells are reported to do the same? These questions create some confusion regarding this study's conclusions.
13. The study lacks clinical immunotherapy data analyses even though the hypothesis is derived from a clinical therapeutic problem. The authors show TCGA analyses, but these data are not relevant since large majority of TCGA patients didn't receive immunotherapy. This creates doubts on clinical applicability of these results for immunotherapy. Proper clinical data analyses with immunotherapy clinical trials, needs to be added to validate the clinical significance of these findings.

Reviewer #4

(Remarks to the Author)

The manuscript is clearly written and the findings are supported by clear state-of-the-art experimental approaches. The authors elaborate on the role of DNNGR-1/CLECG9a on immune-editing of FABP neoantigens in MCA-induced tumors. They convincingly show that Dngr1 KO mice compared to wt exhibit decreased priming of FABP neoantigen-specific T cells without markedly affecting the response to other neoantigens. Reconstruction experiments using MCA205 fibrosarcoma cell lines expressing high or low levels of WT or mutant-spectrin- β 2 confirmed that the DNNGR-1 immunomodulation is not an "all-or-none" phenomenon but rather quantitatively modulated.

Conceptually and experimentally I have no major comments.

Experimentally, in view of the focus of the paper on DNNGR-1 the authors may comment on the fact they did not include littermate controls for Dngr1 KO, though I admit that the experiments comparing "Dngr1 KO to wt strains included the use of mice that were co-housed for at least 3 weeks to eliminate any microbiota-dependent effects". "All mice were bred in-house". Have the authors used in particular experiments such littermate mice to exclude potential confounding genetic differences?

The authors could spend a paragraph for a less specialized readership explaining why DNNGR1-mediated acquiring of FABP neoantigens is a cross presentation event (and not a cross dressing).

Decision Letter:

7th Oct 2024

Dear Professor Reis e Sousa,

As you are aware, your Article, "Cross-presentation of dead cell-associated antigens shapes the neoantigenic landscape of tumour immunity" has been seen by 4 referees and they had various concerns, including mechanistic deficits and reviewer 3 was rather critical of the clinical applicability of the findings owing to the models used, for example.

We have now looked over your Author Response to these concerns, thanks for sending that to me. I am pleased to say that we are happy with your revision plan. We do think that the mechanistic insight you plan to 'take' from another paper you were preparing is quite important to add here, even more so given this concern around the clinical relevance of the present study. If the reviewers are happy with the results you get from these revision plans, then we would certainly over-rule reviewer 3 regarding that criticism, as we think that some discussion of the clinical limitations would be enough if other issues were addressed. The only other comment I have, is that you seem to have had trouble interpreting one of the reviewer 3 comments (number 11). It seemed like a fairly clear request to us that they want you to add an ICB treatment to your tumor models to give some clinical translatability, so not quite clear to us what the confusion was here. In any case, again, we would not necessarily expect you to do this experiment for the same reasons stated above, so would leave that up to you.

In short, we would be very interested in considering a revised manuscript as per your plan.

If you choose to revise your manuscript taking into account all reviewer and editor comments, please highlight all changes in the manuscript text file in Microsoft Word format.

* If you have not done so already please begin to revise your manuscript so that it conforms to our Article format instructions at <http://www.nature.com/ni/authors/index.html>. Refer also to any guidelines provided in this letter.

The Reporting Summary can be found here:

- that unprocessed scans are clearly labelled and match the gels and western blots presented in figures.
- that control panels for gels and western blots are appropriately described as loading or sample processing controls
- all images in the paper are checked for duplication of panels and for splicing of gel lanes.

Extended Data figures and tables are online-only (appearing in the online PDF and full-text HTML version of the paper), peer-reviewed display items that provide essential background to the Article but are not included in the printed version of the paper due to space constraints or being of interest only to a few specialists. A maximum of ten Extended Data display items (figures and tables) is typically permitted. When re-submitting your manuscript, please ensure that any supplementary figures and tables that are more critical to the manuscript's conclusions are converted to Extended data to increase these data's visibility.

Link Redacted

If you wish to submit a suitably revised manuscript we would hope to receive it within 6 months. If you cannot send it within this time, please let us know. We will be happy to consider your revision so long as nothing similar has been accepted for publication at Nature Immunology or published elsewhere.

Nature Immunology is committed to improving transparency in authorship. As part of our efforts in this direction, we are now requesting that all authors identified as 'corresponding author' on published papers create and link their Open Researcher and Contributor Identifier (ORCID) with their account on the Manuscript Tracking System (MTS), prior to acceptance. ORCID helps the scientific community achieve unambiguous attribution of all scholarly contributions. You can create and link your ORCID from the home page of the MTS by clicking on 'Modify my Springer Nature account'. For more information please visit www.springernature.com/orcid.

Thank you for the opportunity to review your work.

Sincerely,

Nick Bernard, PhD
Senior Editor
Nature Immunology

Reviewers' Comments:

Reviewer #1 (Remarks to the Author):

Lim et colleagues examine the role of DNGR1 in shaping the repertoire of immunogenic antigens in chemically induced cancers. The group of Caetano Reis e Sousa has previously shown that DNGR1 is an F-actin-binding receptor involved in cross-presentation of dead cell-associated antigens by dendritic cells. Here, the authors show that chemically induced tumors develop earlier in DNGR1 KO mice in comparison to WT mice. Furthermore, tumors from DNGR1 KO mice transplanted in WT host are more efficiently rejected than tumors from WT mice. These results show that tumors from DNGR1 mice are more immunogenic and DNGR1 is playing a role in the priming of immunity against chemically-induced tumors. Interestingly, and as previously shown, tumors from DNGR1 KO mice transplanted in DNGR1 host are similarly rejected. Next the authors assessed the neoantigen repertoire of tumors developed in WT, DNGR1 KO and RAG KO mice. They did not find differences in tumor mutation burden, or total numbers of predicted neoantigens. In contrast, they found an accumulation of predicted neoantigens in F-actin binding proteins in DNGR1 KO and RAG KO tumors in comparison to WT tumors, and in agreement with their previous work. Finally, they authors suggest that the antigenic visibility of F-actin binding proteins can be dependent on DNGR1. The manuscript is well written, the data are convincing, and the findings are interesting and highly relevant for cancer immunotherapy as they may help improve the identification of cancer neoantigens for vaccines or cell therapy.

Fig. 1a: Once mice develop tumors, do they grow at the same rate? A growth map similar to the one in Fig 2 would be nice to have.

Fig. 2b: Can the authors comment on the regression of 40% of the tumors from WT mice after transplantation in WT mice. The authors seem to suggest that the visibility of neoantigens may be different in spontaneous tumors and in transplanted tumors because of differential mechanism of uptake by DCs. Alternatively, the immunosuppressive tumor microenvironment could allow for tumors expressing immunogenic neoantigens to hide from the immune system.

What is the MHC class I level of expression on the tumors?

Extended Data Fig. 4: c) is missing

Fig. 3: The authors should provide more information on their neoantigen prediction method. It does not appear that they used RNAseq to select for expressed mutations. How did they choose a binding IC50 of 150nm as a cutoff? Binders with IC50<50nm are usually considered strong binders. Additionally other factors, such as pMHC stability may play an important role in immunogenicity.

Fig. 4: The authors' claim that tumors from DNGR1 KO and RAG2 KO mice are enriched for immunogenic mutations in actin binding proteins would have been much stronger if they indeed showed these mutations are immunogenic. Neoantigen prediction enriches for immunogenic mutations but only ~10% of the predicted mutations are truly immunogenic. They authors could easily evaluate neoantigen immunogenicity by immunizing mice with peptide-based vaccines, or RNA-based vaccines.

The authors suggest that DNGR1 plays a role in antigenic visibility and show that the growth of MCA205 tumors expressing low levels of mutated spectrin-beta 2 is slower in sGSN ko mice with hyper active DNGR1. How does tumor growth of tumors from DNGR1 KO mice compare after transplantation in WT, DNGR1 KO and sGSN KO mice? This experiment could expand and confirm the findings with MCA205 line. How do the levels of mutated and WT spectrin-beta 2 in the engineered MCA205 line in comparison to the endogenous levels of spectrin-beta 2?

Reviewer #2 (Remarks to the Author):

While cancer cells mutate and produce neoantigens, not all neoantigens can activate immune responses (PMID: 27198675). It is unknown how cDC1 chooses which neoantigens to cross-present and this is focus of Lim et. al., in this work. In an exciting set of studies, the authors extend Schreiber's immunoediting hypothesis, originally showing roles for NK cells and $\gamma\delta$ T cells, now to cDC1 via the F-actin binding necrotic cell sensor signaling receptor DNGR1. The authors show that 3-methylcholanthrene (MCA)-induced tumors derived from DNGR1KO mice are more immunogenic (regressors), suggesting impaired immunoediting. The authors convincingly show that the rejection of these immunogenic tumors was immune dependent (growing in RAG1KO mice) but also importantly dependent on BATF3 expressing cDC1 although the role of DNGR1 in control of these tumors was dispensable. They also show the presence of significantly more neoantigens in the 'less edited' tumors derived from DNGR1KO mice as well as RAG1KO mice, and that the neoantigens in tumors derived from DNGR1KO mice were surprisingly mostly derived from F-actin binding proteins (FABP). They compare these to microtubule binding proteins as specificity controls, which they find were in turn abundant in RAG1KO but not DNGR1KO. These are novel and exciting findings.

The authors show that a number of the FABPs in DNGR1KO derived tumors contained mutations, concordant with the higher total predicted neoantigens they identified in regressor vs progressor tumors derived from DNGR1KO mice. They establish tumor cell lines from the immunogenic regressor tumors and identify two cell lines, one from DNGR1KO and the other from RAG1KO mice, where two (as opposed to one) mutations were found within spectrin- β and Myh1 (striated muscle myosin heavy chain) in DNGR1KO and RAG1KO mice, respectively. Spectrin- β was notable because of the former identification of a different mutant spectrin- β 2 (R913L) isoform by Schreiber as a neoantigen subject to immunoediting in

MCA-induced tumors. The authors use this mutation as a validated neoantigen emerging in less edited tumors to test its emergence under conditions of impaired DNGR1 mediated immune editing. They find that mutant spectrin- β 2 (R913L)-high tumors were not as aggressive as WT spectrin- β 2-high tumors growing to smaller volumes in both WT and sGSNKO mice. Also, only the growth of mutant spectrin- β 2 (R913L)-low tumors was significantly retarded in soluble gelsolin (sGSN)KO mice where DNGR1 activity is enhanced (sGSN inhibits DNGR1).

The study presents evidence in support of the thesis that DNGR-1 engagement during early stages of tumor formation comprises an important barrier against tumor growth and expression of neoantigens. This is an important and novel finding. However, there are major concerns with the conclusions drawn and relate to the incompleteness of the studies as presented. T cell investigations demonstrating priming of anti-tumor CD8 T cells specific for mutant spectrin- β 2 is lacking and would provide valuable evidence. In short, the molecular mechanism driving the process is missing. The major concerns are outlined below.

It is assumed that DNGR1 favors the cross-presentation of antigens associated with the actin cytoskeleton. This would be a significant finding in and of itself. It is presently an assumption that is critical for the conclusions drawn, yet there is no data to support this assumption. It is important to examine whether DNGR1 favors the cross-presentation of antigens associated with the actin cytoskeleton of necrotic dying cells over non-actin associated antigens.

What is the nature of the two mutations identified in 88D2 within spectrin- β encoding Sptb? Do either of them lead to the R913L mutation? Why was spectrin- β 2 (R913L) and not these mutations (if different) selected for further examination of DNGR1 dependent clearance of tumors bearing FABP neoantigens?

There is an equal number of FABP mutations in RAG1KO derived regressor tumors, which diminishes the case that loss of DNGR1 mediated immune editing specifically in FABP and the differences in the number of FABP neoantigens between DNGR1KO vs RAG1KO shown in Fig. 3d is not statistically significant. $P=0.84$.

Derivation of high and low expressing WT spectrin- β and mutant spectrin- β 2 (R913L) MCA205 fibrosarcoma cell lines and transplantation into WT, sGSNKO or sGSN/DNGR1DKO mice comprised a series of experiments designed to examine whether DNGR1 engagement promotes effective tumor clearance of tumors harboring FABP neoantigens. This series of experiments open a number of questions, which the authors should have addressed leaving these studies presently as incomplete. Specifically:

-Why is DNGR1 control of mutant spectrin- β 2 tumors evident only for low and not high spectrin- β 2 (R913L)-expressing tumors? F-actin and not proteins associated with F-actin should determine the level of DNGR1 engagement, and as such, the schematic model shown in Extended Data Fig. 7 equates the level of F-actin, and by extension F-actin binding proteins, with the level of cross-presentation; higher levels leading to higher DNGR-1 dependent cross-presentation. However, the results shown in Fig. 4b,c show a correlation between mutant spectrin- β 2 levels and DNGR1-dependent tumor control. Given that F-actin is the ligand for DNGR-1, another interpretation of these results is that low mutant spectrin- β 2 expressing tumor cells present higher F-actin levels to DNGR1. Why do the authors equate the level of F-actin exposed in a dying cell to the level of FABP expression? What is the level of F-actin expression in high vs low spectrin- β 2 expressing tumor groups in Fig. 4b, and how does it correlate with DNGR-1 mediated tumor control?

-Is tumor control in these experiments dependent on CD8 T cells? This is important to the message that DNGR1 dependent immunoeediting shapes visibility of tumors to CD8 T cells.

-The results presented suggest that mutations render an FABP a neoantigen but this needs to be demonstrated directly. Does mutant spectrin- β 2 specifically generate an anti-tumor CD8 T cell response? Is this response specific to mutant and not WT spectrin- β 2? Is this CD8 T cell response dependent on DNGR1? Can it be mapped to the spectrin- β 2 R913L mutation by testing the T cell response to peptides overlapping the R913 residue.

Other points:

- Which pairwise comparison does $P=0.002$ using unpaired test refer to in Fig. 1a? Consider moving Extended Data Fig. 2b adjacent to Fig. 1a.
- Consider moving Extended Data Fig. 2d adjacent to Fig. 1b.
- Statistical analyses comparing growths of spectrin- β 2 (R913L)-high vs WT spectrin- β 2-high in Fig. 4b should be drawn.
- The authors need to indicate the significance of the P-values shown in Fig. 3c-e in the figure legend.
- Tumor growth curves of high vs low spectrin- β /spectrin- β R913L expressing MCA205 cells should be shown on the same graph with statistics drawn to formally exclude a role for spectrin- β 2 expression levels in tumor immunogenicity and/or tumor growth.
- Lines 181-182 "...immunoeedited escapees that have lost mutated FABP epitopes, which we detect in WT but with lower prevalence in DNGR1KO mice" is confusing and should be re-written. As stated, it is not concordant with the data in Fig. 4a which show prevalence of mutations in FABP encoding genes in DNGR1KO compared to WT.
- Extended Fig. 5: Does the genomic mutational landscape include both progressive and regressive primary tumor cell lines? Could this be the reason why there is no significant variation in tumor mutational burden?

Reviewer #3 (Remarks to the Author):

In this study, the authors perform a series of in vivo experiments to conclude that cross-presentation of dead cell-associated antigens is necessary to shape the neoantigenic landscape of tumor immunity. While the authors use a relatively logical approach yet the initial enthusiasm for this study is dampened by its very broad and generalized assumptions based on a limited research design or lack of supportive data. It seems that the study frequently overinterprets the impact of DNDR1 and makes very strong conclusions about its "causative role" in defining the neoantigenic landscape. In its present form, this study seems too preliminary for publication. Below I elaborate my major concerns.

1. The authors don't seem to present clear evidence that supports this very strong statement (and other similarly strong statements throughout the manuscript): "Priming of anti-cancer cytotoxic CD8+ T cell depends largely on the proficient (neo-)antigen cross-presentation (XP) and cross-dressing capabilities of type 1 conventional dendritic cells (cDC1s)." While cDC1 are definitely important, depending on the context, cDC2, pDCs, moDCs and even macrophages have been found to be important for neoantigen presentation. This very generalized assumption about cDC1 seems to pave the way for a research design that is completely biased toward cDC1 in its entire study. If this is true, then the authors need to use depletion strategies to exclude the roles of cDC2, pDCs and macrophages before completely concentrating their study on cDC1.
2. The study makes generalized assumptions about DNDR1 being the most important player in the phagocytosis/efferocytosis system. There are many receptor-ligand systems important for phagocytosis/efferocytosis like CD47-SIRP1a, calreticulin-LRP, complement receptors, several scavenger receptors on macrophages etc. that can be equally important. The authors haven't provided any comparative evidence to support their claims since all experiments were immediately done with DNDR1-KO mice. This is too preliminary. For such broad conclusions as drawn by the authors, a lot more comparative evidence needs to be provided before settling on experiments with the "best hit", which may very well be DNDR1 or perhaps another target?
3. This conclusion "DNDR1 restrains carcinogenesis" is not entirely supported by the data. The analyses is restricted to one class of chemical carcinogen i.e., MCA. However, most cancers of carcinogenic-origin are not induced by this class of chemical carcinogen. Most cancers are driven by tobacco-derived carcinogens (e.g., lung cancer, head & neck cancer, some GI/GU cancers) or UV radiation (e.g., melanoma). Thus the authors do not use an appropriate carcinogen for such generalized conclusions. While it is true that some classical experiments in past were performed with MCA yet this choice is outdated for current contexts. Even for sarcoma in humans, the high neoantigenic version might be a byproduct of dMMR or MSI-like pathology rather than MCA per se. Thus, this part of the study design may not be up-to-date and/or clinically relevant.
4. A large majority of neoantigen-enriched CRC have MLH1 disruptions behind it, creating dMMR or MSI-like pathology. Such MLH1 disruptions are not necessarily chemical carcinogen-driven. The authors need to knock-out MLH1 in cancer cells to truly connect CRC-relevant neoantigen drivers to DNDR1 system (or others, as suggested above).
5. Lack of use of physiologically-relevant carcinogenic or genetic drivers of neoantigen landscape in this study might explain some of the confusing observations on overall mutational burden or neoantigen quality that the authors observe, which do not align with what has emerged in the clinical immunotherapy research in the last few years.
6. The author's results of few dominant neoantigens derived from F-actin coding genes driving the entire system do not align with the clinical reality of anticancer immunity relevant neoantigenic landscape i.e., physiological anticancer immunity in many cases is driven by stochasticity within the neoantigenic landscape rather than a few selected dominant neoantigens. This can again indicate that the author's setup may not be physiologically relevant.
7. Immunoediting is only concluded on the basis of transplantation rejection/latency in WT mice – which might be a very old school way of analyses. It would be more appropriate to subsequently deplete the relevant immunoediting players via antibody-based depletion in WT mice e.g., anti-CD8, anti-CD4 or NK1.1 to see if rejection/latency can be ameliorated thereby truly proving immune-driven responses rather than any transplantation-relevant "artefacts".
8. BATF3 KO mice may not be appropriate. While it is presented as a model where only cDC1 are disrupted yet it has been repeatedly proven recently that BATF3 also has a role in T cell biology and thus BATF3 KO also have direct T cell defects that may have nothing to do with cDC1 or their interface with T cells. This is an important problem that requires urgent attention considering the cDC1-centered conclusions of this study.
9. The authors have not provided any detailed immunophenotyping data for the tumors for various myeloid (macrophages, DC subsets, and their polarization states) or T cell subsets (Tregs, CD8T subsets, exhausted/cytotoxic cells etc.) thereby making it impossible to understand the immunological basis of these phenotypes. This makes it impossible to fully understanding possible confounding immune-factors.
10. The authors draw a specific conclusion about CD8+T cell priming without specifically proving this via anti-CD8 depletion or genetic targeted disruptions in CD8+T cell compartments. This is essential.
11. Any immune-checkpoint blockade based analyses is missing in this study which makes it impossible to connect immunotherapy-relevant processes to this study's conclusions.
12. The authors need to quantify cell death in vivo and prove to what extent are the antigens derived via dead/dying cells via phagocytosis by only cDC1 and no other phagocytic innate immune cells or even B cells also present in the tumor. How are cDC1 favored over other myeloid cells or B cells for a specific as well as preferential uptake of neoantigens? How are these cells the only one that form an interface with CD8+T cells, even though other cells are reported to do the same? These questions create some confusion regarding this study's conclusions.
13. The study lacks clinical immunotherapy data analyses even though the hypothesis is derived from a clinical therapeutic problem. The authors show TCGA analyses, but these data are not relevant since large majority of TCGA patients didn't receive immunotherapy. This creates doubts on clinical applicability of these results for immunotherapy. Proper clinical data analyses with immunotherapy clinical trials, needs to be added to validate the clinical significance of these findings.

Reviewer #4 (Remarks to the Author):

The manuscript is clearly written and the findings are supported by clear state-of-the-art experimental approaches. The authors elaborate on the role of DNGR-1/CLEC9a on immune-editing of FABP neoantigens in MCA-induced tumors. They convincingly show that Dngr1 KO mice compared to wt exhibit decreased priming of FABP neoantigen-specific T cells without markedly affecting the response to other neoantigens. Reconstruction experiments using MCA205 fibrosarcoma cell lines expressing high or low levels of WT or mutant-spectrin- β 2 confirmed that the DNGR-1 immunomodulation is not an "all-or-none" phenomenon but rather quantitatively modulated.

Conceptually and experimentally I have no major comments.

Experimentally, in view of the focus of the paper on DNGR-1 the authors may comment on the fact they did not include littermate controls for Dngr1 KO, though I admit that the experiments comparing "Dngr1 KO to wt strains included the use of mice that were co-housed for at least 3 weeks to eliminate any microbiota-dependent effects". "All mice were bred in-house". Have the authors used in particular experiments such littermate mice to exclude potential confounding genetic differences?

The authors could spend a paragraph for a less specialized readership explaining why DNGR1-mediated acquiring of FABP neoantigens is a cross presentation event (and not a cross dressing).

Version 1:

Reviewer comments:

Reviewer #1

(Remarks to the Author)

I appreciate the authors' thoughtful responses to my comments and the revision made to the manuscript.

My comments are below:

The authors showed that 2/9 predicted mutated FABP neoantigens from DNGR1 KO mice are immunogenic (fig4b). The low frequency of immunogenic mutations is not surprising, but it also somewhat weakens the authors' claim that there is accumulation of FABP neoantigens in DNGR1 KO mice due to reduced immunoeediting. The n is small and drawing such a conclusion is difficult. Did the authors examine the immunogenicity of the mutated FABPs in WT mice? if none were immunogenic it would strengthen the conclusion.

In addition, I disagree with the authors' decision not to use RNA-seq as a filter for neoantigen expression (Fig 3). If this data are available, it should be included, as immunoeediting can occur through the selection of cells that downregulate the mutant protein. Again, it is difficult for me to see an association between the enrichment of predicted FABP neoantigens and reduced immunoeediting in absence of expression - especially since only a minority of the predicted neoantigens are actually immunogenic. The data from Fig. 4b provides some additional nuance, as it seems that DNGR1 and sGSN may be more critical in cross-presentation of FABP neoantigens expressed at lower level.

(Remarks on code availability)

link above does not work - https://github.com/FrancisCrickInstitute/DNGR1_XP

Reviewer #2

(Remarks to the Author)

The authors have skillfully addressed my critiques, significantly enhancing the manuscript. The strategic use of ovalbumin variants (LA-OVA and mutLA-OVA) to modulate F-actin binding and its impact on the DNGR-1-dependent cross-presentation, by both Mutu DC and primary cDC1, is a compelling approach, clearly demonstrated in Fig. 5a-d. The authors extended their results to in vivo immunization showing the superiority of F-actin anchored antigen in eliciting CD8 T cell responses and included immunization of secreted gelsolin deficient mice (Fig. 5f,g). The presented data are robust and engaging, adding substantial value to the work.

The inclusion of the CD8 T cell depletion assay is a notable strength, providing clear evidence of the direct role of CD8 T cells (Ext. Fig. 3b). Additionally, the RMA-S stabilization assay effectively supports the direct interaction of the FABP neoantigen with H-2kb/H-2db (Fig. 4a,b), and validation in vivo (Fig. 4c) further bolstering the study's conclusions. These well-executed additions have strengthened the manuscript, making it a strong contribution to the field.

J. Magarian Blander

(Remarks on code availability)

The link was broken. I cannot comment as to whether I could assess the code's suitability.

Reviewer #4

(Remarks to the Author)

The authors have adequately responded to my remarks.

(Remarks on code availability)

Decision Letter:

17th Sep 2025

Dear Professor Reis e Sousa,

Your Article, "Cross-presentation of dead cell-associated antigens shapes the neoantigenic landscape of tumour immunity" has now been seen by the original reviewer 1,2 and 4.

Given the very clinically oriented comments of reviewer 3 and that you had mostly not addressed their comments with new data, we decided to leave them out of the re-review process and instead we asked reviewer 1 and reviewer 4 to comment on your response to reviewer 3. These mediation comments were provided in private comments to the editors only and the reviewers are mostly satisfied by your responses to reviewer 3. Reviewer 2 and 4 are also generally happy with the overall revision now, but reviewer 1 has some lingering concerns that we would like addressed if at all possible. Also two of the reviewers noted that the link to your despoited code was not working.

All that said, are very keen to consider publication of this study but first we invite you to revise the manuscript taking into account all reviewer and editor comments. Please highlight all changes in the manuscript text file in Microsoft Word format.

* If you have not done so already please begin to revise your manuscript so that it conforms to our Article format instructions at <http://www.nature.com/ni/authors/index.html>. Refer also to any guidelines provided in this letter.

* Please include a revised version of any required reporting checklist. It will be available to referees to aid in their evaluation of the manuscript goes back for peer review. They are available here:

Reporting summary:

Please note, Extended Data figures and tables are online-only (appearing in the online PDF and full-text HTML version of the paper), peer-reviewed display items that provide essential background to the Article but are not included in the printed version of the paper due to space constraints or being of interest only to a few specialists. A maximum of ten Extended Data display items (figures and tables) is typically permitted. When re-submitting your manuscript, please ensure that any supplementary figures and tables that are more critical to the manuscript's conclusions are converted to Extended data to increase these data's visibility.

Link Redacted

We hope to receive your revised manuscript within two weeks. If you cannot send it within this time, please let us know. We will be happy to consider your revision so long as nothing similar has been accepted for publication at Nature Immunology or published elsewhere.

Nature Immunology is committed to improving transparency in authorship. As part of our efforts in this direction, we are now requesting that all authors identified as 'corresponding author' on published papers create and link their Open Researcher and Contributor Identifier (ORCID) with their account on the Manuscript Tracking System (MTS), prior to acceptance. ORCID helps the scientific community achieve unambiguous attribution of all scholarly contributions. You can create and link your ORCID from the home page of the MTS by clicking on 'Modify my Springer Nature account'. For more information please visit www.springernature.com/orcid.

Sincerely,

Nick Bernard, PhD
Senior Editor
Nature Immunology

Reviewers' Comments:

Reviewer #1 (Remarks to the Author):

I appreciate the authors' thoughtful responses to my comments and the revision made to the manuscript.

My comments are below:

The authors showed that 2/9 predicted mutated FABP neoantigens from DNGR1 KO mice are immunogenic (fig4b). The low frequency of immunogenic mutations is not surprising, but it also somewhat weakens the authors' claim that there is accumulation of FABP neoantigens in DNGR1 KO mice due to reduced immunoediting. The n is small and drawing such a conclusion is difficult. Did the authors examine the immunogenicity of the mutated FABPs in WT mice? if none were immunogenic it would strengthen the conclusion.

In addition, I disagree with the authors' decision not to use RNA-seq as a filter for neoantigen expression (Fig 3). If this data are available, it should be included, as immunoediting can occur through the selection of cells that downregulate the mutant protein. Again, it is difficult for me to see an association between the enrichment of predicted FABP neoantigens and reduced immunoediting in absence of expression - especially since only a minority of the predicted neoantigens are actually immunogenic. The data from Fig. 4b provides some additional nuance, as it seems that DNGR1 and sGSN may be more critical in cross-presentation of FABP neoantigens expressed at lower level.

Reviewer #1 (Remarks on code availability):

link above does not work - https://github.com/FrancisCrickInstitute/DNGR1_XP

Reviewer #2 (Remarks to the Author):

The authors have skillfully addressed my critiques, significantly enhancing the manuscript. The strategic use of ovalbumin variants (LA-OVA and mutLA-OVA) to modulate F-actin binding and its impact on the DNGR-1-dependent cross-presentation, by both Mutu DC and primary cDC1, is a compelling approach, clearly demonstrated in Fig. 5a-d. The authors extended their results to in vivo immunization showing the superiority of F-actin anchored antigen in eliciting CD8 T cell responses and included immunization of secreted gelsolin deficient mice (Fig. 5f,g). The presented data are robust and engaging, adding substantial value to the work.

The inclusion of the CD8 T cell depletion assay is a notable strength, providing clear evidence of the direct role of CD8 T cells (Ext. Fig. 3b). Additionally, the RMA-S stabilization assay effectively supports the direct interaction of the FABP neoantigen with H-2kb/H-2db (Fig. 4a,b), and validation in vivo (Fig. 4c) further bolstering the study's conclusions. These well-executed additions have strengthened the manuscript, making it a strong contribution to the field.

J. Magarian Blander

Reviewer #2 (Remarks on code availability):

The link was broken. I cannot comment as to whether I could assess the code's suitability.

Reviewer #4 (Remarks to the Author):

The authors have adequately responded to my remarks.

Version 2:

Decision Letter:

Our ref: NI-A38558B

24th Sep 2025

Dear Dr. Reis e Sousa,

Thank you for submitting your revised manuscript "Cross-presentation of dead cell-associated antigens shapes the neoantigenic landscape of tumour immunity" (NI-A38558B). It has not been returned to the reviewers again as we are satisfied by your responses and revisions.

Therefore we'll be happy in principle to publish it in Nature Immunology, pending minor revisions to comply with our editorial and formatting guidelines.

We will now perform detailed checks on your paper and will send you a checklist detailing our editorial and formatting requirements in about a week. Please do not upload the final materials and make any revisions until you receive this additional information from us.

If you had not uploaded a Word file for the current version of the manuscript, we will need one before beginning the editing process; please email that to immunology@us.nature.com at your earliest convenience.

Thank you again for your interest in Nature Immunology Please do not hesitate to contact me if you have any questions.

Sincerely,

Nick Bernard, PhD
Senior Editor
Nature Immunology

Response to reviewers (in blue)

We thank the reviewers for their appraisal of our study and their suggested improvements and clarifications. We now provide a revised manuscript containing a large body of additional data. We believe these new data not only alleviate the reviewers' major concerns but considerably extend our original observations, enhancing the overall quality and impact of the study.

We report a proof-of-principle study that reveals a potential mechanism by which the antigenic visibility of tumours can be shaped by cross-presentation. We are pleased to read overall supportive recommendations by Reviewers 1, 2 and 4. With regards to Reviewer 3, we believe that there is a misunderstanding of the fundamental premise of our hypothesis-driven study. While we agree that it remains to be shown whether our findings are clinically relevant, we believe that our results suggest that this is certainly immunologically relevant and warrants further attention.

Our point-by-point reply follows below.

Reviewer #1

(Remarks to the Author)

Lim et colleagues examine the role of DNGR1 in shaping the repertoire of immunogenic antigens in chemically induced cancers. The group of Caetano Reis e Sousa has previously shown that DNGR1 is an F-actin-binding receptor involved in cross-presentation of dead cell-associated antigens by dendritic cells. Here, the authors show that chemically induced tumors develop earlier in DNGR1 KO mice in comparison to WT mice. Furthermore, tumors from DNGR1 KO mice transplanted in WT host are more efficiently rejected than tumors from WT mice. These results show that tumors from DNGR1 mice are more immunogenic and DNGR1 is playing a role in the priming of immunity against chemically-induced tumors. Interestingly, and as previously shown, tumors from DNGR1 KO mice transplanted in DNGR1 host are similarly rejected. Next the authors assessed the neoantigen repertoire of tumors developed in WT, DNGR1 KO and RAG KO mice. They did not find differences in tumor mutation burden, or total numbers of predicted neoantigens. In contrast, they found an accumulation of predicted neoantigens in F-actin binding proteins in DNGR1 KO and RAG KO tumors in comparison to WT tumors, and in agreement with their previous work. Finally, they authors suggest that the antigenic visibility of F-actin binding proteins can be dependent on DNGR1. The manuscript is well written, the data are convincing, and the findings are interesting and highly relevant for cancer immunotherapy as they may help improve the identification of cancer neoantigens for vaccines or cell therapy.

Fig. 1a: Once mice develop tumors, do they grow at the same rate? A growth map similar to the one in Fig 2 would be nice to have.

We thank the reviewer for this comment. We now provide these data as a new Extended Data Fig. 2a.

Fig. 2b: Can the authors comment on the regression of 40% of the tumors from

WT mice after transplantation in WT mice. The authors seem to suggest that the visibility of neoantigens may be different in spontaneous tumors and in transplanted tumors because of differential mechanism of uptake by DCs. Alternatively, the immunosuppressive tumor microenvironment could allow for tumors expressing immunogenic neoantigens to hide from the immune system.

The possibility of immunosuppression raised by the reviewer is correct. Indeed, some tumours escape immunity by suppressing the immune response rather than by getting rid of neoantigens. We ourselves have published that some cancers produce prostaglandin E₂ to avoid rejection (e.g., *Cell* 162, 1257–1270 (2015)). We now highlight this issue in the revised discussion and believe that it only makes our findings all the more remarkable.

It is also worth mentioning that susceptibility to MCA carcinogenesis differs between mouse strains. In our models, we have used WT C57BL/6 mice (so that we could compare to DNGR-1-deficient mice on that background). Many other studies, including those of Schreiber (the father of immunoediting), have used 129/Sv or FVB mice, which are more permissive for carcinogenesis and may therefore allow for greater immunoediting. We point this out more clearly in our revised version.

What is the MHC class I level of expression on the tumors?

We now provide flow cytometric quantification of MHC class I expression by the regressor cell lines generated from DNGR-1 KO mice, as well as (in response to reviewers #2 and #3) showing that their control upon secondary transplantation is fully dependent on CD8⁺ T cells (new Extended Data Fig. 3).

Extended Data Fig. 4: c) is missing

We apologise that it appeared to be missing - it was actually shown as a single panel, bottom right. The figure is now Extended Data Fig. 3e and we have made the panel more prominent.

Fig. 3: The authors should provide more information on their neoantigen prediction method. It does not appear that they used RNAseq to select for expressed mutations. How did they choose a binding IC₅₀ of 150nM as a cutoff? Binders with IC₅₀<50nM are usually considered strong binders. Additionally other factors, such as pMHC stability may play an important role in immunogenicity.

While 50nM has been considered as a threshold for strong binders in some papers, there is currently no consensus on an ideal threshold. To identify a reasonable threshold in our dataset, we compared the distributions of binding affinities in regressors and progressors, reasoning that this would highlight a biologically meaningful value (Fig. 3a), and arrived at a threshold of IC₅₀ <150nM, which also has support in the literature (Tang *et al.* 2020, PMID 33208106). This is now better explained in the Methods section. We did not use RNAseq to filter for expressed genes, which can improve specificity but at the cost on sensitivity. The latter was crucial to maximise given the relatively small sample size. Notably, validating our prediction, 9 of 13 predicted neoantigens bind to either H-2K^b or H-

2D^b, as mentioned in response to the next comment by the reviewer.

Fig. 4: The authors' claim that tumors from DNGR1 KO and RAG2 KO mice are enriched for immunogenic mutations in actin binding proteins would have been much stronger if they indeed showed these mutations are immunogenic. Neoantigen prediction enriches for immunogenic mutations but only ~10% of the predicted mutations are truly immunogenic. They authors could easily evaluate neoantigen immunogenicity by immunizing mice with peptide-based vaccines, or RNA-based vaccines.

This is an important and useful suggestion. We took the predicted FABP-neoantigens from DNGR-1^{KO} regressor sarcoma lines and synthesised them as peptides. We show that 9 out of 13 bind to either H-2K^b or H-2D^b, as predicted. By immunising mice and assessing subsequent CD8⁺ T cell reactivity, we find that at least 2 of these 9 elicit a strong response. We thank the reviewer for encouraging us to do these experiments, which are now presented in a new Fig. 4a-c and have increased the validity of our predictions.

The authors suggest that DNGR1 plays a role in antigenic visibility and show that the growth of MCA205 tumors expressing low levels of mutated spectrin-beta 2 is slower in sGSN ko mice with hyper active DNGR1. How does tumor growth of tumors from DNGR1 KO mice compare after transplantation in WT, DNGR1 KO and sGSN KO mice? This experiment could expand and confirm the findings with MCA205 line. How do the levels of mutated and WT spectrin-beta 2 in the engineered MCA205 line in comparison to the endogenous levels of spectrin-beta 2?

We thank the reviewer for this point. We believe that the levels of expression are made clearer in the replotted Extended Data Fig. 6a. It shows that the low expressor lines have only slightly above endogenous levels of mRNA encoding spectrin-β2.

Reviewer #2

(Remarks to the Author)

While cancer cells mutate and produce neoantigens, not all neoantigens can activate immune responses (PMID: 27198675). It is unknown how cDC1 chooses which neoantigens to cross-present and this is focus of Lim et. al., in this work. In an exciting set of studies, the authors extend Schreiber's immunoediting hypothesis, originally showing roles for NK cells and γδ T cells, now to cDC1 via the F-actin binding necrotic cell sensor signaling receptor DNGR1. The authors show that 3-methylcholanthrene (MCA)-induced tumors derived from DNGR1KO mice are more immunogenic (regressors), suggesting impaired immunoediting. The authors convincingly show that the rejection of these immunogenic tumors was immune dependent (growing in RAG1KO mice) but also importantly dependent on BATF3 expressing cDC1 although the role of DNGR1 in control of these tumors was dispensable. They also show the presence of significantly more neoantigens in the 'less edited' tumors derived from DNGR1KO mice as well as RAG1KO mice, and that the neoantigens in tumors derived from DNGR1KO mice were surprisingly mostly derived from F-actin binding proteins (FABP). They

compare these to microtubule binding proteins as specificity controls, which they find were in turn abundant in RAG1KO but not DNGR1KO. These are novel and exciting findings.

The authors show that a number of the FABPs in DNGR1KO derived tumors contained mutations, concordant with the higher total predicted neoantigens they identified in regressor vs progressor tumors derived from DNGR1KO mice. They establish tumor cell lines from the immunogenic regressor tumors and identify two cell lines, one from DNGR1KO and the other from RAG1KO mice, where two (as opposed to one) mutations were found within spectrin- β and Myh1 (striated muscle myosin heavy chain) in DNGR1KO and RAG1KO mice, respectively. Spectrin- β was notable because of the former identification of a different mutant spectrin- β 2 (R913L) isoform by Schreiber as a neoantigen subject to immunoediting in MCA-induced tumors. The authors use this mutation as a validated neoantigen emerging in less edited tumors to test its emergence under conditions of impaired DNGR1 mediated immune editing. They find that mutant spectrin- β 2 (R913L)-high tumors were not as aggressive as WT spectrin- β 2-high tumors growing to smaller volumes in both WT and sGSNKO mice. Also, only the growth of mutant spectrin- β 2 (R913L)-low tumors was significantly retarded in soluble gelsolin (sGSN)KO mice where DNGR1 activity is enhanced (sGSN inhibits DNGR1).

The study presents evidence in support of the thesis that DNGR-1 engagement during early stages of tumor formation comprises an important barrier against tumor growth and expression of neoantigens. This is an important and novel finding. However, there are major concerns with the conclusions drawn and relate to the incompleteness of the studies as presented. T cell investigations demonstrating priming of anti-tumor CD8 T cells specific for mutant spectrin- β 2 is lacking and would provide valuable evidence. In short, the molecular mechanism driving the process is missing. The major concerns are outlined below.

It is assumed that DNGR1 favors the cross-presentation of antigens associated with the actin cytoskeleton. This would be a significant finding in and of itself. It is presently an assumption that is critical for the conclusions drawn, yet there is no data to support this assumption. It is important to examine whether DNGR1 favors the cross-presentation of antigens associated with the actin cytoskeleton of necrotic dying cells over non-actin associated antigens.

The reviewer is correct in pointing out that we inferred but did not formally show that the DNGR-1 pathway favours the cross-presentation of antigens associated with the actin cytoskeleton. In this revised version, we provide a new Fig. 5 with a large body of new data to address this point. We constructed cells expressing a model antigen (ovalbumin) that we artificially anchored to F-actin by two different orthogonal methods. With either method, we find that those cells, upon dying, are superior substrates for DNGR-1-dependent cross-presentation by cDC1 compared to cells expressing control versions in which F-actin anchoring of ovalbumin is abolished. Notably, *in vivo* immunisation confirms the superiority of the F-actin anchored antigen within dead cells for eliciting CD8⁺ T cell responses. This is even more prominent in mice deficient in secreted gelsolin, connecting directly to the old Fig. 4b (now Fig. 4d) of the manuscript. We thank the reviewer for encouraging us

to extend the study in this direction and agree that it has resulted in a much stronger manuscript.

What is the nature of the two mutations identified in 88D2 within spectrin- β encoding Sptb? Do either of them lead to the R913L mutation? Why was spectrin- β 2 (R913L) and not these mutations (if different) selected for further examination of DNGR1 dependent clearance of tumors bearing FABP neoantigens?

We picked the R913L mutation of spectrin- β described Schreiber and colleagues because we wanted to test an established FABP neoantigen and link our findings to the original work on immunoediting. We felt that this was important because, otherwise, we would only be able to speculate as to whether we were looking at the same process as Dr. Schreiber or something else altogether. We make these points much clearer in the revised text.

There is an equal number of FABP mutations in RAG1KO derived regressor tumors, which diminishes the case that loss of DNGR1 mediated immune editing specifically in FABP and the differences in the number of FABP neoantigens between DNGR1KO vs RAG1KO shown in Fig. 3d is not statistically significant. $P=0.84$.

This might be a misunderstanding. T cells are required for the anti-tumour effect of DNGR-1-mediated cross-presentation by cDC1s. Therefore, having no difference in number of FABP mutations in RAG1^{KO} vs DNGR-1^{KO} derived regressor lines is exactly what you would expect. It fully supports our case that that both DNGR-1 and T cells are key to the immunoediting effect that we observe.

Derivation of high and low expressing WT spectrin- β and mutant spectrin- β 2 (R913L) MCA205 fibrosarcoma cell lines and transplantation into WT, sGSNKO or sGSN/DNGR1DKO mice comprised a series of experiments designed to examine whether DNGR1 engagement promotes effective tumor clearance of tumors harboring FABP neoantigens. This series of experiments open a number of questions, which the authors should have addressed leaving these studies presently as incomplete. Specifically:

-Why is DNGR1 control of mutant spectrin- β 2 tumors evident only for low and not high spectrin- β 2 (R913L)-expressing tumors? F-actin and not proteins associated with F-actin should determine the level of DNGR1 engagement, and as such, the schematic model shown in Extended Data Fig. 7 equates the level of F-actin, and by extension F-actin binding proteins, with the level of cross-presentation; higher levels leading to higher DNGR-1 dependent cross-presentation. However, the results shown in Fig. 4b,c show a correlation between mutant spectrin- β 2 levels and DNGR1-dependent tumor control. Given that F-actin is the ligand for DNGR-1, another interpretation of these results is that low mutant spectrin- β 2 expressing tumor cells present higher F-actin levels to DNGR1. Why do the authors equate the level of F-actin exposed in a dying cell to the level of FABP expression? What is the level of F-actin expression in high vs low spectrin- β 2 expressing tumor groups in Fig. 4b, and how does it correlate with DNGR-1 mediated tumor control?

We believe that some of these issues are addressed by the inclusion in this

revised version of the aforementioned new data (Fig. 5) on antigen anchoring to F-actin. As for the levels of F-actin exposed in high vs low spectrin- β 2 expressing tumors, we know this does not vary substantially across cell lines. Actin is very abundant and there are so many FABPs that manipulation of any given one does not affect overall F-actin levels. We have also replotted Extended Data Fig. 6a (previously Ext. Data Fig. 8a) as also requested by Reviewer #1. We show that the expression of spectrin- β 2 between WT and mutant lines is similar, and therefore, we infer that the levels of spectrin/actin are unlikely to account for differences in tumour growth.

-Is tumor control in these experiments dependent on CD8 T cells? This is important to the message that DNGR1 dependent immunoediting shapes visibility of tumors to CD8 T cells.

We thank the reviewer for this comment. In the previous version, we showed that regressor tumour control was lost in RAG KO mice. In this version, we extend that conclusion by carrying out CD8 depletion experiments (as also suggested by Reviewer 3). The new data (Extended Data Fig. 3b) show that regressor tumour control is fully dependent on CD8⁺ T cells.

-The results presented suggest that mutations render an FABP a neoantigen but this needs to be demonstrated directly. Does mutant spectrin- β 2 specifically generate an anti-tumor CD8 T cell response? Is this response specific to mutant and not WT spectrin- β 2? Is this CD8 T cell response dependent on DNGR1? Can it be mapped to the spectrin- β 2 R913L mutation by testing the T cell response to peptides overlapping the R913 residue.

We thank the reviewer for pointing out that we did not directly demonstrate that our predicted FABP neoantigens were immunogenic. This is similar to a comment by reviewer #1. We have now extended the study by showing that 9 out of 13 putative FABP neoantigens bind to either H-2K^b or H-2D^b, as predicted, and at least 2 of these 9 elicit a strong CD8⁺ T cell response upon mouse immunisation (new Fig. 4a-c).

As for mutant spectrin- β 2, we confirm Dr. Schreiber's data that it binds H-2D^b and that it is immunogenic in C57BL/6 mice. We also confirm that the wild type spectrin- β 2 version does not bind to H-2D^b, which is why it is non-immunogenic (new Figs. 4a-c).

Other points:

- Which pairwise comparison does P=0.002 using unpaired test refer to in Fig. 1a? Consider moving Extended Data Fig. 2b adjacent to Fig. 1a.

We now show a table with all the pairwise comparisons. To maximise clarity, as suggested, we have moved Extended Data Fig. 2b adjacent to the Kaplan-Meier plot forming part of Fig. 1a.

- Consider moving Extended Data Fig. 2d adjacent to Fig. 1b.

We have moved Extended Data Fig. 2d to new Fig. 1b.

- Statistical analyses comparing growths of spectrin- β 2 (R913L)-high vs WT spectrin- β 2-high in Fig. 4b should be drawn.

Thank you – apologies for this omission. This is now added – Fig. 4b is now the new Fig. 4d.

- The authors need to indicate the significance of the P-values shown in Fig. 3c-e in the figure legend.

Thanks, we have now added a statement in the figure legend regarding the significance of the annotated P values on the panels.

- Tumor growth curves of high vs low spectrin- β /spectrin- β R913L expressing MCA205 cells should be shown on the same graph with statistics drawn to formally exclude a role for spectrin- β 2 expression levels in tumor immunogenicity and/or tumor growth.

In this particular case, they were done as separate experiments (not at the same time), hence presented as such.

- Lines 181-182 "...immunoedited escapees that have lost mutated FABP epitopes, which we detect in WT but with lower prevalence in DNGR1KO mice" is confusing and should be re-written. As stated, it is not concordant with the data in Fig. 4a which show prevalence of mutations in FABP encoding genes in DNGR1KO compared to WT.

We apologise for the impenetrable wording. We have now broken down the sentence and, hopefully, made our argument clearer.

- Extended Fig. 5: Does the genomic mutational landscape include both progressive and regressive primary tumor cell lines? Could this be the reason why there is no significant variation in tumor mutational burden?

Yes, the analysis includes both progressors and regressors to be as complete as possible. We have also done the comparison of regressors vs progressors and found no significant differences between the two subgroups.

Reviewer #3

(Remarks to the Author)

In this study, the authors perform a series of in vivo experiments to conclude that cross-presentation of dead cell-associated antigens is necessary to shape the neoantigenic landscape of tumor immunity. While the authors use a relatively logical approach yet the initial enthusiasm for this study is dampened by its very broad and generalized assumptions based on a limited research design or lack of supportive data. It seems that the study frequently overinterprets the impact of DNGR1 and makes very strong conclusions about its "causative role" in defining the neoantigenic landscape. In its present form, this study seems too preliminary for publication. Below I elaborate my major concerns.

1. The authors don't seem to present clear evidence that supports this very strong statement (and other similarly strong statements throughout the manuscript): "Priming of anti-cancer cytotoxic CD8+ T cell depends largely on the proficient (neo-)antigen cross-presentation (XP) and cross-dressing capabilities of type 1 conventional dendritic cells (cDC1s)." While cDC1 are definitely important, depending on the context, cDC2, pDCs, moDCs and even macrophages have

been found to be important for neoantigen presentation. This very generalized assumption about cDC1 seems to pave the way for a research design that is completely biased toward cDC1 in its entire study. If this is true, then the authors need to use depletion strategies to exclude the roles of cDC2, pDCs and macrophages before completely concentrating their study on cDC1.

An overwhelming body of data indicates that cDC1s are non-redundant for anti-tumour immunity in mice and that cDC1 abundance in tumours correlates with better outcomes and response to immunotherapy in cancer patients. We now make this clearer in our revised Introduction. However, the reviewer is correct in pointing out that other APCs may also be important in neoantigen cross-presentation. Where we respectfully disagree with the reviewer is that this therefore means that we have to exclude a role for cDC2, pDCs and macrophages in our study. We would argue the opposite. Indeed, if cDC2, pDCs and macrophages are substituting for cDC1s, this only serves to decrease the likelihood that we would see any DNGR-1 effect at all, thereby making our results all the more notable. In other words, given the premise that multiple cell types can function as APCs in anti-tumour immunity, is it not even more remarkable that, against all odds, we see immunoediting shaped by a single cDC1-restricted receptor?

2. The study makes generalized assumptions about DNGR1 being the most important player in the phagocytosis/efferocytosis system. There are many receptor-ligand systems important for phagocytosis/efferocytosis like CD47-SIRP1a, calreticulin-LRP, complement receptors, several scavenger receptors on macrophages etc. that can be equally important. The authors haven't provided any comparative evidence to support their claims since all experiments were immediately done with DNGR1-KO mice. This is too preliminary. For such broad conclusions as drawn by the authors, a lot more comparative evidence needs to be provided before settling on experiments with the "best hit", which may very well be DNGR1 or perhaps another target?

For clarity, our previous work demonstrates that DNGR-1 does not recognise apoptotic cells, hence is not an efferocytosis receptor. Indeed, DNGR-1 only detects "necrotic" cells that have lost membrane integrity and exposed F-actin. Furthermore, DNGR-1 is dispensable for the actual uptake of necrotic cells by cDC1s. Rather, in a paper published in *Nature Immunology* in 2021 (Canton *et al.*, PMID 33349708), we demonstrated that DNGR-1 signals after uptake of dead cell debris to induce phagosomal/endosomal rupture, favouring cross-presentation. In other words, DNGR-1 is unique not because it promotes uptake of dead cells by cDC1 but because it is the only receptor reported to-date that specifically regulate cross-presentation of dead cell-associated antigens. We feel that this justifies the focus of our study on DNGR-1 as our objective is to assess how cross-presentation of dead cancer cell antigens impacts CD8⁺ T cell immunity to cancer. We hope that this is clearer in our revised Introduction.

Irrespective of the above, our reply is similar to that in response to point 1. All the other receptors mentioned by the reviewer and expressed on various cell types should compensate for DNGR-1 loss in our KO mouse model. As such, is it not remarkable that we see immunoediting being shaped by DNGR-1?

3. This conclusion “DNGR-1 restrains carcinogenesis” is not entirely supported by the data. The analyses is restricted to one class of chemical carcinogen i.e., MCA. However, most cancers of carcinogenic-origin are not induced by this class of chemical carcinogen. Most cancers are driven by tobacco-derived carcinogens (e.g., lung cancer, head & neck cancer, some GI/GU cancers) or UV radiation (e.g., melanoma). Thus the authors do not use an appropriate carcinogen for such generalized conclusions. While it is true that some classical experiments in past were performed with MCA yet this choice is outdated for current contexts. Even for sarcoma in humans, the high neoantigenic version might be a byproduct of dMMR or MSI-like pathology rather than MCA per se. Thus, this part of the study design may not be up-to-date and/or clinically relevant.

We agree with the Reviewer’s point on the limitations of the MCA carcinogenesis mouse model and we did not mean to imply that it is clinically-relevant. Rather, our paper should be seen as a basic immunology proof-of-principle study to ask if cross-presentation by APCs can shape anti-cancer immunity and act as a selection “funnel” for certain neoantigen classes. Whether it does so in the clinic remains to be established and is certainly likely to differ from one cancer to another. However, if published, our study would provide an immunological conceptual framework for such investigations. We clarify in our revised Discussion section that we show that cross-presentation and DNGR-1 can shape the immune visibility of tumours but to what extent it does so in clinical settings remains to be established.

4. A large majority of neoantigen-enriched CRC have MLH1 disruptions behind it, creating dMMR or MSI-like pathology. Such MLH1 disruptions are not necessarily chemical carcinogen-driven. The authors need to knock-out MLH1 in cancer cells to truly connect CRC-relevant neoantigen drivers to DNGR1 system (or others, as suggested above).

We respectfully disagree with the reviewer as we did not set out to study CRC. Our AOM/DSS model of colon carcinogenesis is included in a main figure as a single panel only as another example of the fact that DNGR-1-deficient mice are more susceptible to tumour induction. We believe that this extends the data obtained with the MCA model of fibrosarcoma induction but it is not intended as a model of human CRC.

5. Lack of use of physiologically-relevant carcinogenic or genetic drivers of neoantigen landscape in this study might explain some of the confusing observations on overall mutational burden or neoantigen quality that the authors observe, which do not align with what has emerged in the clinical immunotherapy research in the last few years.

Again, as mentioned above, our work is intended as a basic immunology study to establish whether cross-presentation by APCs can shape anti-cancer immunity. We believe that our data show that this is the case and would argue that it is important to demonstrate it in mice even if we use carcinogens other than those that cause cancer in humans.

As for the comment about genetic drivers, the issue with genetically-engineered models of mouse cancer is that they do not accumulate neoantigens. As such, they cannot be used to study and assess immunoediting.

6. The author's results of few dominant neoantigens derived from F-actin coding genes driving the entire system do not align with the clinical reality of anticancer immunity relevant neoantigenic landscape i.e., physiological anticancer immunity in many cases is driven by stochasticity within the neoantigenic landscape rather than a few selected dominant neoantigens. This can again indicate that the author's setup may not be physiologically relevant.

We respectfully disagree with the reviewer. As stated in our Discussion, T cells specific for mutated FABPs have been found in patients with cancer (refs. 34-36). Furthermore, our previous analysis showed that the prevalence of mutations in FABPs correlates with better outcomes in cancer patients bearing low intratumoural levels of sGSN transcripts (ref. 18), which correlates perfectly with the results presented in the current manuscript. However, we are not trying to argue that immune selection against mutated FABPs happens in every cancer. In fact, we have been very cautious to point out that this is not an "all-or-none" effect but a quantitative one, as also highlighted by Reviewer 4. For many reasons (including the ones suggested by this reviewer in points 1+2), we expect the prevalence of antigens derived from mutated FABPs to differ across patients and cancers. This, and other factors (e.g., MHC diversity), will contribute to the heterogeneity and apparent stochasticity of the neoantigenic landscape in human tumours.

7. Immunoediting is only concluded on the basis of transplantation rejection/latency in WT mice – which might be a very old school way of analyses. It would be more appropriate to subsequently deplete the relevant immunoediting players via antibody-based depletion in WT mice e.g., anti-CD8, anti-CD4 or NK1.1 to see if rejection/latency can be ameliorated thereby truly proving immune-driven responses rather than any transplantation-relevant "artefacts".

Even though it might be "old school", immunoediting was discovered and proven using the approach we utilised in this study. We do not know of another approach and do not understand the suggested experiment or the reference to transplantation-relevant "artefacts" given that we are using chemical carcinogenesis. When would the Reviewer have us use anti-CD8, anti-CD4 or NK1.1 in WT mice? MCA carcinogenesis takes 80-150 days and, therefore, makes such depletion experiments impossible due to development of a mouse antibody response against the depleting antibodies. However, even if feasible, we are unclear as to what such an experiment would tell us about immunoediting. Why is it substantially different from our use of RAG KO mice in Fig. 1?

8. BATF3 KO mice may not be appropriate. While it is presented as a model where only cDC1 are disrupted yet it has been repeatedly proven recently that BATF3 also has a role in T cell biology and thus BATF3 KO also have direct T cell defects that may have nothing to do with cDC1 or their interface with T cells. This is an

important problem that requires urgent attention considering the cDC1-centered conclusions of this study.

We agree with the reviewer that BATF3-deficient mice also have defects in T cells and we now explicitly refer to this point in the revised Results section. However, it is important to note that those mice are used only in Fig. 1a and are not essential to the narrative. If the data with BATF3-deficient mice are felt to be objectionable, we could leave them out. However, we believe that they add value, despite the caveat cited by the Reviewer.

9. The authors have not provided any detailed immunophenotyping data for the tumors for various myeloid (macrophages, DC subsets, and their polarization states) or T cell subsets (Tregs, CD8T subsets, exhausted/cytotoxic cells etc.) thereby making it impossible to understand the immunological basis of these phenotypes. This makes it impossible to fully understanding possible confounding immune-factors.

We did not mean to imply that editing of the tumour FABP mutational landscape explains all of our data. We agree that there may be additional confounding immune factors and now explicitly mention this in the revised discussion. We thank the reviewer for reminding us of the need to be more nuanced in our interpretation.

10. The authors draw a specific conclusion about CD8+T cell priming without specifically proving this via anti-CD8 depletion or genetic targeted disruptions in CD8+T cell compartments. This is essential.

In the previous version, we showed that regressor tumour control was lost in RAG KO mice. In this version, we extend that conclusion by carrying out CD8 depletion experiments (as also suggested by Reviewer 2). The new data (Extended Data Fig. S3b) show that regressor tumour control is fully dependent on CD8⁺ T cells.

11. Any immune-checkpoint blockade based analyses is missing in this study which makes it impossible to connect immunotherapy-relevant processes to this study's conclusions.

The reviewer is correct that we did not ask whether immunotherapy can also lead to cross-presentation-driven immunoediting. This would be interesting to assess, especially with sequencing of human tumours comparing pre- and post-treatment. We hope to have an opportunity to carry out such a study in future, subject to having a means of dealing with HLA polymorphisms (see also reply to point 13).

12. The authors need to quantify cell death in vivo and prove to what extent are the antigens derived via dead/dying cells via phagocytosis by only cDC1 and no other phagocytic innate immune cells or even B cells also present in the tumor. How are cDC1 favored over other myeloid cells or B cells for a specific as well as preferential uptake of neoantigens? How are these cells the only one that form an interface with CD8+T cells, even though other cells are reported to do the same? These questions create some confusion regarding this study's conclusions.

We refer to our reply to points 1+2.

13. The study lacks clinical immunotherapy data analyses even though the hypothesis is derived from a clinical therapeutic problem. The authors show TCGA analyses, but these data are not relevant since large majority of TCGA patients didn't receive immunotherapy. This creates doubts on clinical applicability of these results for immunotherapy. Proper clinical data analyses with immunotherapy clinical trials, needs to be added to validate the clinical significance of these findings.

Is the reviewer suggesting that we interrogate cancer mutanomes from patients that respond or not to immunotherapy to see if there is enrichment in the former group for predicted neoantigens derived from mutated FABPs? This is an interesting suggestion. However, one complication is HLA polymorphisms. In our mouse data, all animals were H-2^b and we could use an algorithm to predict binders to H-2K^b and H-2D^b in order to identify predicted neoantigens. This will not be feasible with patient data unless we accumulate cohorts large enough to allow us to have sufficient individuals of a given haplotype. We are in discussions with clinical colleagues to assess feasibility.

Reviewer #4

(Remarks to the Author)

The manuscript is clearly written and the findings are supported by clear state-of-the-art experimental approaches. The authors elaborate on the role of DNGR-1/CLEC9a on immune-editing of FABP neoantigens in MCA-induced tumors. They convincingly show that Dngr1 KO mice compared to wt exhibit decreased priming of FABP neoantigen-specific T cells without markedly affecting the response to other neoantigens. Reconstruction experiments using MCA205 fibrosarcoma cell lines expressing high or low levels of WT or mutant-spectrin-β2 confirmed that the DNGR-1 immunomodulation is not an “all-or-none” phenomenon but rather quantitatively modulated.

Conceptually and experimentally I have no major comments.

Experimentally, in view of the focus of the paper on DNGR-1 the authors may comment on the fact they did not include littermate controls for Dngr1 KO, though I admit that the experiments comparing “Dngr1 KO to wt strains included the use of mice that were co-housed for at least 3 weeks to eliminate any microbiota-dependent effects”. “All mice were bred in-house”. Have the authors used in particular experiments such littermate mice to exclude potential confounding genetic differences?

Thank you for raising this issue. We have not carried out experiments with littermates. However, our two strains of DNGR-1^{KO} mice have been backcrossed to C57BL/6 and subject to regular genetic monitoring. We are completely confident that our DNGR-1 KO strains are 100% comparable to WT C57BL/6 mice even if they are not littermates.

We now provide the following details in a revised version of the manuscript:

DNGR-1 EGFP KI mice were generated in 129S6/C57BL/6 F1 ES cells and the expression of NK1.1 (the Klrk1c gene "a" allele) was linked to Clec9a deficiency in early backcross KO mice indicating that the homologous recombination step targeted the chromosome of C57BL/6 origin. This strain has subsequently been backcrossed a total of 20x to C57BL/6J before crossing again to homozygosity. Genetic monitoring (carried out at Transnetyx) confirmed that three N20 mice (one each at generations F4, F5 & F6) were 100% BL/6 (vs 129) when tested using a panel of 120 distinguishing SNPs. Three F6 mice were also 100% BL/6J (vs BL/6N) when tested using a panel of 24 distinguishing SNPs.

DNGR-1 iCre KI mice were generated in the Primogenix PRX-B6N ES cell line. The strain was backcrossed 10x to C57BL/6J before crossing again to homozygosity. Four N10F11 animals were tested as 99.9-100% BL/6J by the MiniMUGA 10K SNP panel at Transnetyx.

The authors could spend a paragraph for a less specialized readership explaining why DNGR1-mediated acquiring of FABP neoantigens is a cross presentation event (and not a cross dressing).

We thank the reviewer for highlighting this point, which we hope is now better explained in the revised Introduction.

Response to reviewers (in blue)

Reviewer #1

(Remarks to the Author)

I appreciate the authors' thoughtful responses to my comments and the revision made to the manuscript.

My comments are below:

The authors showed that 2/9 predicted mutated FABP neoantigens from DNGR1 KO mice are immunogenic (fig4b). The low frequency of immunogenic mutations is not surprising, but it also somewhat weakens the authors' claim that there is accumulation of FABP neoantigens in DNGR1 KO mice due to reduced immunoediting. The n is small and drawing such a conclusion is difficult. Did the authors examine the immunogenicity of the mutated FABPs in WT mice? if none were immunogenic it would strengthen the conclusion.

In addition, I disagree with the authors' decision not to use RNA-seq as a filter for neoantigen expression (Fig 3). If this data are available, it should be included, as immunoediting can occur through the selection of cells that downregulate the mutant protein. Again, it is difficult for me to see an association between the enrichment of predicted FABP neoantigens and reduced immunoediting in absence of expression - especially since only a minority of the predicted neoantigens are actually immunogenic. The data from Fig. 4b provides some additional nuance, as it seems that DNGR1 and sGSN may be more critical in cross-presentation of FABP neoantigens expressed at lower level.

We thank the reviewer for his/her perceptive remarks. We agree with the reviewer that the n is small. For logistic, financial and ethical reasons (as well as good research practice), we had carried out power calculations on the probability of finding immunoediting at the level of FABP neoantigen selection before starting the study. Based on those calculations, we generated 79 primary fibrosarcomas in 3 different mouse strains. These had to then be transplanted into WT, immunodeficient mice or mice depleted of CD8⁺ T cells, thereby using up to 1000 mice over several years. We are happy with the fact that this was not in vain - our comparisons revealed a statistically significant increased prevalence of predicted FABP neoantigens in tumours from DNGR1^{KO} mice - but we take the reviewer's point that our conclusion could be further strengthened.

An interesting suggestion (if we understood correctly) is to assess the lack of immunogenicity of the predicted mutated FABPs that are found in 5 tumours from WT mice. However, we are not sure how those data would be interpreted as we fear that, again, the n is small (only 5) and it is therefore possible that lack of detectable immunogenicity could occur by chance. Furthermore, it is possible that our failure to detect a larger number of immunogenic peptides within the pool of nine peptides that bound to MHC class I was due to limitations of our in vivo immunisation and re-stimulation assay. It would be premature to conclude that the 7 peptides that did not show a re-stimulation response are non-immunogenic in the tumour context.

The other point of the reviewer is that downregulation of neoantigen expression could also underlie immunoediting. We carried out exome sequencing of each tumour but, regrettably, not RNAseq. In retrospect, this would have been useful as a proxy for the level of expression of the FABP neoantigens - indeed, mRNA abundance can correlate with neoantigen immunogenicity (as shown in Fig. 3D of Wells *et al. Cell* **183**, 818-834.e13 (2020)). However, filtering by RNAseq also reduces sensitivity, as peptides with lower expression might still be immunogenic (as shown in the same figure in Wells *et al.*). Given our small sample size, applying a stringent expression-based filter reduces detection power in our (small) dataset although we agree that filtering could be useful in future with larger sample sizes.

Irrespective of the above, it is important to note that downregulation of expression, if it occurs, strengthens rather than diminishes our findings: if tumours can escape immune attention by downregulating expression of mutant FABP, the selection against such neoantigens is attenuated and, hence, our probability of finding evidence of selection at the level of the mutanome is decreased. Yet, we find such mutations, selectively in FABPs (Fig. 3). As such, we believe that the analysis we carried out (without RNAseq filtering), if anything, is biased against us, thereby increasing the robustness of our conclusions.

In sum, we agree with the reviewer that there are additional layers of complexity to immunoediting to be investigated in further studies, with more tumours and RNAseq filtering. We have now revised the second paragraph of the discussion to bring up these points.

(Remarks on code availability):

link above does not work - https://github.com/FrancisCrickInstitute/DNGR1_XP

We apologise for the broken link. This has now been fixed

Reviewer #2

(Remarks to the Author)

The authors have skillfully addressed my critiques, significantly enhancing the manuscript. The strategic use of ovalbumin variants (LA-OVA and mutLA-OVA) to modulate F-actin binding and its impact on the DNGR-1-dependent cross-presentation, by both Mutu DC and primary cDC1, is a compelling approach, clearly demonstrated in Fig. 5a-d. The authors extended their results to *in vivo* immunization showing the superiority of F-actin anchored antigen in eliciting CD8 T cell responses and included immunization of secreted gelsolin deficient mice (Fig. 5f,g). The presented data are robust and engaging, adding substantial value to the work.

The inclusion of the CD8 T cell depletion assay is a notable strength, providing clear

evidence of the direct role of CD8 T cells (Ext. Fig. 3b). Additionally, the RMA-S stabilization assay effectively supports the direct interaction of the FABP neoantigen with H-2kb/H-2db (Fig. 4a,b), and validation in vivo (Fig. 4c) further bolstering the study's conclusions. These well-executed additions have strengthened the manuscript, making it a strong contribution to the field.

We thank the reviewer for her support and encouraging remarks.

(Remarks on code availability):

The link was broken. I cannot comment as to whether I could assess the code's suitability.

We apologise for the broken link. This has now been fixed

Reviewer #4

(Remarks to the Author)

The authors have adequately responded to my remarks.

We thank the reviewer for his/her support.